# Gene-level alignment of single-cell trajectories

Dinithi Sumanaweera [1,2,12], Chenqu Suo [1,3,12], Ana-Maria Cujba[1], Daniele Muraro[1], Emma Dann [1], Krzysztof Polanski [1], Alexander S. Steemers [1,4], Woochan Lee[1,5], Amanda J. Oliver[1], Jong-Eun Park [1,6], Kerstin B. Meyer [1], Bianca Dumitrascu [7,8] & Sarah A. Teichmann [1,9,10,11] ✉

Single-cell data analysis can infer dynamic changes in cell populations, for example across time, space or in response to perturbation, thus deriving pseudotime trajectories. Current approaches comparing trajectories often use dynamic programming but are limited by assumptions such as the existence of a definitive match. Here we describe Genes2Genes, a Bayesian information-theoretic dynamic programming framework for aligning single-cell trajectories. It is able to capture sequential matches and mismatches of individual genes between a reference and query trajectory, highlighting distinct clusters of alignment patterns. Across both real world and simulated datasets, it accurately inferred alignments and demonstrated its utility in disease cell-state trajectory analysis. In a proof-of-concept application, Genes2Genes revealed that T cells differentiated in vitro match an immature in vivo state while lacking expression of genes associated with TNF signaling. This demonstrates that precise trajectory alignment can pinpoint divergence from the in vivo system, thus guiding the optimization of in vitro culture conditions.

Single-cell technologies, especially single-cell RNA sequencing (scRNA-seq), have revolutionized our understanding of biology and opened up new avenues of research[1]. Their ability to observe thousands of genes per cell simultaneously enables the description of transitional cell states and dynamic cellular processes (for example differentiation/development; response to perturbations). The computational task of deriving a 'timeline' for a dynamic process (for example based on transcriptomic similarity) is referred to as 'pseudotime trajectory inference'[2,3]. One key challenge is how to compare two (or more) trajectories, for example in control versus drug treatment groups, or in vitro cell differentiation versus in vivo cell development (Fig. 1a) where identifying differentially regulated genes can guide us to refine in vitro cell differentiation.

Trajectory comparison poses a time series alignment problem, which is addressable using dynamic programming[4] (DP). A popular DP algorithm to align two single-cell trajectories is dynamic time warping[5] (DTW). The goal is an optimal mapping (pairwise sequential correspondences between the time points of two single-cell trajectories),

[1]Wellcome Sanger Institute; Wellcome Genome Campus, Hinxton, Cambridge, UK. [2]Theory of Condensed Matter, Cavendish Laboratory, Department of Physics, University of Cambridge, Cambridge, UK. [3]Department of Paediatrics, Cambridge University Hospitals; Hills Road, Cambridge, UK. [4]Princess Máxima Center for Pediatric Oncology, Utrecht, Netherlands. [5]Department of Biomedical Sciences, Seoul National University, Seoul, Korea. [6]Graduate School of Medical Science and Engineering, Korea Advanced Institute of Science and Technology (KAIST), Daejeon, Korea. [7]Department of Statistics, Columbia University, New York, NY, USA. [8]Irving Institute for Cancer Dynamics, Columbia University, New York, NY, USA. [9]Cambridge Stem Cell Institute, Jeffrey Cheah Biomedical Centre, Cambridge Biomedical Campus, University of Cambridge, Cambridge, UK. [10]Department of Medicine, University of Cambridge, Cambridge, UK. [11]Co-director of CIFAR Macmillan Research Program, Toronto, Ontario, Canada. [12]These authors contributed equally: Dinithi Sumanaweera, Chenqu Suo. ✉e-mail: sat1003@cam.ac.uk

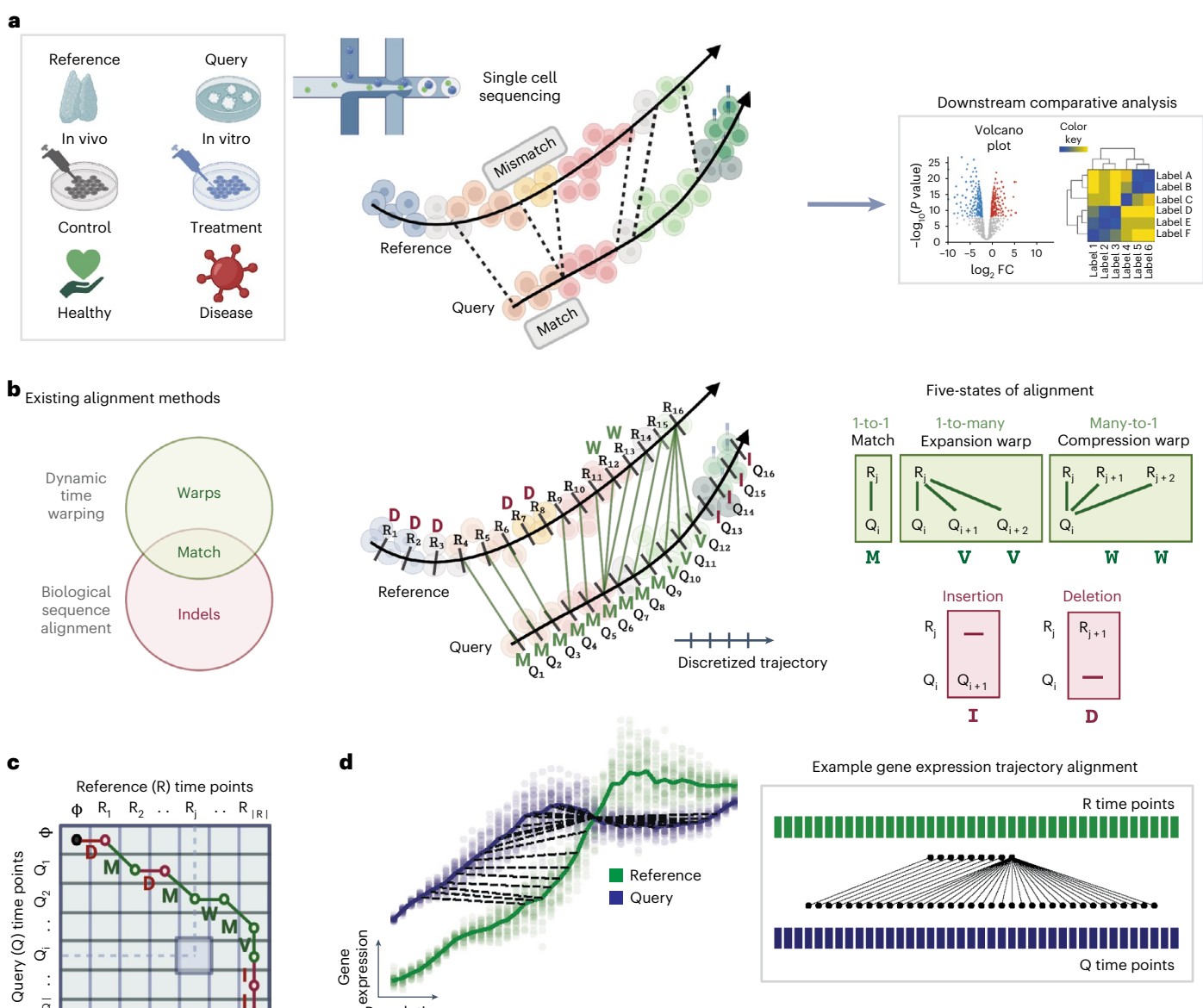

**Fig. 1 | Computational alignment of single-cell transcriptomic trajectories.**
**a**, Schematic of the concept of single-cell trajectory alignment. The input is single-cell transcriptomic data of a reference and a query that change dynamically (left), for example, in vivo cell development and in vitro cell differentiation, control and drug-treated cells in response to perturbation, responses to vaccination or pathogen challenge in healthy and diseased individuals. Aligning the reference and query can capture matches and mismatches (middle), supporting further downstream analysis (right).
**b**, Different alignment states and their theoretical origins. Dynamic time warping and biological sequence alignment complement each other[5,15,16] when capturing matches (including warps) and indels (left). An alignment (nonlinear mapping) between time points of the discretized reference (R) and query (Q) trajectories shown in **a** (middle). Between a reference time point $R_j$ and query time point $Q_i$, there may exist five different states of alignment: 1-1 match (M), warps (1-to-many expansion (V) or many-to-1 compression (W) match) and mismatch (insertion (I)/ deletion (D) denoting a significant difference in one system compared to the

other) (right). **c**, Example alignment path across a pairwise time point matrix between R and Q trajectories. Diagonal lines (green) refer to matches; vertical lines refer to either insertions (red) or expansion warps (green); horizontal lines refer to deletions (red) or compression warps (green). Any matrix cell (*i*, *j*) denotes the pairing of two $R_j$ and $Q_i$ time points. **d**, An example gene alignment generated by the Genes2Genes framework. Interpolated log1p-normalized (per-cell total raw transcript count normalized to 10,000 and log1p-transformed) expression (*y* axis) between reference (green) and query (blue) against their pseudotime (*x* axis) (left). The bold lines represent mean expression trends and faded data points are 50 random samples from the estimated expression distribution at each time point. Black dashed lines visualize matches (including warps) between time points. Corresponding nonlinear mapping between R and Q time points shown in the left (right). Corresponding five-state alignment string where subsequences over [M, V, W] and [I, D] denote matched regions and mismatched regions, respectively (bottom). Illustrations in **a–c** were created using BioRender (https://biorender.com).

which captures matched and mismatched cell states. Several studies[6–11] including the widely-used CellAlign[7] employ DTW to analyze correspondences and timing differences[12]. Current practice is to first interpolate gene expression time series, and then minimize the Euclidean distance of expression between the matched time points to find their

optimal alignment. While DTW is a powerful approach, its main limitations are: (1) the assumption that every time point in reference matches with at least one time point in query; (2) the inability to identify mismatches (unobserved state or substantial differences between two series) occurring as insertions and/or deletions (indels); and (3) a

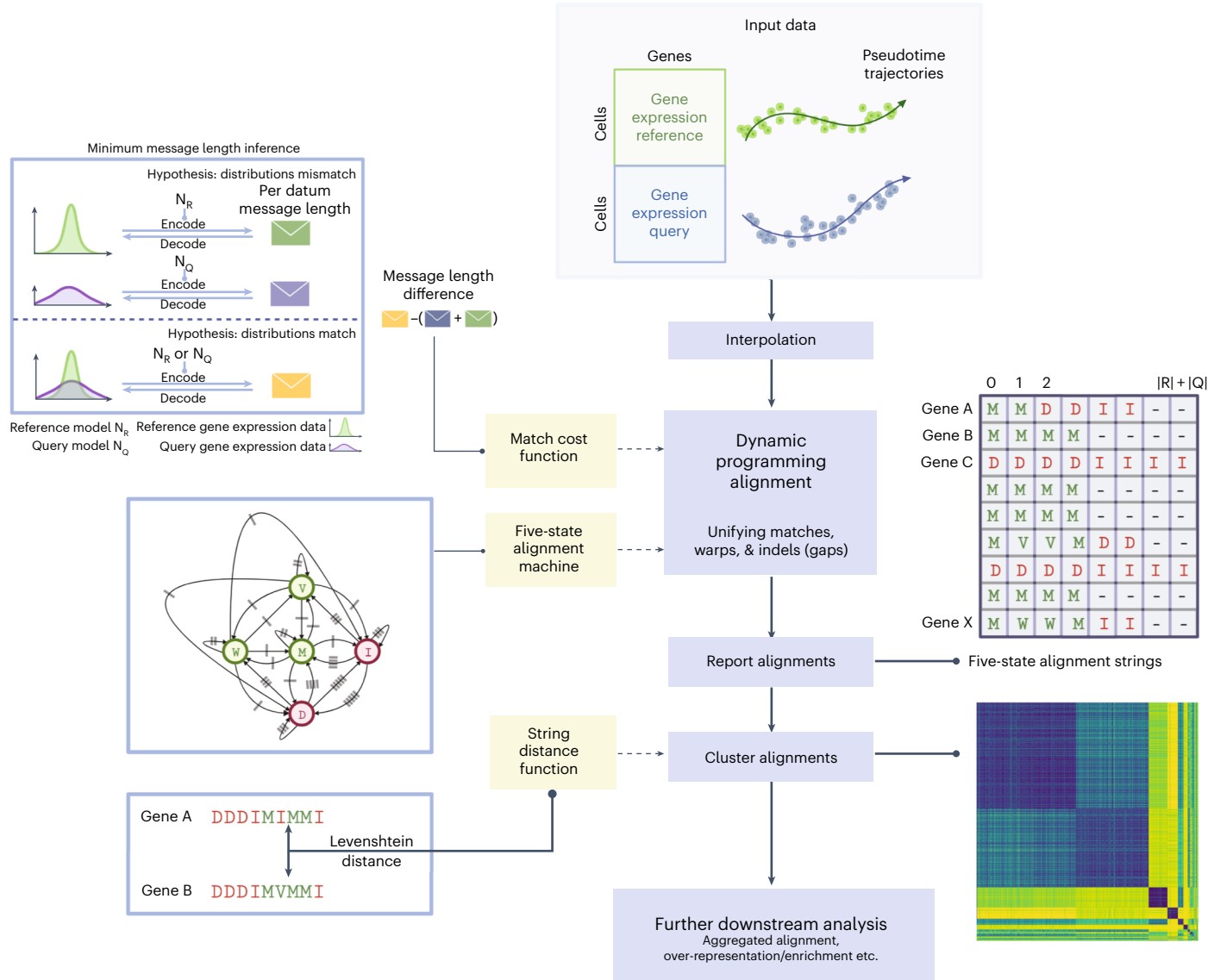

**Fig. 2 | Overview of the Genes2Genes alignment framework and workflow for comparing single-cell transcriptomic trajectories.** Given log1p-normalized cell-by-gene expression matrices of a reference (R) and query (Q) and their pseudotime estimates, G2G infers individual alignments for all genes of interest. It first interpolates data by extending mean-based interpolation in Alpert et al. (2018)[7] to distributional interpolation and then runs Gotoh's DP algorithm[16] adapted for all the five alignment states (M, W, V, I, D) defined in Fig. 1b. All reported alignments are then clustered and used to deliver statistics on the overall alignment between R and Q, supporting further downstream analyses. The DP algorithm utilizes a match cost function defined under MML[26] inference framework (top left). Given a hypothesis (model) and data, MML defines the total message length of encoding them for lossless compression along an imaginary message transmission. G2G defines two hypotheses: (1) $\Phi$: $R_j$ and $Q_i$ time points mismatch and (2) A: $R_j$ and $Q_i$ time points match. Under $\Phi$, the message length is the sum of independent encoding lengths of their interpolated expression data

and corresponding Gaussian distributions. Under A, the message length is the joint encoding length of their interpolated expression data under a single Gaussian distribution (either of $R_j$ or $Q_i$). The match cost is computed as the difference of A and $\Phi$ per-datum encoding lengths. The DP algorithm incorporates a symmetric five-state machine which can generate a string over the alphabet, $\Omega$ = [M, W, V, D, I] describing the optimal sequential alignment states (Fig. 1b) between R and Q time points (middle left). Each arrow represents a state transition. Arrows with the same hatch mark implies equal probability of state transition. G2G computes a pairwise Levenshtein distance matrix across all five-state alignment strings to cluster genes of similar alignment pattern (bottom left). Example output of five-state alignment strings for all genes (top right). Example clustermap showcasing the clustering structure of alignments resulted from agglomerative hierarchical clustering (bottom right). The color represents the Levenshtein distance. Illustrations were created using BioRender (https://biorender.com).

distance metric that only evaluates the difference of means rather than the distributions of gene expression.

Warps and indels are fundamentally distinct (Fig. 1b,c), as high-lighted in discussions[13,14] about integrating DTW with the concept of gaps in sequence alignment[15,16]. Both matches and mismatches between trajectories inform our understanding of temporal gene expression dynamics, specifically patterns such as divergence and convergence (Fig. 1d). A mismatch either implies an unobserved state or differential

expression (DE), indicating a transit through a different cell state in one of the systems, or when cells in one condition have a significantly different distribution of expression for some genes. On the other hand, matches imply similar cell states, with warps indicating differences in their relative speeds of transition. Approaches such as analyzing correlation or mutual information of binned expression along pseudotime will have limited accuracy in detecting warped/unobserved states, as it only assumes one-to-one mappings. In contrast, alignments can

properly identify DE genes between trajectories. Laidlaw et al.[17] also showed that trajectory alignment successfully captures DE genes undetectable by non-alignment methods[18,19].

Here we present Genes2Genes (G2G; Fig. 2 and Methods), a new framework for aligning single-cell pseudotime trajectories of a reference and query system at single-gene resolution. G2G utilizes a DP algorithm that handles matches and mismatches in a formal way, by combining the classical Gotoh's algorithm[16] with DTW[5] and employing a Bayesian information-theoretic scoring scheme to quantify distances of gene expression distributions. This overcomes ad hoc thresholding[7] and/or post hoc processing of typical DTW outputs (as in TrAGEDy[17], the recent advancement built on CellAlign[7]). G2G (1) generates descriptive gene alignments; (2) identifies gene clusters of similar alignment patterns; (3) derives aggregate, cell-level alignment across all or subset of genes; (4) identifies genes with differential dynamic expression; and (5) explores their associated biological pathways.

We validate G2G's ability to accurately capture different alignment patterns in simulated datasets, benchmarking against CellAlign[7] and TrAGEDy[17] (the current state-of-the-art of single-cell trajectory alignment) and demonstrate gene-level alignment between two conditions in a published real dataset[20]. We further utilize G2G in a healthy versus disease comparison in idiopathic pulmonary fibrosis (IPF)[21]. Finally, we show how G2G aligns in vitro and in vivo T cell development, finding that TNF signaling in in vivo T cell maturation is not recapitulated in vitro and validate G2G's use for optimizing in vitro cell engineering.

## Results

### Genes2Genes aligns trajectories using dynamic programming

G2G is a new DP framework to infer and analyze gene trajectory alignments between a single-cell reference and query. Given a reference sequence $R$ ($\{R_i\}_{i=1}^{|R|}$) and query sequence $Q$ ($\{Q_i\}_{i=1}^{|Q|}$), two discrete series of time points, a computational alignment between them can inform us of the one-to-one matches, one-to-many matches (expansion warps), many-to-one matches (compression warps) and indels between their time points in sequential order, denoted by the five states: M, V, W, I, D, respectively (Fig. 1b). While matches imply similarity between transcriptomic states of $R$ and $Q$, indels (also called gaps) imply mismatches (differential/unobserved transcriptomic states compared to each other). A standard DP alignment algorithm optimizes the mapping between two sequences by constructing a pairwise cost matrix and generating the path that minimizes the total cost (Fig. 1c). This uses a scoring scheme to quantify correspondences between every pair of $R$ and $Q$ time points.

Unlike DTW and biological sequence alignment, G2G implements a DP algorithm that handles both matches (including warps) and mismatches jointly, querying each gene. This extends Gotoh's three-state algorithm[16] (defining time-efficient DP recurrences with affine gap scheme[22–24] over M, I, D states) to accommodate V, W warp states

(Fig. 1b), allowing a nonlinear mapping between the pseudotime axes of $R$ and $Q$. Figure 1d exemplifies a gene alignment generated by G2G, described as a five-state string defining matches and mismatches of $R$ and $Q$ time points in sequential order (left to right), similar to how a DNA–protein pairwise alignment is reported.

Our DP scoring scheme incorporates a cost function based on minimum message length (MML) inference[25–27] (top left of Fig. 2 and Supplementary Fig. 1) and the state transition probabilities from a five-state machine (middle left of Fig. 2). The MML criterion allows computing a symmetric cost (named MML distance) for matching any two $R_j$ and $Q_i$ time points based on their gene expression distributions. This accounts for their differences in both mean and variance, acknowledging that either trajectory may be noisier. The five-state machine defines a symmetric cost of assigning an alignment state for $R_j$ and $Q_i$. This machine has been empirically fine-tuned on a simulated dataset. Each cost term is computed as the Shannon information[28] $I$ measured in 'nits' under the probability model of the corresponding events $E$, that is, $I(E) = -\log(\Pr(E))$ nits (Methods).

### Overview of the G2G framework

G2G is composed of several components, which include input preprocessing, DP alignment, alignment clustering and downstream analysis (Fig. 2).

G2G's inputs are log1p-normalized (per-cell total raw transcript count normalized to a constant over all genes and transformed to log(normalized count + 1)) scRNA-seq matrices of the reference and query systems, and their pseudotime estimates. G2G first interpolates each gene expression trajectory. This initially transforms the pseudotime axis to the [0,1] range using min–max normalization, over which we take a predefined number of equispaced interpolation time points, similar to CellAlign[7]. For each interpolation time point, we estimate gene expression as a Gaussian distribution, considering all cells kernel-weighted[7] by their pseudotime distance to this interpolation time point.

The interpolated gene trajectories of the reference and query are aligned using our DP algorithm, generating optimal gene alignments described as five-state strings (Fig. 1d and top right matrix of Fig. 2). The five-state string of a gene informs the percentage of match calling (M, V, W), termed 'alignment similarity'. (Note that under symmetric costs, the alignment string is symmetric regardless of which dataset is the reference, only swapping between symmetric states I-D, W-V). The pairwise Levenshtein distance matrix between these strings can be used to reveal the diversity of gene alignments (for example 100% mismatched, 100% matched, 30% early-matched and late-mismatched), by running agglomerative hierarchical clustering (where an optimal grouping is determined by inspecting the mean silhouette coefficients under different distance thresholds of the linkage criterion). G2G generates a representative alignment for a cluster by aggregating its gene-level alignments (for example cluster of 100% matches represented by a string over M, V, W; cluster of 100%

---

**Fig. 3 | Genes2Genes outperforms the current state-of-the-art of trajectory alignment. a**, Differences in the algorithms and outputs of CellAlign[7], TrAGEDy[17] and G2G. CellAlign runs DTW, defining the state space $\Omega = $ [M, W, V] (Fig. 1b). TrAGEDy performs DTW post hoc processing, while G2G unifies DTW and gap modeling. Both of them define the state space $\Omega = $ [M, W, V, I, D] (Fig. 1b). **b**, Comparing features across CellAlign, TrAGEDy and G2G. **c**, A Gaussian process-based simulator is used to generate 3,500 simulated pairs of reference and query gene trajectories for benchmarking G2G against CellAlign and TrAGEDy, testing under three main classes of alignment patterns: matching, divergence and convergence. divergence and convergence are subcategorized based on their approximate time of bifurcation (early, mid and late), resulting in seven total patterns (each with 500 alignments). **d**, The three-state, cell-level alignment generated by CellAlign for each pattern (under 15 equispaced time points). **e**, The five-state, cell-level alignments generated by both modes of TrAGEDy (referred to as TrAGEDy$_{\text{MINIMUM}}$ and TrAGEDy$_{\text{NULL}}$) and G2G. **f**, Percentages of accurate alignments by TrAGEDy and G2G across all patterns (left). Clustergram

of the pairwise Levenshtein distance matrix across all G2G alignments, separating the distinct patterns using agglomerative hierarchical clustering (right). **g**, Comparing hierarchical clustering of the gene alignments generated by CellAlign, TrAGEDy and G2G; x axis is the number of clusters (representing varying clustering resolutions) in log scale; y axis is the mis-clustering rate (outlier percentage across all clusters). **h**, Cell-level alignment of two simulated trajectories with no shared process, with three example gene alignments generated by TrAGEDy and G2G. Five-state alignment strings from each method (left) and expression plots (right) of the three example genes. Column 1 shows interpolated gene expression (y axis) against pseudotime (x axis). The bold lines represent mean expression trends, while the faded data points are 50 random samples from the estimated expression distribution at each time point as generated under G2G. Columns 2–3 show the actual log1p-normalized expression (y axis) against pseudotime (x axis). Each point represents a cell. Illustrations in **a**–**c** were created using BioRender (https://biorender.com). All interpolations and alignment statistics were generated using our G2G framework.

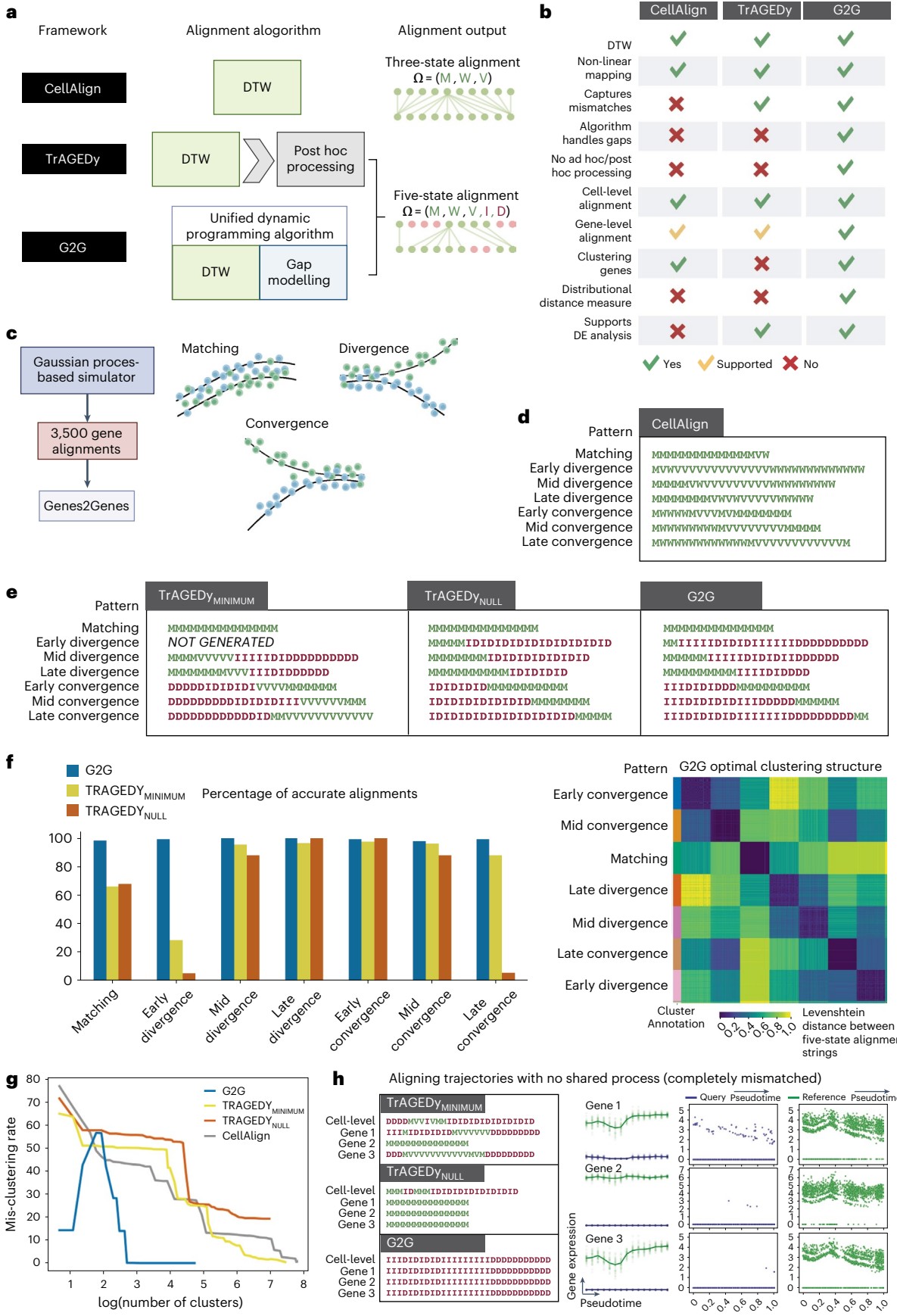

mismatches represented by a string over I, D;). G2G finally aggregates all gene-level alignments into a single, cell-level alignment, informing an average mapping between the trajectories. Both gene-level and cell-level alignments are useful when the alignment patterns are heterogeneous across genes. Altogether, these enable downstream analysis (for example gene set over-representation analysis).

**G2G expands the capacity of DTW.** G2G infers statistically-consistent matches and mismatches between reference and query time points. Such output is impossible from DTW (for example CellAlign[7]) as it maps all time points including those with transcriptomic differences (Fig. 3a). One could perform local DTW with user-defined thresholds[7] or post hoc processing of DTW alignment (as in TrAGEDy[17]) to unmap dissimilar time points, yet the underlying assumption of a definite match remains. This is particularly problematic for datasets with no shared process[17]. In contrast, G2G systematically disconnects mismatching time points without thresholding or post-processing (see Fig. 3a,b and Supplementary Table 1 for summarized comparison of the features that fundamentally distinguish G2G from CellAlign[7] and TrAGEDy[17]).

**G2G captures different alignment patterns in simulated data**
To benchmark G2G against CellAlign[7] and TrAGEDy[17], we experimented on (1) a dataset with seven alignment patterns; (2) a real dataset with artificial perturbations; and (3) a negative control dataset. CellAlign[7] and TrAGEDy[17] alignments were converted into five-state strings before comparison. TrAGEDy[17] prunes DTW matches based on a 'minimum dissimilarity score' (hereafter, 'TrAGEDy$_{MINIMUM}$'), with an alternative to disregard the minimum (hereafter, 'TrAGEDy$_{NULL}$').

**Experiment 1.** We simulated 3,500 pairs of matching, divergence and convergence trajectories with seven distinct patterns (Fig. 3c and Extended Data Fig. 1a–c; Methods). Each trajectory comprises 300 cells spread across pseudotime range [0,1]. Divergence and convergence groups represent early, mid and late bifurcation (approximately at time points $t_b \in$ (0.25, 0.5 and 0.75), respectively). We examined alignments for each pattern under 15 interpolation time points. The expected alignments were: 100% match for matching; matched region (start-match) + mismatched region (end-mismatch) for divergence; and mismatched region (start-mismatch) + matched region (end-match) for convergence. The match/mismatch lengths for divergence/convergence depend on $t_b$ (Extended Data Fig. 1d,e). We used the accuracy rate (proportion of correct alignments) to fine-tune the five-state machine parameters set by G2G as default (Methods and Supplementary Table 2).

Figure 3d,e reports cell-level alignments from all methods. Both G2G and TrAGEDy correctly described the seven patterns (Fig. 3e). In contrast, CellAlign[7] could not describe divergence and convergence

(Fig. 3d). Across all patterns, G2G outperformed TrAGEDy in gene-level alignment (Fig. 3f left) with higher accuracy rates of 98.6%, 99.4%, 99.8%, 100%, 99.2%, 98.2% and 99.2%, for matching, divergence (early, mid and late) and convergence (early, mid and late) pairs, respectively. All distributions of match/mismatch lengths in divergence/convergence alignments fell within the expected ranges (Extended Data Fig. 2a,b). TrAGEDy$_{MINIMUM}$ gave 66.26%, 28.57%, 95.87%, 96.86%, 97.35, 96.15 and 88.2% accuracy rates, respectively, with divergence/convergence alignments showing higher variability in their match/mismatch lengths, thus falling beyond the expected ranges. TrAGEDy$_{NULL}$ gave 68.2%, 5.4%, 88%, 100%, 100%, 88% and 5.8% accuracy rates, respectively, with better length distributions than TrAGEDy$_{MINIMUM}$. G2G showed fewer false mismatches on average for matching alignments compared to TrAGEDy, while also having fewer intermediate false mismatches compared to TrAGEDy$_{MINIMUM}$. Notably, TrAGEDy$_{NULL}$ generated no intermediate false mismatches, yet yielded higher inaccuracy due to 100% matched or expected-order-swapped alignments.

G2G clustering separated the patterns very well (Fig. 3f, right); hierarchical clustering of alignments at the optimally chosen 0.22 distance threshold gave 15 clusters, with only a 0.1% mis-clustering rate. (Extended Data Fig. 2c; see Methods for details on optimal threshold selection). We compared this to CellAlign's[7] k-means clustering of genes based on their pseudotime shifts (differences between matched time points in gene-level DTW alignments). All mis-clustering rates were substantially higher (falling within the range of 42.6% and 60.4%) for k ∈ [7, 50] (Extended Data Fig. 2d) than G2G's mis-clustering rate. CellAlign[7] and TrAGEDy displayed higher noise and mis-clustering rates compared to G2G (Fig. 3g).

**Experiment 2.** To test G2G's match detection in scRNA-seq data, we used a murine pancreatic development dataset[29] subsetted to β-cell lineage (1,845 cells), considering 769 lineage-driver genes. We randomly split cells into reference and query, and simulated mismatches as a deleted portion (perturbation scenario 1) or changed portion (perturbation scenario 2) of increasing size at the beginning of the trajectory (Extended Data Fig. 3a). We then performed gene-level alignments using G2G and TrAGEDy (under 50 interpolation time points) for each scenario and calculated their alignment similarities (Extended Data Fig. 3b,c). For perturbation scenario 1, both G2G and TrAGEDy alignment similarity decreased with increasing deletion sizes as expected across smaller perturbation sizes, although the detected mismatch length was shorter than expected for deletions larger than 20%. This is due to the relatively nonvarying gene expression between pseudotime bin 10–20 (Extended Data Fig. 3d), which caused warps instead of mismatches. Both methods were consistent in capturing this behavior. For perturbation scenario 2, alignment similarity had an expected maximum and minimum (Extended Data Fig. 3e). Generally, both

**Fig. 4 | Genes2Genes captures matches and mismatches at gene-level resolution. a**, G2G alignment on a published time-course dataset[7,20] of murine bone-marrow-derived dendritic cells stimulated with PAM (reference) or LPS (query). **b**, Aggregate alignment over the alignments of 99 'core antiviral' genes (top). Stacked barplots represent reference and query cell compositions across 14 equispaced pseudotime points, colored by post-stimulation sampling time; boxed segments represent mismatches; black lines represent matches. Pairwise time point matrix between reference and query (bottom). Color represents total gene count showing a match between corresponding time points. White line represents the average alignment path. **c**, Gene expression of three representative core antiviral genes (*IRF7*, *STAT2* and *IFIT1*) in query (blue) and reference (green). Interpolated log1p-normalized (per-cell total raw transcript counts normalized to 10,000 and log1p-transformed) expression (*y* axis) against pseudotime (*x* axis) (left). Bold lines represent mean expression trends and faded data points indicate 50 random samples from the estimated expression distribution at each time point. Black dashed lines represent time point matches (captured by the alignment string below). Actual log1p-normalized expression

(*y* axis) against pseudotime (*x* axis) (right). Each point represents a cell. Red circles highlight early cells ('precocious expressers') with high expression. **d**, Same plots as **b** for 89 'peaked inflammatory' genes, clustered following their alignments (Extended Data Fig. 2). Dashed, colored lines represent example cluster-specific alignment paths. **e**, Same plots as **c** for representative genes (*CXCL2*, *PLK2*, *CXCL1* and *CD44*) from each cluster shown in **d**. **f**, Alignment similarity (*y* axis) against log$_2$ fold change of mean expression (*x* axis) for peaked inflammatory genes (middle). Color represents alignment similarity. Surrounding plots show interpolated log1p-normalized expression (*y* axis) against pseudotime (*x* axis) on the left and the gene expression violin plot on the right, for four selected genes (*SGMS2*, *CCRL2*, *TNF* and *C5AR1*). Green and blue violin plots include *n* = 179 PAM-stimulated and *n* = 290 LPS-stimulated cells, respectively. Violin shows expression distribution across cells as a kernel density estimation. The box inside each violin shows the interquartile range (25–75% quantiles, with a point indicating median). The illustration in **a** was created using BioRender (https://biorender.com). All interpolations and statistics were generated using our G2G framework.

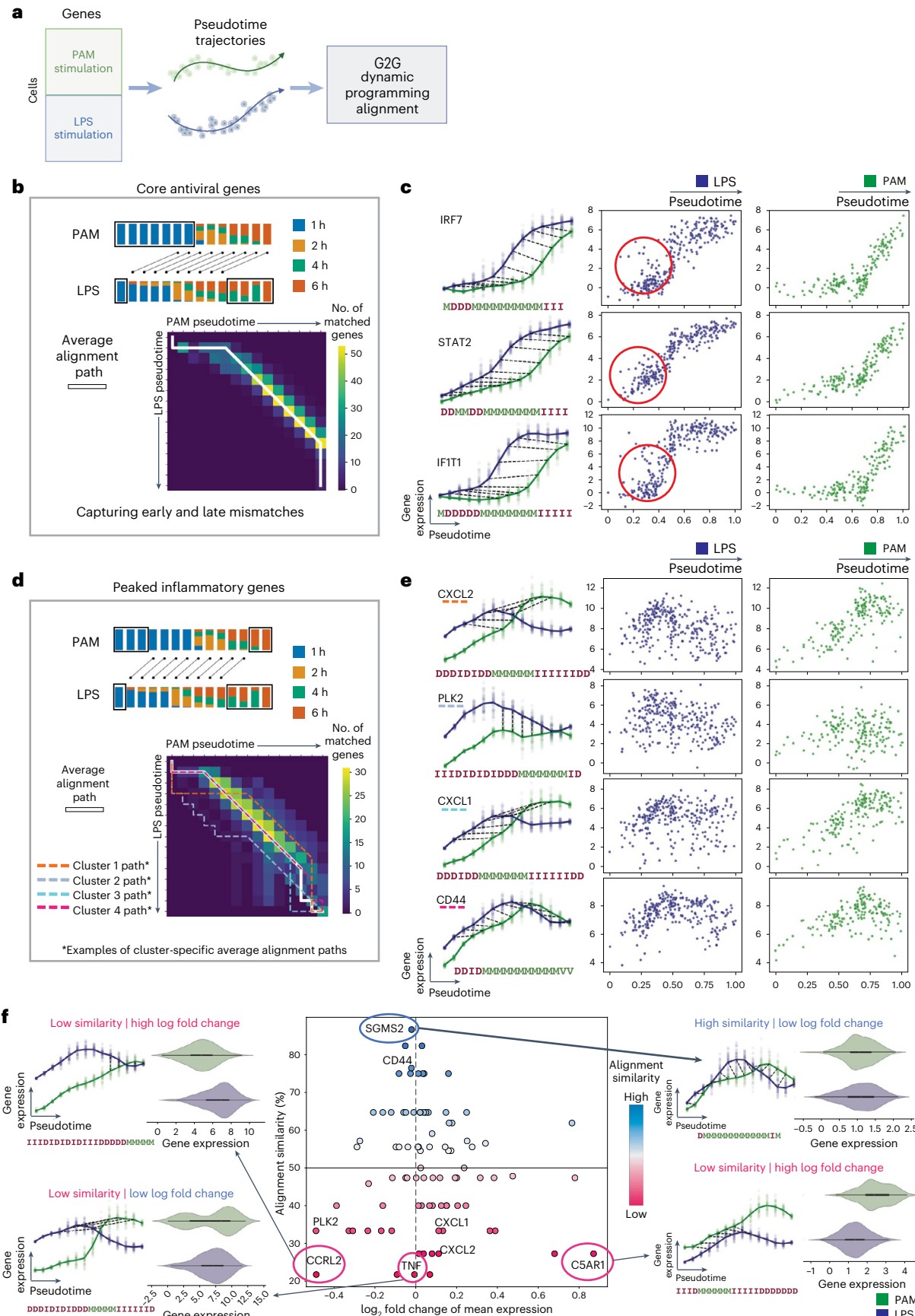

methods showed expected trends, falling within the expected ranges for larger perturbation sizes. Notably, TrAGEDy$_{NULL}$ outperformed TrAGEDy$_{MINIMUM}$ in both scenarios. TrAGEDy$_{NULL}$ also showed higher accuracy for perturbation sizes <6%. Overall, G2G and TrAGEDy$_{NULL}$ closely performed with better match detection than TrAGEDy$_{MINIMUM}$; however, G2G showed relatively less variability in results overall.

**Experiment 3.** Examining two simulated datasets with no shared process (referred to as negative control, tested by TrAGEDy[17]), G2G generated an aggregate alignment of 100% mismatch as expected, whereas TrAGEDy[17] falsely inferred match segments (Fig. 3h); similar results were observed for three genes with completely mismatched trajectories.

In conclusion, G2G outperformed existing methods by accurately aligning and clustering genes with different alignment patterns.

## G2G captures matches and mismatches at gene-level resolution

To further demonstrate our framework's features, we performed G2G alignment on the time-course dataset[20] tested by CellAlign[7] (Fig. 4a). This involved murine bone-marrow-derived dendritic cells treated with PAM3CSK (PAM) or lipopolysaccharide (LPS) to simulate responses to different pathogens.

G2G's ability to capture mismatches is revealed when aligning genes from the 'core antiviral module' (Extended Data Fig. 4a). CellAlign[7] demonstrated a 'lag' in gene expression after PAM stimulation compared to LPS[7], which was also captured by G2G aggregate alignment (Fig. 4b). In addition, G2G identified mismatches in the early and late pseudotime points. Clustering alignments revealed low diversity, implying that all genes generally follow the average pattern (Extended Data Fig. 4b). At early pseudotime points, the gene expression was consistently low in the PAM condition, whereas some LPS-stimulated cells were already showing elevated expression (for example *IRF7*, *STAT2* and *IF1T1*; Fig. 4c). These have also been noticed and described as 'precocious expressers' in the original paper[20]. The mismatch in late LPS pseudotime points was caused by the peaked expression, whereas the expression of PAM-stimulated cells was still on the rise, not yet reaching a peak.

For genes in the 'peaked inflammatory module', Fig. 4d shows their G2G aggregate alignment. Clustering of genes revealed cluster-specific average alignments that differed from the main average alignment (Fig. 4d and Extended Data Fig. 4c–e). Representative genes from different clusters (Fig. 4e) displayed subtle differences in the length and position of matches. Using G2G alignment similarity statistics (Fig. 4f), we identified *SGMS2* as the most similar gene (with low log fold change) and *CCRL2* and *C5AR1* as highly dissimilar genes (with high log fold change) between PAM- and LPS-stimulated trajectories. *CCRL2* alignment showed a late convergence. We also note *TNF* as highly dissimilar

despite its negligible log fold change, undetectable by a standard DE test (for example Wilcoxon rank-sum $P = 0.2$), hence highlighting the importance of trajectory alignment.

The above results again showcase how G2G captures mismatched regions between scRNA-seq trajectories.

## G2G finds early/late differences in disease epithelial cells

Next, we compared two cell differentiation trajectories from healthy lung versus diseased lung in idiopathic pulmonary fibrosis (IPF). IPF is an incurable and irreversible disease characterized by deposition of extracellular matrix by myofibroblasts, scarring and progressive loss of lung function, with an estimated survival rate of 3–5 years after diagnosis[30]. Using the Adams et al. (2020) dataset[21], we investigated the differentiation of alveolar type 2 (AT2) cells into alveolar type 1 (AT1) cells in the healthy lung versus AT2 differentiation into aberrant basaloid cells (ABCs) in the IPF lung[31,32] (Fig. 5a). ABCs have only recently been characterized in single-cell studies of patients with IPF[21,31,33,34]; Their origin and role in IPF pathogenesis is still unclear.

We inferred trajectories for healthy and IPF data using diffusion pseudotime[35] (Supplementary Fig. 2) and aligned them using G2G across 994 highly variable genes (under 13 interpolation time points). The alignment distribution (Extended Data Fig. 5a) shows ~62% mean similarity. As expected, their aggregate alignment showed mismatches only at late pseudotime points (Fig. 5b), given that both healthy and IPF lung epithelial differentiation start from AT2 cells, but give rise to AT1 in healthy versus ABCs in IPF. Moreover, examining the ABC-specific marker genes (Fig. 5c and Supplementary Fig. 3), we observe a diverging pattern as reported by other studies[21,31].

We performed gene set over-representation analysis on the top mismatched genes (alignment similarity ≤40%) and found that epithelial mesenchymal transition (EMT) was the most significantly enriched pathway (Fig. 5d and Supplementary Table 3). While most EMT genes show mismatches only at later stages, consistent with dysregulated EMT being implicated in ABC development in IPF[21,31–34], some EMT genes already show differences at early/mid differentiation stages (for example *NNMT*, *CXCL1* and *CXCL8*). These could be potential therapeutic targets to prevent differentiation into the pathological ABC state.

Downstream clustering revealed additional alignment patterns (Extended Data Fig. 5b,c). For example, cluster 3 represents almost-completely mismatched genes, including upregulation of *CAMK1D* (Fig. 5e), a known target of TGF-β1 (ref. 36), a key regulator of IPF development[37]. Overall, G2G captured the expected alignments and some new early/mid mismatches between the healthy and IPF trajectories.

## G2G reveals differences of T cell development in vitro

We next employed G2G to compare in vitro and in vivo human T cell development. The thymus is the key site for T cell development, where

---

**Fig. 5 | Genes2Genes compares cell differentiation trajectories between healthy lung and disease lung in idiopathic pulmonary fibrosis.** **a**, Schematic of the healthy and IPF cell differentiation trajectories of focus, that is, differentiation of alveolar type 2 (AT2) cells into alveolar type 1 (AT1) cells in the healthy lung (reference) versus ABCs in the IPF lung (query). **b**, Aggregate alignment over the alignments of all highly variable genes (HVGs) (top). Stacked barplots represent reference and query cell-type compositions across 13 equispaced pseudotime points; boxed segments represent mismatches; black lines represent matches. The pairwise time point matrix between healthy and IPF pseudotime (bottom). Color represents total gene count showing a match between corresponding healthy and IPF time points. White line represents the average alignment path. **c**, Aggregate alignment over the alignments of 88 ABC marker genes (Supplementary Fig. 3) plotted as in **b**, with the aggregate alignment schematic on top, and the pairwise time point matrix in the middle. Gene expression plots for three example ABC marker genes (*KRT17*, *MMP7* and *FN1*) between IPF (blue) and healthy (green) data along pseudotime, plotting

interpolated log1p-normalized (per-cell total raw transcript counts normalized to 10,000 and log1p-transformed) expression (*y* axis) against pseudotime (*x* axis) (bottom). Bold lines represent mean expression trends; faded data points are 50 random samples from the estimated expression distribution at each time point. Black dashed lines represent matches between time points. **d**, Aggregate alignment path (white) for all EMT pathway genes, plotted on the pairwise time point matrix between healthy and IPF as in **b**, with the schematic on the right (top right). Heatmap of the smoothened (interpolated) and z-normalized mean log1p gene expression of genes in the EMT pathway along pseudotime (bottom right). **e**, Gene expression of *CAMK1D* between IPF (blue) and healthy (green) along pseudotime. Interpolated log1p-normalized expression (*y* axis) against pseudotime (*x* axis) as in **c** (top). Actual log1p-normalized gene expression versus pseudotime plots (bottom). The illustration in **a** was created using BioRender (https://biorender.com). All interpolations and statistics were generated using our G2G framework.

---

lymphoid progenitors differentiate through stages of double-negative (DN) and double-positive T cells to acquire T cell receptors (TCRs) (Fig. 6a top and Extended Data Fig. 6). If the TCR recognizes self-antigen presented on the major histocompatibility complex during the

process of positive selection, the developing T cells further differentiate through abT(entry) cells and finally mature into single-positive (SP) T cells. There are different subsets of SP T cells, including CD4⁺ T, CD8⁺ T and regulatory T (T_reg) cells, as well as the newly recognized

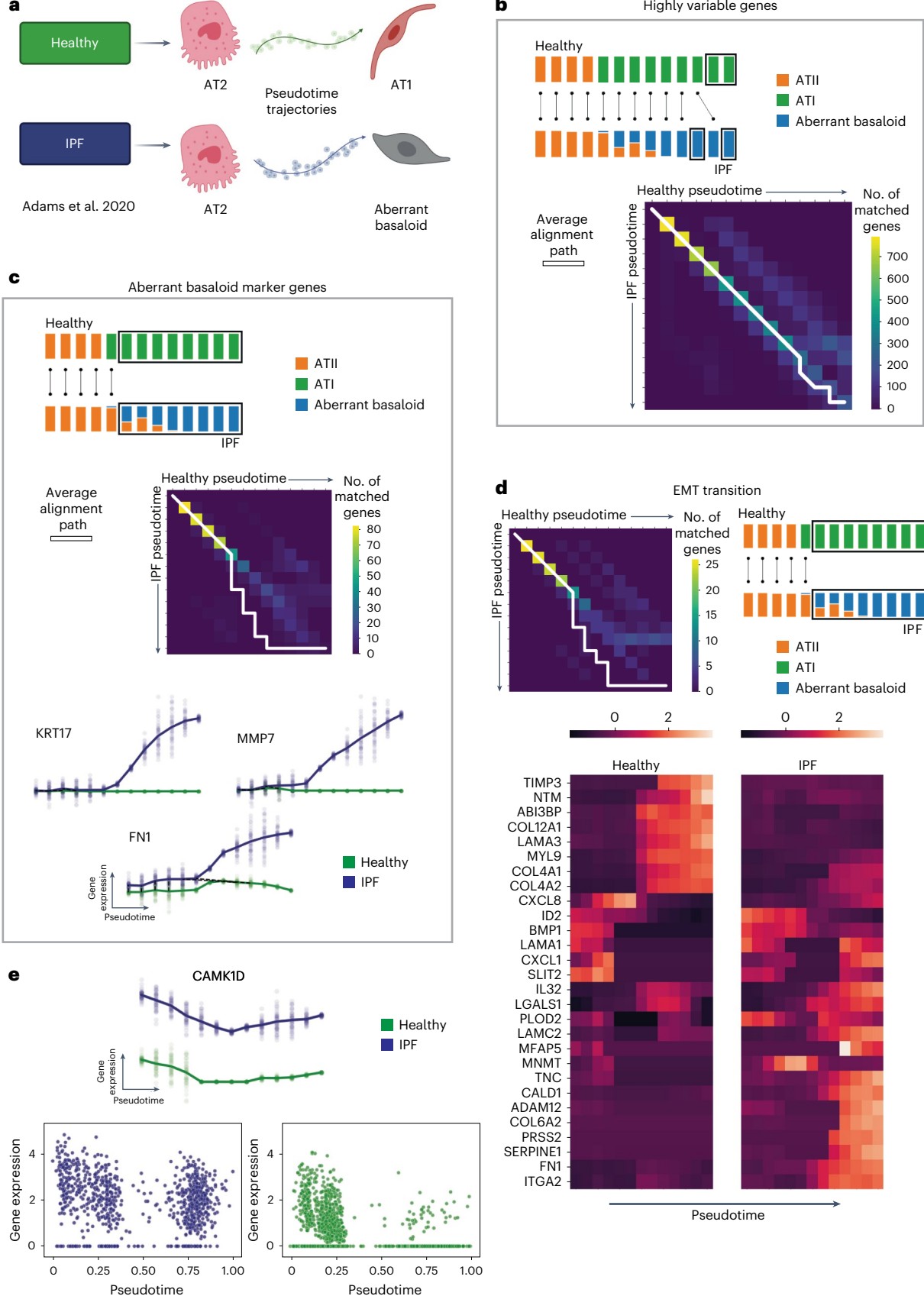

unconventional type 1 and type 3 innate and CD8AA T cells[38,39]. To investigate human T cell development in a model system in vitro, we differentiated induced pluripotent stem (iPS) cells into mature T cells using artificial thymic organoids (ATOs)[40]. We previously collected differentiated cells from week 3, 5 and 7, and reported that the mature T cells in ATOs were most similar to in vivo type 1 innate T cells[39]. To further explore, we performed scRNA-seq analysis of cells collected at regular intervals throughout differentiation, including the early time points (Fig. 6a bottom and Extended Data Fig. 6a). Cell types were annotated using CellTypist[41] and marker gene analysis (Extended Data Figs. 6b–8). The ATOs capture differentiation from stem cells, through mesodermal progenitors and endothelium to the hematopoietic lineage and then further to the T cell lineage.

We integrated the ATO cells with the relevant in vivo cells from our developing human immune atlas[39] (hereafter, 'pan fetal reference') into a common latent embedding using scVI[42], and estimated their pseudotime (Extended Data Fig. 6c–e). The ATO pseudotime was estimated using a Gaussian process latent variable model (GPLVM)[43] with sampling times as priors. The pan fetal reference cells' pseudotime was computed similarly by estimating their time priors from the nearby ATO cells.

G2G alignment between the ATO and in vivo trajectories was performed (under 14 interpolation time points) using all transcription factor (TF) genes[44] (1,371 TFs), as many TFs function as 'master regulators' of cell states and have been used to induce cell differentiation. Their aggregate alignment showed mismatches at the beginning and at the end (Fig. 6b), with ~66% mean alignment similarity in their distribution (Extended Data Fig. 9a). Independently, TrAGEDy[17] high-dimensional alignment also verified this strong mismatch in early and late stages between in vitro and in vivo T cell differentiation.

**Clustering alignments finds interesting groups of genes.** TF alignments were hierarchically clustered and explored at several resolutions (Extended Data Fig. 9b,c). At low resolution (Extended Data Fig. 9c), cluster 2 includes pluripotent TFs showing insertions at early pseudotime (Supplementary Table 5). Well-known stemness TFs *POU5F1*, *NANOG* and *TBX3* (ref. 45) were present in early ATO development, but missing from the reference. This is expected for pluripotent stem cell TFs (Fig. 6c), as in vitro differentiation started from iPS cells, whereas the earliest in vivo cells were hematopoietic stem cells (HSCs). Among them, *HHEX*[46–48] demonstrated another pattern: a match between in vivo and in vitro HSCs and DN T cells as expected, although with lower maximum *HHEX* expression in in vitro versus in vivo cells (Fig. 6c). Notably, clustering also revealed TF mismatches only at the middle time points (for example *POU6F1*, *SOX18* and *CSRNP3* in cluster 0 at low resolution and *BATF2* in cluster 13 at high resolution). This might represent a missing cell state, for example, *BATF2* is expressed sparsely in endothelial cells, which are present only in the in vitro system. On the other hand, *LEF1* (necessary for early stages of thymocyte maturation[49])

stands out as a single cluster showing almost 100% matching between the trajectories, whereas two other clusters include almost 100% mismatching TFs, for example, *GATA6*, *SALL4*, *HOXB6*, *NACC2* and *PRDM6*. See Supplementary Fig. 4 for expression and alignment plots of all aforementioned genes.

**TNF as a potential target for in vitro optimization.** Gene set overrepresentation among the most mismatched genes (alignment similarity ≤40%; Supplementary Table 4) revealed genes associated with TNF signaling via the nuclear factor (NF)-κB pathway. Many of the TFs in this pathway (for example *FOSB*, *JUNB* and *NR4A2*) showed an increasing trend at the last stage of in vivo T cell development, whereas this increase is missing in the in vitro T cells (Fig. 6d). We further validated this by showing that these genes have higher expression in the thymic medulla (where mature T cells reside) than in the cortex (where T cell progenitors reside), to ensure that this is not due to handling artifacts of tissue digestion (Supplementary Note). There are exceptions to this overall pattern, for example, *KLF2*, whose expression is higher in vitro than in vivo (Fig. 6d), possibly due to each gene being regulated by more than one signaling pathway. Alignment of all 196 genes in the TNF pathway also confirms a significant mismatch in the last stage (Extended Data Fig. 10a), suggesting this pathway as a potential target for further in vitro optimization. Restricting the analysis to T cell lineages, DN stage onwards (Extended Data Fig. 10b, left), TNF signaling via NF-κB pathway remained the most enriched gene set among the mismatched TFs (Supplementary Table 6). We remark that although it is possible to recover these differences via direct DE analysis between cell subsets, for example, end products of ATOs versus in vivo T cells, a key advantage of trajectory alignment is the ability to systematically identify the time point where the mismatch occurred during differentiation. This in turn informs us when to introduce TNF in in vitro optimizations.

**In vitro SP T versus in vivo CD8⁺ T lineages.** The above alignments were between in vivo type 1 innate T cells and the relevant precursors, as we previously found that the in vitro mature T cells were closest to the in vivo type 1 innate T cells[39]; however, in vitro cell differentiation to conventional CD8⁺ T cells might also provide promising routes for cell therapies. We therefore performed another G2G alignment using in vivo conventional CD8⁺ T cells and the relevant T lineage precursors (DN T cells onwards). Differences in the two alignment results suggest that potential targets such as *SOX4*, *FOXP1* and *ARID5B* may tune cells toward in vivo CD8⁺ T cells (Supplementary Note, Extended Data Fig. 10b,c and Supplementary Table 7).

**Preliminary experiment targeting TNF signaling.** G2G alignments revealed potential targets for further optimization of in vitro T cell differentiation (Fig. 6e). We experimentally validated the impact of TNF signaling by adding TNF into the ATO medium between weeks 6–7 (Fig. 6f), and comparing SP T cells of our ATOs (wild-type) and the

---

**Fig. 6 | Genes2Genes aligns in vivo, in vitro human T cell development.**
**a**, Schematic illustration of T cell development in the human thymus. **b**, Aggregate alignment over the alignments of 1,371 TFs between in vitro organoid (ATOs) and in vivo[39] human T cell developmental trajectories, shown in the pairwise time point matrix between organoid and reference. Color represents total gene count showing a match between corresponding time points. White lines represent the average alignment path. Stacked barplots represent reference (top) and query (left) cell-type compositions across 14 equispaced pseudotime points. **c**, Aggregate alignment over all TFs in the pluripotency signaling pathway, plotted on the pairwise time point matrix (top left) as in **b**; schematic of this mapping between reference and organoid cell-type compositions across pseudotime; boxed segment represents the mismatched ATO pluripotency stage; black lines represent matches. Interpolated log1p-normalized (per-cell total raw transcript counts normalized to 10,000 and log1p-transformed) expression (*y* axis) against pseudotime (*x* axis) for selected genes (bottom left). Heatmap of

the smoothened (interpolated) and z-normalized mean gene expression along pseudotime (bottom right). **d**, Same plots as **c** for all TFs in the TNF signaling via NF-κB pathway. The boxed segment in the right-top plot represents the mismatched last stage in vivo T cell maturation. **e**, Schematic illustration of potential targets for further optimization of in vitro T cell differentiation toward either type 1 innate T cells or conventional CD8⁺ T cells. **f**, Schematic illustration of the comparison between SP T cells from the wild-type ATOs and the TNF-treated ATOs against in vivo type 1 innate T cells. SP T cells from ATOs after TNF treatment show more maturity toward in vivo type 1 innate T cells. **g**, Heatmap of mean log1p-normalized gene expression of TFs within the TNF signaling via NF-κB pathway (same gene list as in **d**) in reference (in vivo type 1 innate T cells), SP T cells from wild-type ATOs and SP T cells from TNF-treated ATOs (ATO_TNF). Illustrations in **a**,**e**,**f** were created using BioRender (https://biorender.com). All interpolations and statistics were generated using our G2G framework.

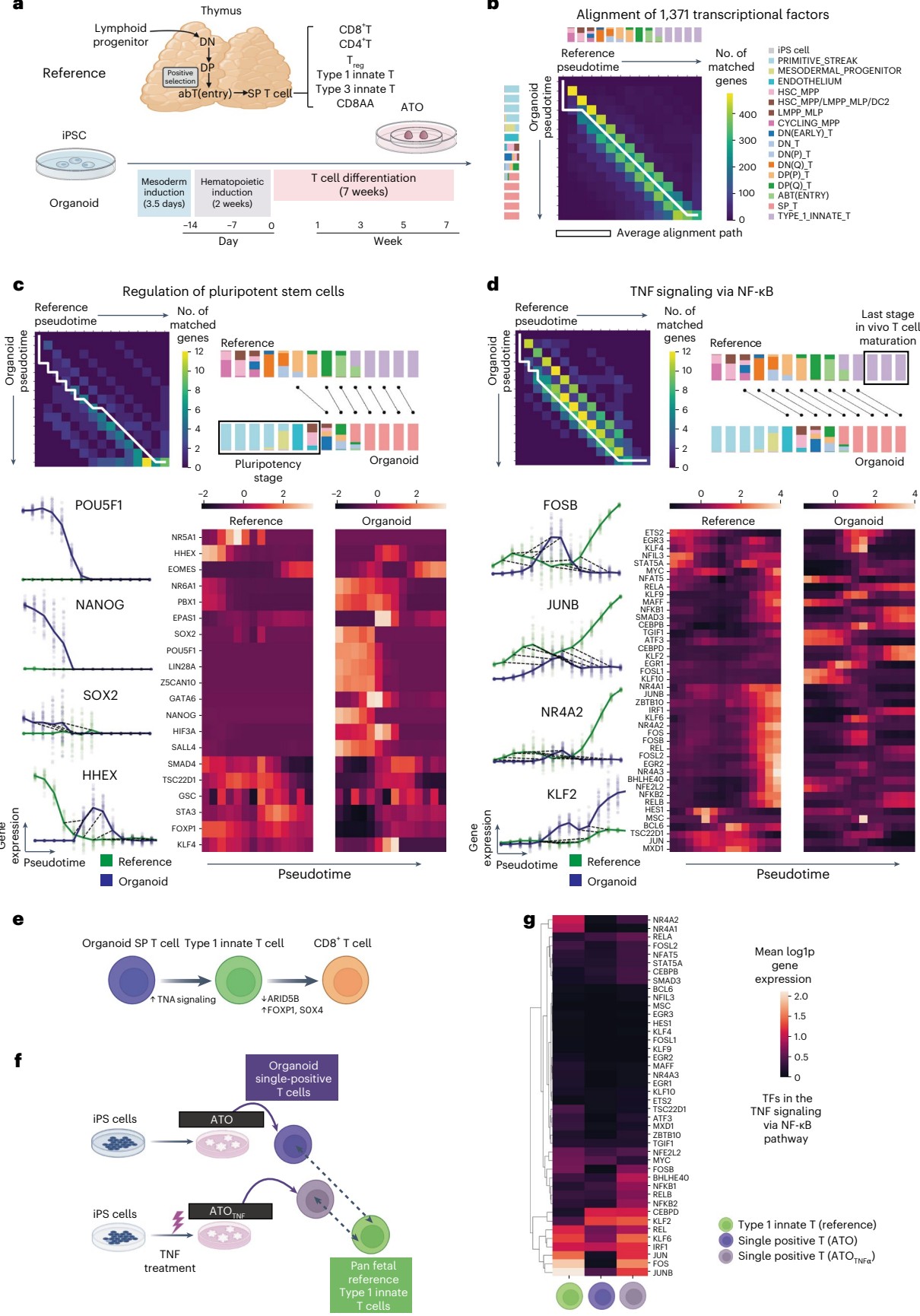

TNF-treated ATOs (ATO$_{TNF}$) to the type 1 innate T cells in the pan fetal reference. We observed that, in the scVI[42] latent space of all in vivo and in vitro cells, the Euclidean distance between the mean vectors of in vitro SP T cells and the in vivo type 1 innate T cells decreased by ~5% after TNF treatment. Further examining the effect, TFs and all genes within the TNF signaling pathway (Fig. 6g and Extended Data Fig. 10d) in T cells showed higher expression in ATO$_{TNF}$ compared to ATOs, as expected. The mean distance of gene expression distributions also dropped for all genes and TFs that were significantly distant between ATO T cells and in vivo type 1 innate T cells (Methods and Supplementary Note). We note the distance change in several known SP T cell maturation markers (*IL7R*, *KLF2*, *FOXO1*, *S1PR1* and *SELL*)[50,51] (Extended Data Fig. 10e,f and Supplementary Fig. 5). *IL7R* has shown to initiate in mature SP thymocytes, with its expression dependent on NF-κB signaling[52–55]. The distance of in vitro and in vivo *IL7R* expression significantly dropped after TNF treatment with an increased expression as expected from mature T cells. *KLF2* was also further upregulated. The rest of the markers maintained expression. Overall, these results suggest more mature SP T cells in ATO$_{TNF}$.

It is worth noting that another in vitro T cell differentiation protocol added TNF throughout differentiation to improve T cell production efficiency[56,57]; however, G2G identified that the TNF pathway mismatches at late T cell differentiation. Therefore, targeting this pathway, we added TNF in the last week of differentiation to improve T cell maturation, which enabled us to successfully push in vitro T cells to better match the in vivo T cells (Fig. 6f). Our results suggest that this is a potential direction to refine the ATO protocol toward mature type 1 innate T cells, subject to future functional validation studies.

## Discussion

Trajectory alignment can capture transcriptomic similarities and differences of temporal dynamics between cell populations. We developed G2G, a framework to align single-cell pseudotime trajectories at single-gene resolution, and demonstrated its utility and versatility in single-cell studies (for instance, discerning differential genes or pathways that drive pathogenicity in diseases or potential targets for refining organoid protocols toward better in vivo recapitulation). G2G outperforms existing methods through more descriptive and accurate alignments, and our work provides proof of concept of the power of gene-level alignment.

Given cell-by-gene matrices and pseudotime estimates of a reference and query, G2G generates an alignment for each gene by unifying DTW and gap modeling. The distribution of alignments can inform gene clusters with broadly similar alignment patterns and their average alignments. As such aggregated results depend on the genes we choose to align, we recommend selecting genes that are as informative as possible (for example, lineage-relevant driver genes or regulons). For instance, we can align the significantly upregulated and downregulated genes in the reference to investigate whether the query follows the same dynamics. Aligning TFs can inform differential regulation. When aligning all or highly variable genes, we can inspect gene clusters (paired with over-representation analysis) to extract biologically meaningful groups, for example, revealing biological or signaling pathways that drive mismatches at different times. These can be a basis for protocol intervention when comparing in vivo or in vitro trajectories and for mechanistic molecular interpretation of differences between any trajectories.

An important feature of G2G is gene-specific alignment. Most existing approaches produce a single alignment by computing high-dimensional Euclidean distances over all genes. Such metrics suffer from 'the curse of dimensionality' by losing accuracy as the number of genes increases[58]. A single alignment also masks gene alignment heterogeneity. Alpert et al.[7] recommend aligning the largest gene set with significant DE over time, to avoid noise from stably expressed genes. Our method goes further and fully resolves all gene groups with

individual matching and mismatching patterns at different stages of time.

The reliability of trajectory alignment depends on the quality of inputs. We recommend selecting a pseudotime inference method[2] suitable for the datasets at hand[2]. One could run G2G with estimates from different methods and evaluate how robust the results are. Future work is also needed to calibrate trajectory input. For instance, an adaptive Gaussian kernel interpolation may optimize the method's sensitivity to the variance of expression in nearby cells. We also recommend inspecting whether the cell density along pseudotime represents the entire dynamic process. When there are missing (unobserved) cells representing sudden changes, the assumption of a smooth trajectory breaks and limits G2G from generating accurate alignments, as the data estimation at each interpolation point is controlled by the observed cells in its neighborhood. Furthermore, G2G only compares two linear trajectories. We are aware of existing DTW approaches for branched trajectory alignment[10]. Output pairs of correspondences from them could undergo G2G alignment to capture gene-level mismatches.

In summary, G2G enables deeper understanding of the diversity of gene alignments between single-cell datasets. It is available as an open-source Python package at https://github.com/Teichlab/Genes2Genes. We demonstrated that regenerative medicine can specifically benefit from trajectory comparisons by extracting cues to guide refinement of in vitro cell engineering. We envision that G2G will be useful to the community for exploring other scenarios such as cell activation or stimulation responses in control and disease, generating insights to advance our understanding of cell development and function in health and disease.

## Online content

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

## Methods

### Genes2Genes: a new alignment framework for single-cell trajectories

DP[4] remains central to many sequence alignment algorithms.

G2G performs gene-level pseudotime trajectory alignment between a single-cell reference and query, by running DP alignment independently for all genes of interest. It aims to generate an optimal sequence of matched and mismatched time points for each gene. There are five different alignment states possible between two reference and query time points (Fig. 1b). For each time point in any gene trajectory, there is a respective expression distribution as observed via scRNA-seq measurements. G2G evaluates the distances of these distributions in reference and query to infer an optimal gene alignment.

### Pairwise time series alignment for trajectory comparison

Trajectory is a continuous path of change through some feature space, along an axis of progression (such as time)[59]. In single-cell transcriptomics, this feature space is often defined by genes. A trajectory through a high-dimensional gene space describes the state of a cell as a function of time. The pseudotime of cells represents a discretization of their cell-state trajectory. Their genes form a multivariate time series of expression, with each gene as univariate. In this work, we consider pairwise alignment of univariate time series, which enables gene-level trajectory alignment.

Given two discrete time series (sequences), reference $R$ and query $Q$ of length (a finite number of time points) $|R|$ and $|Q|$, their pairwise alignment describes an optimal sequential mapping between their time points. As an optimization problem, this has two key properties: (1) an optimal substructure; and (2) overlapping set of subproblems, which make it solvable by DP. Property (1) means, the optimal alignment of any two prefixes $R_{1..j}$ and $Q_{1..i}$ depends on the optimality of three subalignments: (i) $R_{1..j-1}$ and $Q_{1..i-1}$; (ii) $R_{1..j-1}$ and $Q_{1..i}$; and (iii) $R_{1..j}$ and $Q_{1..i-1}$. Property (2) means, there exists prefix alignments that are overlapping. DP begins optimizing prefix alignment, starting from null ($\Phi$) sequences until it completes aligning the entire two sequences. This process computes overlapping subproblems only once and reuses them through a memoization (history) matrix $Hist$. In standard DP alignment, $Hist(i,j)$ stores the optimal alignment cost of the two prefixes: $R_{1:j}$ and $Q_{1:i}$, by optimizing an objective function that quantifies the alignment through a set of recurrence relations. Once $Hist$ is computed, the optimal alignment can be retrieved by backtracking, starting from $Hist(|Q| + 1, |R| + 1)$ until reaching $Hist(0, 0)$.

### Preprocessing a trajectory time series by distributional interpolation

Interpolation of time series is necessary to ensure smoothly changing and uniformly distributed data (at least approximately). This is because non-interpolated data cannot guarantee a reliable alignment[7,13]. We interpolate all reference/query gene expression trajectories before alignment, by extending CellAlign's[7] mean-based interpolation method to distributional interpolation.

Given pseudotime series $t$ of (log1p-normalized) expression in gene $g_j$ of a single-cell dataset, we first transform the pseudotime axis to [0,1] range using min–max normalization.

Then, $m$ equispaced artificial (interpolation) time points are defined, and for each interpolation time point $t'$, we estimate a Gaussian distribution (of mean $g_j(t')_{mean}$ and s.d. $g_j(t')_{s.d.}$) using the Gaussian kernel-based weighted approach. For each cell annotated with pseudotime $t_i$, a weight is computed with respect to each $t'$ as:

$$w_i = \exp\left(-\frac{(t_i - t')^2}{\text{window\_size}^2}\right),$$

where $\text{window\_size} = 0.1$. Below equations estimate $g_j(t')_{mean}$ and $g_j(t')_{s.d.}$:

$$g_j(t')_{mean} = \frac{1}{\Sigma_i w_i} \sum_{i=1}^{n} w_i g_j(t_i)$$

$$g_j(t')_{s.d.} = c_{t'}\sqrt{n\frac{\sum_{i=1}^{n} w_i[g_{j\_mean} - g_j(t_i)]^2}{(n-1)\Sigma_i w_i}}$$

where $g_{j\_mean} = \frac{\sum_{i=1}^{n} g_j(t_i)}{n}$, $n$ is the total number of cells, and $c_{t'}$ is the expected weighted cell density at $t'$, that is, $c_{t'} = \frac{\sum_{i=1}^{n} w_i}{n}$, used to account for cell abundance when estimating variance (otherwise a very few cells may give very high variance). Next, we generated 50 random points from Gaussian distribution $N(g_j(t')_{mean}, g_j(t')_{s.d.})$ for each $t'$, representing the interpolated distribution of single-cell gene expression. Note that we used a predefined $m$ for both reference and query. The interpolation has $O(nm)$ time complexity due to taking weighted contribution from all cells at each $t'$. For efficiency, one could subsample datasets and/or restrict the contributing cells to the nearest neighborhood.

**Extreme cases.** When the smooth trajectory assumption breaks (for example, pluripotent genes suddenly dropping to zero mid-way after highly expressed in early development), the interpolated variance might not reflect the true observation. Also, when a gene is (almost) zero-expressed, there is no distribution to model. To handle such extreme cases when interpolating reference and query gene trajectories, G2G applies the below steps:

- For either trajectory, check for regions (adjacent interpolation points) showing abrupt zero-expression (<3 cells expressed) in considerable lengths (exceeding 0.2), by sliding-window-scanning; apply a common $\sigma$ (10% of the minimum $\sigma$ estimated across all interpolation points) for those regions to have a very low $\sigma$ with zero mean.
- If either gene is zero-expressed (<3 cells expressed) across pseudotime, apply 10% of the minimum $\sigma$ estimated across all interpolation points of the other trajectory as the $\sigma$ for the extreme-case trajectory and vice versa.

### A new DP algorithm for gene-level trajectory alignment

Our DP algorithm is inspired by biological sequence alignment discussed in the related literature[22–24]. It generates an alignment between $R$ and $Q$ expression time series of a specified gene by adapting Gotoh's algorithm[16] and DTW[5] to accommodate five alignment states (Fig. 1b), that is, one-to-one match (M), one-to-many match (V), many-to-one match (W), insertion (I) and deletion (D), between a pair of $R$ and $Q$ time points. The five-state space is denoted by $\Omega = [M, V, W, D, I]$. V and W represent warps.

DTW[5] is extensively used to align time series. Sankoff and Kruskal (1983)[13] previously discussed how to capture both warps and indels from a single algorithm and provided DP recurrences for evaluating all states in $\Omega$ to assign an optimal state for each pair of $R$ and $Q$ time points. Extending this further, we implemented Gotoh's algorithm (of $O(|R||Q|)$ time complexity) to generate an optimal five-state alignment string for $R$ and $Q$ using:

- a Bayesian information-theoretic distance measure between two expression distributions under the minimum message length inference criterion[13,26].
- a five-state machine that models state transitions along an alignment.

The DP scoring scheme evaluates every pair of $R$ time point $j$ ($R_j$) and $Q$ time point ($Q_i$) by computing two costs: (1) the cost of matching them (denoted by $\text{Cost}_{match}(i,j)$) based on their interpolated gene expression distributions, and (2) the cost of assigning an alignment state $x \in \Omega$ for them.

## The DP scoring scheme

**The cost of match between $R_j$ and $Q_i$.** $R_j$ and $Q_i$ are expected to match if they have similar expression distributions. To score their match likelihood, we define a cost (distance measure) between the two expression distributions of $R_j$ and $Q_i$, modeled as two Gaussians. We compute this over the interpolated single-cell expression data at $R_j$ (denoted by $R(j)$ under $N(\mu_{R(j)}, \sigma_{R(j)})$) and $Q_i$ (denoted by $Q(i)$ under $N(\mu_{Q(i)}, \sigma_{Q(i)})$) with respective mean ($\mu$) and s.d. ($\sigma$) statistics. Accordingly, if $D_{R(j)} = \{d_k\}_{k=1}^{|R(j)|}$ and $D_{Q(i)} = \{d_k\}_{k=1}^{|Q(i)|}$ are their expression data vectors, then: $d_k \sim N(\mu_{R(j)}, \sigma_{R(j)}) \,\forall\, d_k \in D_{R(j)}$ and $d_k \sim N(\mu_{Q(i)}, \sigma_{Q(i)}) \,\forall\, d_k \in D_{Q(i)}$.

Hereafter, we denote $N(\mu_{R(j)}, \sigma_{R(j)})$ by $N_{R(j)}$ and $N(\mu_{Q(i)}, \sigma_{Q(i)})$ by $N_{Q(i)}$.

We implement the cost function, $\text{Cost}_{\text{match}}(i,j)$, to consider both data ($D_{R(j)}$ and $D_{Q(i)}$) and models ($N_{R(j)}$ and $N_{Q(i)}$) when computing the distance between $R(j)$ and $Q(i)$, using the MML criterion[26,27]. See Fig. 2 (top left) and Supplementary Fig. 1a for illustrations of our MML framework.

*Primer on MML.* MML is an inductive inference paradigm for model comparison and selection, grounded on Bayesian statistics and information theory. Given a hypothesis (model) $H$ and data $D$, it lays an imaginary message transmission from a sender who jointly encodes $H$ and $D$, for lossless decoding at a recipient's side. Bayes theorem defines their joint probability as:

$$\Pr(H,D) = \Pr(H)\Pr(D|H) = \Pr(D)\Pr(H|D)$$

Separately, Shannon information[28] ($I$) defines the optimal encoding length of an event $E$ with probability $\Pr(E)$ as:

$$I(E) = -\log_e(\Pr(E))$$

measured in nits. Applying this to Bayes theorem describes the information needed to encode $H$ and $D$ jointly as:

$$I(H,D) = I(H) + I(D|H) \tag{1}$$

This gives a two-part total encoding length for $H$ and $D$, where $I(H)$ quantifies the information of $H$, and $I(D|H)$ quantifies the information of $D$ using $H$. When two hypotheses, $H_1$ and $H_2$, describe $D$, MML enables selecting the best hypothesis with model complexity versus model-fit tradeoff, by evaluating a compression statistic $\Delta = I(H_1, D) - I(H_2, D)$, which gives the log odds posterior ratio between them.

$$\Delta = \log\left(\frac{\Pr(H_2, D)}{\Pr(H_1, D)}\right) = \log\left(\frac{\Pr(D)\Pr(H_2|D)}{\Pr(D)\Pr(H_1|D)}\right) = \log\left(\frac{\Pr(H_2|D)}{\Pr(H_1|D)}\right) \tag{2}$$

$\Delta > 0$ implies that $H_2$ is $e^\Delta$ times more likely than $H_1$ and vice versa.

*Casting $\text{Cost}_{\text{match}}(i,j)$ under MML.* Given the data $D$ ($D_{R(j)}$ and $D_{Q(i)}$) and Gaussian models ($N_{R(j)}$ and $N_{Q(i)}$), we formulate two hypotheses:

- Hypothesis A ($R_j$ and $Q_i$ match): explains $D$ with a single, representative model $N(\mu_*, \sigma_*)$ denoted by $N_*$ (= either $N_{R(j)}$ or $N_{Q(i)}$),
- Hypothesis $\Phi$ ($R_j$ and $Q_i$ mismatch): explains $D_{R(j)}$ with $N_{R(j)}$ and $D_{Q(i)}$ with $N_{Q(i)}$, independently,

and compute, $I(A,D)$ and $I(\Phi,D)$ according to equation (1):

$$I(A,D) = I(A) + I(D|A) \tag{3}$$

$$I(\Phi,D) = I(\Phi) + I(D|\Phi) \tag{4}$$

where, $A = [N_*]$ and $\Phi = [N_{R(j)}, N_{Q(i)}]$. Accordingly, equation (3) becomes:

$$I(A,D) = I(N_*) + I(D|N_*) = I(\mu_*, \sigma_*) + I(D|\mu_*, \sigma_*)$$

Similarly, equation (4) becomes:

$$\begin{aligned} I(\Phi,D) &= I(N_{R(j)}, N_{Q(i)}) + I(D|N_{R(j)}, N_{Q(i)}) \\ &= I(N_{R(j)}) + I(N_{Q(i)}) + I(D|N_{R(j)}, N_{Q(i)}) \\ &= I(\mu_{R(j)}, \sigma_{R(j)}) + I(\mu_{Q(i)}, \sigma_{Q(i)}) \\ &\quad + I(D_{R(j)}|\mu_{R(j)}, \sigma_{R(j)}) + I(D_{Q(i)}|\mu_{Q(i)}, \sigma_{Q(i)}) \end{aligned}$$

The next section describes how equations (3) and (4) terms are calculated. We normalize $I(\Phi,D)$ and $I(A,D)$ to compute per-datum information (entropy):

$$I(A,D)_{\text{entropy}} = \frac{I(A,D)}{|D_{R(j)}| + |D_{Q(i)}|}$$

$$I(\Phi,D)_{\text{entropy}} = \frac{I(\Phi,D)}{|D_{R(j)}| + |D_{Q(i)}|}$$

Note that the $I(A,D)_{\text{entropy}}$ measure is made symmetric as:

$$I(A,D)_{\text{entropy}} = \frac{I(N_{R(j)}, D)_{\text{entropy}} + I(N_{Q(i)}, D)_{\text{entropy}}}{2} \text{ nits per datum}$$

We then define the compression statistic $\Delta$ as our $\text{Cost}_{\text{match}}(i,j)$:

$$\Delta = I(A,D)_{\text{entropy}} - I(\Phi,D)_{\text{entropy}}$$

When $R(j)$ and $Q(i)$ are significantly dissimilar, $I(A,D)_{\text{entropy}} > I(\Phi,D)_{\text{entropy}}$. Thus, $\text{Cost}_{\text{match}}(i,j)$ increases when distributions diverge (Extended Data Fig. 1b and Supplementary Fig. 1b,c).

*Computing the Shannon Information of any Gaussian model and data.* $\text{Cost}_{\text{match}}(i,j)$ computation uses MML Wallace–Freeman approximation[26,28] defined for Gaussian distributions[27,60]. As in equation (1), for any dataset $D$ and hypothesis $H$ describing $D = \{x_k\}_{k=1}^{X}$ under $N(\mu, \sigma)$ with parameters $\vec{\theta} = (\mu, \sigma)$, the information of $H$ and $D$ is:

$$I(H,D) = I(\vec{\theta}, D) = I(\vec{\theta}) + I(D|\vec{\theta})$$

expanding to:

$$I(\vec{\theta}, D) = \frac{d}{2}\log(\kappa_d) - \log[h(\vec{\theta})] + \frac{1}{2}\log(\det[\text{Fisher}(\vec{\theta})]) + L(\vec{\theta}) + \frac{d}{2},$$

where $d$ is the number of free parameters ($d = 2$ for a Gaussian) and $\kappa_d$ is the Conway lattice constant[61] ($\kappa_d$ is $\frac{5}{36\sqrt{3}}$ for $d = 2$); $h(\vec{\theta})$ is the prior over $\vec{\theta}$. $\mu$ and $\log(\sigma)$ are defined with uniform priors over predefined ranges of length $R_\mu$ and length $R_\sigma$, respectively:

$$h(\vec{\theta}) = h(\mu)\,h(\sigma) = \left(\frac{1}{R_\mu}\right)\left(\frac{1}{\sigma R_\sigma}\right)$$

$$\Rightarrow I[h(\vec{\theta})] = \log(\sigma) + \log(R_\mu R_\sigma)$$

We use $R_\mu = 15.0$ and $R_\sigma = 3.0$ as reasonable for log-normalized expression (for example, across 20,240 genes, we observe ~8.1 maximum expression and ~1.7 maximum $\sigma$ in the pan fetal reference).

$L(\vec{\theta})$ is the negative log likelihood:

$$L(\vec{\theta}) = X\log(\sigma) + \frac{X}{2}\log(2\pi) + \frac{1}{2\sigma^2}\sum_{i=1}^{X}(x_i - \mu)^2 - \sum_{i=1}^{X}\log(\epsilon)$$

where, $\epsilon$ is the precision of datum measurement (taken as $\epsilon = 0.001$). $\det[\text{Fisher}(\vec{\theta})]$ is the determinant of the expected Fisher matrix (the matrix of the expected second derivatives of the negative log-likelihood function), which has the closed form $\frac{2X^2}{\sigma^4}$.

**The cost of alignment state assignment for $R_j$ and $Q_i$.** The DP scoring scheme also involves a cost of assigning an alignment state $x \in \Omega = [\text{M}, \text{W}, \text{V}, \text{D}, \text{I}]$ for $R_j$ and $Q_i$. This is computed as the Shannon information[28] required to encode state $x$ given previous state $y$ assigned for the preceding time points, $I(x|y) = -log_e(\Pr(x|y))$. We define a five-state machine (middle left of Fig. 2) to explain these conditional probabilities (also called state transitions), by extending the three-state machine[22,23] of [M, I, D] to accommodate [W, V] warp states (Fig. 1b). We enforce symmetry while treating <I and D> and <W and V> equivalently and prohibiting transitions, $\text{I} \rightarrow \text{W}$ and $\text{D} \rightarrow \text{V}$, as they imply a single M; however, we can allow $\text{D} \rightarrow \text{W}$ and $\text{I} \rightarrow \text{V}$, as there can be a case of a warp match after an insertion or deletion. All outgoing transitions of each state sum up to a probability of 1. Overall, there are 23 state transitions in this machine, yet with only three free transition probability parameters [Pr(M|M), Pr(I|I) and Pr(M|I)] due to its symmetry and characteristics. These probabilities control the expected lengths of matches and mismatches (reflecting an affine gap scheme). In this work, we chose [Pr(M|M) = 0.99, Pr(I|I) = 0.1, Pr(M|I) = 0.7] as the default in G2G based on a grid search that minimized the alignment inaccuracy rate in our simulated dataset 1. An interesting future direction would be to infer them using an added optimization layer on top of DP optimization.

Altogether, the G2G DP scoring scheme utilizes $\text{Cost}_{\text{match}}(i,j)$ and the five-state machine (with state-assignment costs evaluated as $I(x|y) \; \forall \; x, y \in \Omega$), to define DP recurrence relations.

*DP recurrence relations.* We define the $\{\text{Hist}_x\}_{\forall x \in \Omega}$ matrices corresponding to the alignment states in $\Omega$. Every $\text{Hist}_x$ has $(|Q| + 1 \times |R| + 1)$ dimensions, where the columns and rows correspond to $R$ and $Q$ time points, respectively. $\text{Hist}_x(i,j)$ stores the optimal alignment cost of prefixes $R_{1..j}$ and $Q_{1..i}$ ending in state $x$. The DP recurrences to compute $\text{Hist}_x(i,j)$ for $i > 0, j > 0$ are:

$$\text{Hist}_\text{M}(i,j) = \min \begin{cases} \text{Cost}_\text{match}(i,j) + \text{Hist}_\text{M}(i-1,j-1) + I(\text{M}|\text{M}) \\ \text{Cost}_\text{match}(i,j) + \text{Hist}_\text{W}(i-1,j-1) + I(\text{M}|\text{W}) \\ \text{Cost}_\text{match}(i,j) + \text{Hist}_\text{V}(i-1,j-1) + I(\text{M}|\text{V}) \\ \text{Cost}_\text{match}(i,j) + \text{Hist}_\text{D}(i-1,j-1) + I(\text{M}|\text{D}) \\ \text{Cost}_\text{match}(i,j) + \text{Hist}_\text{I}(i-1,j-1) + I(\text{M}|\text{I}) \end{cases}$$

$$\text{Hist}_\text{W}(i,j) = \min \begin{cases} \text{Cost}_\text{match}(i,j) + \text{Hist}_\text{M}(i,j-1) + I(\text{W}|\text{M}) \\ \text{Cost}_\text{match}(i,j) + \text{Hist}_\text{W}(i,j-1) + I(\text{W}|\text{W}) \\ \text{Cost}_\text{match}(i,j) + \text{Hist}_\text{V}(i,j-1) + I(\text{W}|\text{V}) \\ \text{Cost}_\text{match}(i,j) + \text{Hist}_\text{D}(i,j-1) + I(\text{W}|\text{D}) \end{cases}$$

$$\text{Hist}_\text{V}(i,j) = \min \begin{cases} \text{Cost}_\text{match}(i,j) + \text{Hist}_\text{M}(i-1,j) + I(\text{V}|\text{M}) \\ \text{Cost}_\text{match}(i,j) + \text{Hist}_\text{W}(i-1,j) + I(\text{V}|\text{W}) \\ \text{Cost}_\text{match}(i,j) + \text{Hist}_\text{V}(i-1,j) + I(\text{V}|\text{V}) \\ \text{Cost}_\text{match}(i,j) + \text{Hist}_\text{I}(i-1,j) + I(\text{V}|\text{I}) \end{cases}$$

$$\text{Hist}_\text{D}(i,j) = \min \begin{cases} \text{Hist}_\text{M}(i,j-1) + I(\text{D}|\text{M}) \\ \text{Hist}_\text{W}(i,j-1) + I(\text{D}|\text{W}) \\ \text{Hist}_\text{V}(i,j-1) + I(\text{D}|\text{V}) \\ \text{Hist}_\text{D}(i,j-1) + I(\text{D}|\text{D}) \\ \text{Hist}_\text{I}(i,j-1) + I(\text{D}|\text{I}) \end{cases}$$

$$\text{Hist}_\text{I}(i,j) = \min \begin{cases} \text{Hist}_\text{M}(i-1,j) + I(\text{I}|\text{M}) \\ \text{Hist}_\text{W}(i-1,j) + I(\text{I}|\text{W}) \\ \text{Hist}_\text{V}(i-1,j) + I(\text{I}|\text{V}) \\ \text{Hist}_\text{D}(i-1,j) + I(\text{I}|\text{D}) \\ \text{Hist}_\text{I}(i-1,j) + I(\text{I}|\text{I}) \end{cases}$$

They are initialized as:

$$\text{Hist}_x(i,j)_{x \in \{\text{M},\text{W},\text{V}\}} = \begin{cases} 0; & i > 0, j > 0 \\ \infty; & \text{otherwise} \end{cases}$$

$$\text{Hist}_\text{I}(i,j) = \text{Hist}_\text{I}(i-1,0) + I(\text{I}|\text{I})$$

$$\text{Hist}_\text{D}(i,j) = \text{Hist}_\text{D}(0,j-1) + I(\text{D}|\text{D})$$

Note that for the cases of <$i = 1$ and $j = 1$> (before the first state transition), either a uniform transition cost ($I(\text{M}) = I(\text{I}) = I(\text{D}) = -log_e(1/3)$) or a setting with lower cost for M can be assigned.

Once matrices are complete, G2G generates the optimal alignment $Y^*$ for $R$ and $Q$ as a five-state string $Y^*_{str}$ (where character $Y^*_{str}[k] \in \Omega \; \forall k \in \mathbb{Z}_{[0,|Y^*|]}$), by backtracking from:

$$\min \begin{cases} \text{Hist}_\text{M}(|Q|,|R|) \\ \text{Hist}_\text{W}(|Q|,|R|) \\ \text{Hist}_\text{V}(|Q|,|R|) \\ \text{Hist}_\text{D}(|Q|,|R|) \\ \text{Hist}_\text{I}(|Q|,|R|) \end{cases}$$

The optimal cost landscape matrix $L$ can be constructed as:

$$L(i,j) = \min_{\forall x \in \Omega}\{\text{Hist}_x(i,j)\}$$

$Y^*$ describes the set of $R$ and $Q$ time point pairs matched and the set of $R$ and $Q$ time points mismatched, sequentially. Let $T_\text{matched}$ be the set of matched time point pairs $(i,j)$ in $Y^*$. The total alignment cost of $Y^*$ is the sum of the total match cost ($C_\text{match}$) and the total state-assignment cost ($C_\text{state}$), where:

$$C_\text{match}(Y^*) = \sum_{\forall (i,j) \in T_\text{matched}} \text{Cost}_\text{match}(i,j)$$

$$C_\text{state}(Y^*) = \sum_{k=1}^{|Y^*|} I(Y^*[k]|Y^*[k-1])$$

Overall, $Y^* = \arg\min_{\forall Y \in \mathbf{Y}}\{C_\text{match}(Y) + C_\text{state}(Y)\}$, where $\mathbf{Y}$ is the space of all possible five-state alignments.

Note that $\text{Cost}_\text{match}(i,j)$ can be any cost function (for example, KL divergence) that can measure the distance between two expression distributions; however, MML distance enables defining complete descriptions for hypotheses, considering both model complexity and data fit, unlike KL divergence, which computes the expected log-likelihood ratio, disregarding model complexity.

## Reporting alignment statistics over gene-level alignments
**Distribution of alignment similarities.** The distribution of 'alignment similarity' statistics (percentage of [M, V, W] in the five-state string generated by G2G for each gene) and their average 'alignment similarity' statistic across all genes, quantify the degree of concordance between the reference and query. The genes are ranked from the temporally most distant to most similar using those alignment similarities.

**Aggregate alignment.** G2G generates a single, cell-level (average) alignment across all genes (or any subset of genes) using their optimal alignment landscapes ($L$ matrices). $L(i,j)$ gives the optimal ending state of the prefixes, $R_{1..j}$ and $Q_{1..i}$. Across all gene-specific $L$ matrices, there is a five-state frequency distribution for each $R_j$ and $Q_i$. To generate an aggregate alignment, we begin traversal from $L(|Q| + 1, |R| + 1)$ and choose the most probable state $x \in \Omega$ for $R_{|R|}$ and $Q_{|Q|}$ as the most frequent across all genes. Accordingly, we traverse to the next matrix cell

and repeat the same process until we reach $L(0,0)$; for any $L(i,j)$, if $x = $ M, the next will be $L(i-1, j-1)$ and if $x = $ D, the next will be $L(i, j-1)$ and so on. Finally, we have an aggregate five-state alignment string.

**Clustering alignment patterns.** We employ agglomerative hierarchical clustering under average linkage criterion (in sklearn v.1.2.2) to identify groups of genes that show similar alignment patterns, given a pairwise distance matrix between all alignment strings. The distance threshold parameter for the linkage controls where the cluster merge stops, allowing inspection of different clustering structures at different levels in the hierarchy.

*Defining a distance measure for five-state alignment strings.* Clustering alignments require defining a distance measure between two alignment paths. While the polygonal-area-based distance measure[62] works for three-state alignments, it cannot distinguish warps from indels. The commonly used string distance measures are: Levenshtein distance and Hamming distance. Levenshtein distance is the minimum number of edits (substitutions, inserts and deletes) needed to transform one string to another. G2G computes pairwise Levenshtein distances between alignment strings (using leven v.1.0.4), normalized by the maximum length of the strings in comparison. Hamming distance is the minimum number of single-character substitutions needed to transform equal-length strings to one another. G2G computes pairwise Hamming distances using scipy.spatial.distance.cdist (in SciPy v.1.10.1), using the alignment strings encoded as equal-length binary strings (of size $|R| + |Q|$) (Supplementary Fig. 6a). Each alignment string is binary-encoded by traversing through its alignment path, recording for each $R$ and $Q$, the match/mismatch state $x$ of their respective pseudotime points ($x \in [$M, V, W$]$ is encoded by 1; $x \in [$I, D$]$ is encoded by 0). These $R$ and $Q$ binary strings are then concatenated. Note that both Levenshtein and Hamming distances are normalized to range [0,1].

*Choosing the right string distance measure.* We tested both Levenshtein and Hamming distances. In hierarchical clustering, the number of clusters decreases as the distance threshold increases. Ideally, the bottom level of an optimal hierarchical clustering of strings shall represent each unique string in all strings (that is, the maximum number of clusters at the minimum distance threshold shall be equal to the number of unique strings). When clustering with Levenshtein distance, we observe this across all datasets. Hamming distance, however, does not guarantee such capture (Supplementary Fig. 6b). This agrees with the theoretical expectation that Levenshtein distance can distinguish all five individual states, whereas Hamming distance can only distinguish matches and mismatches. Therefore, we recommend Levenshtein distance for alignment clustering.

*Choosing the distance threshold for hierarchical clustering.* A common strategy is to empirically determine the distance threshold based on the mean silhouette coefficient[63] (MSC) over all data samples. MSC ranges [−1,1], where a high positive MSC indicates well-separated clusters, a low positive MSC closer to 0 indicates overlapping clusters, and a low negative MSC indicates incorrect assignments. We obtain clustering for eligible thresholds in the range [0,1.0] with 0.01 step size, and compute their MSCs using sklearn.metrics.silhouette_score.

*Hierarchical clustering of alignment strings requires a tradeoff.* Generally, the best clustering is considered as the one with the highest MSC; however, for strings, we observe that this value is given by the maximum possible number of clusters (equal to the number of unique alignment strings). In gene-level trajectory alignment, many unique alignment patterns can emerge due to subtle differences in their optimal alignment states across pseudotime points. For instance, in our simulated dataset, there are 113 unique strings covering seven alignment patterns (Extended Data Fig. 2c). Our objective is a less noisy,

biologically interpretable clustering, and we note the importance of manual inspection to decide on a tradeoff between the number of clusters versus cluster resolution. We recommend choosing the distance threshold that provides a good tradeoff between MSC and the number of clusters in capturing the main alignment patterns. In our simulated dataset, such a tradeoff is given by the threshold 0.22 corresponding to the second highest locally optimal MSC 0.82. This results in 15 clusters, including the seven major clusters giving only a 0.1% mis-clustering rate (the percentage of the number of outliers in all clusters) (Fig. 3e). The rest of the clusters are mini clusters covering 31 (0.8%) alignments separated due to noise such as warps. G2G enables the user to inspect these cluster diagnostics through the distance threshold versus MSC plots and the cluster-specific average alignment patterns.

**Pathway over-representation analysis.** We select the top $k$ mismatching genes (with ≤40% alignment similarity) to analyze their biological/signaling pathway over-representation. The identified clusters of genes are also analyzed. We use GSEApy (v.1.0.4) Enrichr[62,64,65] wrapper against the MSigDB_Hallmark_2020 (ref. 66) and KEGG_2021_Human pathway gene sets[66,67]. For all analyses, a 0.05 significance threshold of the adjusted $P$ value (computed using the default hypergeometric test and Benjamini–Hochberg false discovery rate (FDR) correction of GSEApy-enrichr interface) was applied.

## Determining the best parameter setting

G2G has several key parameters: interpolation structure, window_size of the Gaussian kernel used for interpolation and the five-state machine parameters.

**Interpolation structure.** The number of equispaced interpolation time points ($m$) over the [0,1] range decide the resolution of a trajectory alignment (a higher $m$ gives higher resolution).

Low resolution can be less representative of the dynamic process, whereas high resolution introduces noise or redundancy. The optimal $m$ is a tradeoff that depends on the datasets. We use optBinning[68] (v.0.18.0) to heuristically decide the optimal $m$ for reference and query, separately. Using ContinuousOptimalBinning, we first infer an optimal binning of the pseudotime distribution and then use the number of bins produced as the $m$ for our equispaced interpolation. In all datasets except for the T cell datasets, optBinning returned an equal number of optimal bins for both reference and query. For T cell datasets, we obtained 15 and 14 bins, respectively. For consistency, we selected the minimum (14). We do not use the optimal splits returned by optBinning as this is an irregular binning structure that is inconsistent for alignment.

**Window_size.** This controls the effective cell neighborhood toward estimating the weighted mean and variance of expression at each interpolation time point. CellAlign[7] found that 0.1 window_size is the most effective for standard single-cell datasets (with a tradeoff between noise and locality); thus, we use the same across all our experiments and analyses.

**Five-state machine.** The parameters [Pr(M|M), Pr(I|I) and Pr(M|I)] were optimized using grid search, while fixing Pr(M|M) = 0.99 to enforce the highest probability for continuous matches rather than single-point-matches. [Pr(I|I) = 0.1, Pr(M|I) = 0.7 yielded the lowest false mismatch rate across all G2G alignments on our simulated dataset. It remained optimal when varying Pr(M|M) in [0.1,1.0] (Supplementary Table 2). Therefore we set it as the default. For initial states, we use [Pr (M) = 0.99 and Pr(D) = Pr(I) = $5 \times 10^{-5}$].

## Benchmarking against CellAlign and TrAGEDy alignment

DTW gene-level and cell-level high-dimensional alignments were generated using CellAlign's[7] (v.0.1.0) globalAlign function,

following interpolation and scaling defined in their documentation. DTW gene-level alignments were clustered using CellAlign's pseudo-timeClust function. Similarly, TrAGEDy's post hoc-processed DTW alignments were generated using the script published by Laidlaw et al.[17], following documentation. The same number of interpolation time points was used across all CellAlign[7], TrAGEDy and G2G alignment. Both CellAlign[7] and TrAGEDy ran with Euclidean distance for DTW (note that TrAGEDy recommends Spearman correlation which is mathematically undefined for single-gene observations, thus we use Euclidean distance for both cell-level and gene-level TrAGEDy alignment for consistency).

## Datasets
**Datasets for simulated experiments.** *Simulating different alignment patterns using Gaussian processes.* We modeled log-normalized expression of gene $x$ as a function $f$ of time $t$ using a Gaussian process (GP):

$$f(t) \sim GP(\vec{\mu}(t), K(t,t'))$$

$\mu$ is the mean vector. $K(t,t')$ is a kernel function evaluating covariance of every pair of finite time points, where $f(t)$ is evaluated, controlling the $f(t)$ characteristics (for example, radial basis function (RBF) kernel for generating smooth, non-branching functions; a change point kernel for generating branching functions). GP with a suitable kernel can simulate different trajectory patterns in single-cell gene expression across pseudotime. Following the standard textbook and kernels discussed in literature[69,70], we implemented a simulator using GPyTorch (v.1.5.1) for three types of alignment patterns (matching, divergence and convergence), comprising 300 cells spread across pseudotime range [0,1] for each trajectory.

*Generating a matching pair of reference and query gene trajectories.* We used a GP with a constant $c$ mean vector $\vec{\mu}_c$ ($c \in [0.5, 9.0]$ uniform random sampled) and RBF kernel $K$ to sample $\mu(t)$ that describes an average expression for each time point. Next, we sampled two trajectories: $GEX_{ref}(t)$ and $GEX_{query}(t)$, from a GP with $\mu(t)$ and kernel $\sigma^2 I$ ($\sigma \in [0.05, 1.0]$ uniform random sampled and $I$ = identity matrix).

$$\mu(t) \sim N(\vec{\mu}_c, K)$$

$$GEX_{ref}(t) \sim N(\mu(t), \sigma^2 I)$$

$$GEX_{query}(t) \sim N(\mu(t), \sigma^2 I)$$

*Generating a divergence pair of reference and query gene trajectories.* We used a change point (CP) kernel, which imposes a bifurcation in a trajectory as it reaches an approximate time point $t_{CP}$ (also called a change point). It activates one covariance function before $t_{CP}$ and another after $t_{CP}$. We used the below CP kernel[69,70]:

$$K_{CP}(t,t') = aK_1(t,t') + a'K_2(t,t')$$

where,

$$a = \sigma(t)\sigma(t')$$

$$a' = [1 - \sigma(t)][1 - \sigma(t')]$$

$$\sigma(x) = \frac{1}{1 + \exp(-s(x - t_{CP}))} \text{ (sigmoid function)}$$

with $s$ acting as CP steepness parameter. Penfold et al.[69] defines a branching process by enforcing a zero kernel ($K_1$) before $t_{CP}$ and another suitable kernel ($K_2$) afterwards. We used RBF for $K_2$. Following is the generative process, starting with a base mean function $\mu(t)$ sampled

from a separate GP with constant $c$ mean vector $\vec{\mu}_c$ ($c \in [0.5, 9.0]$ uniform randomly sampled) and an RBF kernel $K$.

$$\mu(t) \sim N(\vec{\mu}_c, K)$$

$$f_1(t) \sim N(\mu(t), K_{CP})$$

$$f_2(t) \sim N(\mu(t), K_{CP})$$

$$GEX_{ref}(t) \sim N(f_1(t), \sigma^2 I)$$

$$GEX_{query}(t) \sim N(f_2(t), \sigma^2 I)$$

Next, two functions were sampled from a GP with $\mu(t)$ and CP(t,t'), which were then used as mean vectors to generate $GEX_{ref}(t)$ and $GEX_{query}(t)$ with kernel $\sigma^2 I$ ($\sigma = 0.3$, a moderate constant). This ran for [$t_{CP} = 0.25$, $t_{CP} = 0.5$ and $t_{CP} = 0.75$] resulting in three groups of divergence with varying bifurcation points (early divergence, mid divergence and late divergence). We then filtered the generated pairs to include simple/clear divergence patterns (stable ground truth with no complex patterns) using basic heuristics such as the difference between mean expression before divergence and at the end terminals of reference and query.

Extended Data Fig. 1a–c shows that the branching may start approximately before CP. Therefore, we expect the early nondivergent segment to continue at least until time point $i < t_{CP}$, where we begin to see >0.01 covariance in the CP kernel. Accordingly, given our approx_bifurcation_start_point = $i$, we expect the range of match lengths to fall between a lower-limit n_total_pseudotime_points $\times i$ and upper-limit n_total_pseudotime_points $\times$ change_point.

Equivalently, we expect mismatch lengths to fall between lower-limit n_total_pseudotime_points $\times (1 - i)$ and upper-limit n_total_pseudotime_points $\times (1 - \text{change\_point})$.

*Generating a convergence pair of reference and query gene trajectories.* We simply inverted the above generated divergence pairs, as convergence and divergence are complementary to each other.

The final dataset comprises 3,500 pairs, covering seven alignment patterns: 500 matching pairs, 1,500 divergence pairs (500 early + 500 mid + 500 late) and 1,500 convergence (500 early + 500 mid + 500 late).

### Simulating mismatches on real scRNA-seq data
We downloaded the E15.5 mouse pancreas development dataset[29] from the CellRank package[71], and subsetted to β-cell lineage (1,845 cells) using the original author annotations ('Ngn3 low EP', 'Ngn3 high EP', 'Fev+' and 'Beta'). We selected the lineage-driver genes that are significantly associated with differentiation potential to β-cells (769 of 2,000 highly variable genes at 1% FDR), using CellRank (v.1.5.1) following their tutorials (https://cellrank.readthedocs.io/en/stable/auto_examples/estimators/compute_lineage_drivers.html). For trajectories, we used the diffusion pseudotime estimated by the CellRank authors. To simulate trajectories for alignment, we divided the pseudotime (between 0 and 1) equally into 50 bins, assigned cells to bins based on their pseudotime and randomly split cells into reference and query datasets in each bin. To simulate deletions of $n$ bins, we excluded query cells from the first $n$ bins (cells where the pseudotime ≤$n$th bin upper margin). To simulate insertions of $n$ bins, we shifted the query cell expression of the first $n$ bins by the s.d. calculated across all bins for the gene of interest in the query cells. Query pseudotime axis was min–max-normalized after perturbation.

### Datasets for benchmarking G2G
**Dendritic cell stimulation dataset.** The normalized single-cell datasets of PAM/LPS stimulation and their pseudotime estimates (downloaded

from the CellAlign[7] GitHub repository) were converted into Anndata objects. These contain two gene sets: 'core antiviral module' (99 genes) and 'peaked inflammatory module' (89 genes), preselected from the original publication[20] and referred to as 'global' and 'local', respectively by Alpert et al.[7]. The datasets include 179 PAM-stimulated cells and 290 LPS-stimulated cells. CellAlign[7] used 200 interpolation points for their PAM/LPS analysis, whereas the optBinning[68] structure suggests that 14 points are sufficient.

**Simulated dataset containing trajectories with no shared process.** This is a simulated negative control, generated using the published script by Laidlaw et al.[17], which uses DynGen[72], a single-cell data simulator for dynamic processes. This dataset contains two trajectories simulated under two different gene regulatory networks and TF activity, ensuring no shared process between them. The reference and query have 619 genes across 2,000 and 1,940 cells, respectively.

### Dataset for healthy versus IPF case study
We downloaded healthy and IPF datasets[21] from the Gene Expression Omnibus (GEO) (GSE136831) and extracted the lineages AT2 to AT1 cell differentiation for healthy; AT2 to ABC differentiation for IPF (based on original author annotations). We identified a subset of 54 AT2 cells of low quality (with high percentage of ribosomal gene expression) that were filtered out before further analysis. Next, we subsetted healthy and IPF cells to independently preprocess and estimate their pseudotime. This includes 3,157 healthy cells (2,655 AT2 and 502 AT1) from 28 individuals and 890 IPF cells (442 AT2 and 448 ABCs) from 31 individuals. We first normalized their per-cell total transcript count to 10,000 followed by log1p transformation and selected 4,000 HVGs to estimate cell pseudotime using Diffusion Pseudotime[35] implemented in SCANPY[73] (v.1.9.6) with default parameter settings. The root cell for each trajectory was chosen based on the expression score of known AT2 progenitor cell markers: *AXIN2*, *FGFR2*, *ID2*, *FZD6*, *LRP5* and *LRP6* (refs. [74],[75]) (using scanpy.tl.score_genes). ABC-specific marker genes were obtained from the original paper's supplementary file (aba1983_data_s2.txt)[21].

### Dataset preparation for in vivo and in vitro T cell development comparison
**Cell cultures for ATOs and scRNA-seq experiment.** The MS5 line transduced with human DLL4 was obtained from G. Crooks (UCLA) as a gift. The MS5-hDLL4 cells were cultured in DMEM (Gibco, 41966) with 10% FBS (Gibco, 16000044). Two iPS cell lines were used in this study. Cell lines HPSI0114i-kolf_2 (Kolf) and HPSI0514i-fiaj_1 (Fiaj) were obtained from the Human Induced Pluripotent Stem Cell initiative (HipSci: www.hipsci.org) collection. All iPS cell lines were cultured on vitronectin-coated (diluted 1:25 in PBS; Gibco, A14700) plates, in TeSR-E8 medium (STEMCELL Technologies, 05990).

We followed the PSC-ATO protocol as previously described[40]. iPS cells were collected as a single-cell suspension and seeded ($3 \times 10^6$ cells per well) in GFR-reduced Matrigel-coated (Corning, 356231) six-well plates in X-VIVO 15 medium (Lonza, 04-418Q), supplemented with rhActivin A, rhBMP4, rhVEGF, rhFGF (R&D Systems, 338-AC-010/314-BP-010/298-VS-005/233-FB-010) and ROCK inhibitor (Y27632; LKT Labs, Y1000) on day −17 and only rhBMP4, rhVEGF and rhFGF on days −16 and −15. Cells were collected 3.5 days later (day −14) and isolated by fluorescence-activated cell sorting (FACS) for CD326⁻CD56⁺ (PE anti-human CD326 antibody, BioLegend, 324205; APC anti-human CD56 antibody, BioLegend, 318309) human embryonic mesodermal progenitors (hEMPs). Representative FACS plots are shown in Supplementary Fig. 7a.

Isolated hEMPs were combined with MS5-hDLL4 at a ratio of 1:50. Two or three cell-dense droplets ($5 \times 10^5$ cells in 6 µl hematopoietic induction medium) were deposited on top of an insert in each well of a six-well plate. Hematopoietic induction medium composed of EGM2 (Lonza, CC-3162) supplemented with ROCK inhibitor and SB blocker (TGF-β receptor kinase inhibitor SB-431542; Abcam, ab120163) was added into the wells outside the inserts so that the cells sat at the air–liquid interface. The organoids were then cultured in EGM2 with SB blocker for 7 days (days −14 to −7), before the addition of cytokines rhSCF, rhFLT3L and rhTPO (Peprotech, 300-07/300-19/300-18) between days −6 and 0. These 2 weeks formed the hematopoietic induction phase. On day 1, the medium was changed again to RB27 (RPMI (Corning, 10-040-CV) supplemented with B27 (Gibco, 17504-044), ascorbic acid (Sigma-Aldrich, A8960-5G), penicillin–streptomycin (Sigma-Aldrich, P4333) and glutaMAX (Thermo Fisher Scientific, 35050061)) with rhSCF, rhFLT3L and rhIL7 (Peprotech, 200-07). The organoids can be maintained in culture for 7 more weeks in this medium.

For dissociation of organoids on day −7, they were removed from culture insert and incubated in digestion buffer, which consisted of collagenase type IV solution (STEMCELL Technologies, 07909) supplemented with 0.88 mg ml⁻¹ collagenase/dispase (Roche, 10269638001) and 50 U DNase I (Sigma, 9003-98-9 D5025-15KU), for 20 min at 37 °C. Vigorous pipetting was performed in the middle of the incubation and at the end. After complete disaggregation, a single-cell suspension was prepared by passing through a 50-µm strainer.

For dissociation of organoids from day 0 onwards, a cell scraper was used to detach ATOs from cell culture insert membranes and detached ATOs were then submerged in cold flow buffer (PBS (Gibco, 14190144) containing 2% (*v/v*) FBS and 2 mM EDTA (Invitrogen, 15575020)). Culture inserts were washed and detached ATOs were pipetted up and down to form a single-cell suspension before passing through a 50-µm strainer.

Cells were then stained with designed panels of antibodies and analyzed by flow cytometry. FACS was performed at the same time and live human 4,6-diamidino-2-phenylindole (DAPI)⁻ anti-mouse CD29⁻ (Invitrogen, D1306; APC/Cy7 anti-mouse CD29 antibody, BioLegend, 102225) cells were sorted for day −7, day 0 and week 3 samples, and live (DAPI⁻) cells were sorted for week 5 and week 7 samples before loading onto each channel of a Chromium chip from a Chromium single-cell V(D)J kit (10X Genomics). Representative FACS plots are shown in Supplementary Fig. 7b. The metadata for all the ATO samples can be found in Supplementary Table 8. For the day −14 sample, some sorted (both hEMP and the rest of the DAPI⁻ fraction) and unsorted cells were stained with hashtag antibodies (TotalSeq-C antibodies from BioLegend (Supplementary Table 9), following a 10x cell surface protein-labeling protocol) before being mixed together with some mouse stromal cells (MS5-hDLL4) for 10x loading. For week 1 sample, hashtag antibodies were added at the same time as the FACS antibodies, before sorting. All antibodies used were added as 2 µl per antibody in a total of 100 µl staining solution.

Single-cell complementary DNA synthesis, amplification and gene expression (GEX) and cell surface protein (CITE-seq) libraries were generated following the manufacturer's instructions. Sequencing was performed on the Illumina Novaseq 6000 system. The gene expression libraries were sequenced at a target depth of 50,000 reads per cell using the following parameters: Read1: 26 cycles; i7, 8 cycles; i5, 0 cycles; Read2: 91 cycles to generate 75-bp paired-end reads.

**ATO data preprocessing and annotation.** Raw scRNA-seq reads were mapped using Cell Ranger (v.3.0.2) with combined human reference of GRCh38.93 and mouse reference of mm10-3.1.0. Low-quality cells were filtered out (minimum number of reads of 2,000, minimum number of genes of 500, maximum number of genes of 7,000; Scrublet[76] (v.0.2.3) doublet detection score <0.15 and mitochondrial reads fraction <0.2). Cells where the percentage of counts from human genes was <90% were considered as mouse cells and excluded from downstream analysis. Cells were assigned to different cell lines (Kolf or Fiaj) using genotype prediction with Souporcell (v.2.4.0)[77]. The mapping outputs of the eight samples were merged, with sample ID prepended to the barcode IDs in both the BAM and barcodes.tsv to prevent erroneous

cross-sample barcode overlap. Souporcell was run with `-skip_remap True --K 2` and the common variants file based on common (≥2% population allele frequency) single-nucleotide polymorphisms (SNPs) from 1000 Genomes Project data, as distributed in the tool's repository. We selected two clusters due to the already known two cell lines. Next the data went through the standard pipeline of filtering out genes (cell cycle[78] genes, genes detected in <3 cells) and normalizing the per-cell total count to 10,000 followed by log1p transformation and scaling to zero mean and unit variance (with max_value = 10 to clip after scaling), using SCANPY[73]. The final dataset had 31,483 ATO cells with 23,526 genes, which were input to CellTypist[41] (v.0.1.4) (for annotation prediction using pretrained logistic regression classifier -Pan_Fetal_Human model under majority voting). We then obtained the Uniform Manifold Approximation and Projection (UMAP) embedding for this dataset based on its scVI[42] (v.0.14.5) batch-corrected embedding (with ten latent dimensions, two hidden layers, 128 nodes per hidden layer and 0.2 dropout rate for the neural network) and subsetted cells to non-hematopoietic lineage, T/ILC/NK lineage and other hematopoietic lineage cells (Extended Data Fig. 8) using Leiden clustering. For each lineage, scVI and UMAP embeddings were re-computed and cell types were annotated (low-level annotations in Extended Data Fig. 6b, with more refined annotations in Extended Data Fig. 7a) by inspecting both CellTypist predictions (Extended Data Fig. 7b) and marker gene expression (Extended Data Fig. 8).

**Joint embedding of reference and organoid for pseudotime estimation.** We downloaded the annotated human fetal atlas dataset from https://developmental.cellatlas.io/fetal-immune and extracted cell types (79,535 cells in total) representing the T cell developmental trajectory from progenitor cells toward type 1 innate T cells (T1 dataset), including cycling MPP, HSC_MPP, LMPP_MLP, DN(early) T, DN(P) T, DN(Q) T, DP(P) T, DP(Q) T, ABT(entry) and type 1 innate T cells. We then compiled a reduced representation (20,384 cells), preserving their underlying cell-type composition by random subsampling from each cell type (with minimum sample size of 500 cells, aiming for ~20,000 total number of cells) based on their annotations. Such stratified-sampling is practical for dealing with massive single-cell datasets to reduce resource demands. Separately, we extracted cell types from the ATO dataset (19,013 cells) representing the trajectory from iPS cells toward SP T cells, including iPS cells, primitive streak, mesodermal progenitor, endothelium, HSC_MPP, HSC_MPP/LMPP_MLP/DC2, DN(early) T, DN T, DP(P) T, DP(Q) T, ABT(entry) and SP T cells.

T1 and ATO datasets were merged and preprocessed together by filtering out cells with more than 8% total mitochondrial unique molecular identifiers, cell cycle genes[78] and genes expressed in <3 cells. Next, HVGs were selected after normalizing per-cell count to 10,000 reads and log1p transformation. T1 pan fetal reference had 33 batches (due to different 10x chemistry 3′ versus 5′ and donors), whereas the ATOs had two batches (due to two cell lines, Kolf and Fiaj). We constructed a batch-corrected, scVI latent embedding (ten latent dimensions, two hidden layers, 128 nodes per hidden layer and 0.2 dropout rate for the neural network) of the merged dataset. This was taken to build the cell neighborhood graph and UMAP embedding using SCANPY[73]. The final T1 and ATO datasets comprised 20,327 cells and 17,176 cells respectively. A total of 18,436 cells of T1 and 10,089 cells of ATO belong to the T cell lineage (DN T onwards).

We followed a similar preprocessing for the pan fetal reference representing the trajectory toward CD8+ T (CD8+ dataset) (including cycling MPP, HSC_MPP, LMPP_MLP, DN(early) T, DN(P) T, DN(Q) T, DP(P) T, DP(Q) T, ABT(entry) and CD8+ T cells). The initially extracted CD8+ subset (83,177 cells) was reduced to 20,412 cells, which was then merged with the 19,013 ATO cells and subjected to the same filtering and normalization carried out for the T1 + ATO merge before scVI integration. The final CD8+ dataset comprised 20,324 cells, of which 18,490 cells were DN T onwards.

**Pseudotime estimation using GPLVM.** Differentiation pseudotime was estimated separately for T1 reference, CD8+ reference and ATO by employing GPLVM[43,79] (a probabilistic nonlinear dimensionality reduction method that models gene expression as a function $f(X)$ of a set of latent covariates X). It enables incorporating time priors when estimating pseudotime as a latent dimension. GPLVM has previously been successful in single-cell trajectory inference to incorporate useful priors[79–83]. We used the Pyro[84] (v.1.8.0) GPLVM implementation with sparse GP inference (32 inducing points), RBF and Adam optimizer, to obtain an optimal two-dimensional latent embedding, where the two dimensions correspond to pseudotime and a second level of latent effects (for example batch), respectively. The pseudotime was assigned a Gaussian prior with cell sampling times as the mean. The second dimension was zero-initialized. GPLVM loss curve reasonably converged in 2,000 iterations at optimization.

For the ATO data, GPLVM was initialized with cell sampling (capture) days as the prior. As there was no temporal data for the pan fetal reference, we first approximated time prior for each reference cell as the weighted average of their k-nearest organoid neighborhood (kNN) capture time. A k = 3 organoid neighborhood for a reference cell was obtained using the cKDTree-based search method implemented in BBKNN[85] (v.1.5.1) on their scVI-based UMAP embedding. Contribution of each organoid neighbor was weighted according to its distance (the kNN distance vector was softmax-transformed and the normalized reciprocal of each distance was taken as the associated weight, enforcing less contribution from distant neighbors toward the weighted average). This approximation may introduce outliers due to the spatial arrangement of different cell types in the UMAP. Thus, we leveraged the known cell-type annotations to refine the approximation by assigning each reference cell with the average approximated capture time of its cell type. These approximated capture times were scaled to be in [0,1] range and input as the mean of the Gaussian prior of pseudotime to the previously described GPLVM. For T1 and CD8+ GPLVMs, the input gene space was 2,608 genes and 2,616 genes, respectively (the same as at scVI integration). To ensure no outliers, the GPLVM estimated pseudotime was further refined by correcting outliers of each cell type using the cell-type specific average of estimated pseudotime. Outliers were selected based on the interquartile range (IQR) rule (1.5 × IQR below the first quartile and above the third quartile of the cell-type-specific pseudotime distribution).

## G2G alignment

For the complete T1 versus ATO comparison using G2G, the total common gene space of 20,240 genes was considered upon filtering genes with <3 cells expressed, 10,000 total count per-cell normalization and log1p transformation. For the DN T onwards comparison, there were 17,718 genes for T1 versus ATO and 20,183 genes for CD8+ versus ATO. These total gene spaces were subsetted to include only the transcription factors[44] (1,371 TFs) and relevant signaling pathways focused in this work. G2G alignment was performed using 14 equispaced pseudotime points.

**High-dimensional alignment using CellAlign and TrAGEDy[1].** To compare G2G alignment of T1 versus ATO against CellAlign and TrAGEDy alignments, we followed the same steps and parameter settings of those tools as described in the previous section 'Benchmarking against CellAlign[7] and TrAGEDy[17] alignment'. The alignment of 14 equispaced interpolation pseudotime points across the 1,371 TFs gave a three-state alignment string (MVVVVVVMWMWWWVWWWMVVVWW) from CellAlign and a five-state alignment string (IIIIIIMMMWWWWMWWMIDIDID) from TrAGEDy.

## ATO TNF preliminary validation experiment

For the TNF validation experiment, we followed the PSC-ATO protocol described previously using Kolf iPS cell line. One plate (12 organoids) was set for each condition, with four TNF (R&D Systems, 210-TA)

conditions: control (no TNF addition), 1 ng ml$^{-1}$ (final concentration), 5 ng ml$^{-1}$ and 25 ng ml$^{-1}$. TNF was added into the medium between weeks 6–7. Organoids were collected at week 7, stained with hashtag antibodies (TotalSeq-C antibodies from BioLegend) and a designed panel of FACS antibodies and sorted for DAPI$^{-}$CD45$^{+}$ cells before 10x loading. Representative FACS plots are shown in Supplementary Fig. 7c. Library construction, sequencing and data preprocessing was the same as the WT ATO organoids.

### Assessing T cells from ATO before and ATO after TNF treatment

The ATOs with TNF treatment (ATO$_{TNF}$) resulted in 123 good quality T cells (data preprocessing and filtering as above; annotated by Cell-Typist[41] as type 1 innate T cells) where the majority (~94% cells) were 1 ng ml$^{-1}$ TNF-treated (as opposed to wild-type, 5 ng ml$^{-1}$ and 25 ng ml$^{-1}$). Thus, we performed the analysis using T cells treated with 1 ng ml$^{-1}$ TNF (116 cells) that were predicted to be type 1 innate T cells by CellTypist[41].

The wild-type ATOs without TNF treatment have 6,558 SP T cells and the pan fetal reference[39] has 1,413 type 1 innate T cells. To check whether there is an improvement in the maturity of resultant T cells in ATO$_{TNF}$, we evaluated and compared the degree of similarity between ATO$_{TNF}$ and the reference against that between ATOs and the reference. This assessment was conducted via two directions: cell level and gene level. For cell-level assessment, we constructed an scVI[42] latent embedding by integrating the reference, ATOs and ATO$_{TNF}$ together, following the same preprocessing steps carried out for the ATOs and the reference previously. We then computed the Euclidean distance between the mean latent dimensions of in vitro SP T cells and in vivo type 1 innate T cells. For gene-level assessment, we computed the MML distance of gene expression distributions by first identifying significantly distant genes across all the TFs, as well as across several curated pathway gene subsets from KEGG and MSigDB that are associated with TNF signaling (Supplementary Note), before and after TNF treatment. We next tested whether the difference between those distance distributions was significant or not using the Mann–Whitney $U$-test. The following section describes our approach to identifying genes with a statistically significant distance.

### Estimating an empirical null distribution of MML distances to assess significance of a distance

We expect no significant difference in gene expression between any two random subsets of cells belonging to the same cell type in the same system. Thus, we generated such random pairs of cell subsets within our pan fetal reference (type 1 innate T cell) dataset as well as within the ATO (SP T cell) dataset across all 1,371 TFs, to construct an empirical null distribution for MML distance. This enables us to compute a $P$ value for any given MML distance $x$, indicating how likely $x$ is to occur by chance.

For each TF, we performed uniform random sampling of a subset pair without replacement for 50 iterations, resulting in 137,100 total number of pairs. Each subset contains 50 cells. Then, for each pair, we computed the MML distance. All computed MML distances together form a null distribution which we used to evaluate significance of a given MML distance between two expression distributions.

The resultant empirical distribution of MML distances is highly left-skewed. We constructed its empirical cumulative density function (CDF) using distributions.empirical_distribution.ECDF function in statsmodels (v.0.13.5) to test significance in distances under no assumption about the family of distribution. The $P$ values were adjusted for multiple testing using the Benjamini–Hochberg method, before identifying significantly distant genes at a given level of significance.

### Testing statistical significance of the change in gene expression distances to reference from wild-type ATOs versus ATO$_{TNF}$

Given the MML distances of the genes that are significantly different: (1) between the reference and ATOs (sample 1); and (2) between the reference and ATO$_{TNF}$ (sample 2), we tested whether the average gene expression distance dropped after TNF treatment. To test the significance of the change in distance, we performed a nonparametric Mann–Whitney $U$-test implemented in scipy.stats.mannwhitneyu.

### Testing the closeness of SP T cells in the ATO to the reference type 1 innate T cells before and after TNF treatment

We compared the 6,558 SP T cells in ATO and the 116 T cells in ATO$_{TNF}$, against the 1,413 type 1 innate T cells from the pan fetal reference. This was carried out for all 1,371 human TFs, as well as the 196 genes in TNF signaling via the NF-κB pathway (from MSigDb).

For each gene set, we first computed the MML distances of the genes between (1) reference and ATOs; and (2) reference and ATO$_{TNF}$. We then computed a $P$ value for each MML distance under the empirical CDF estimated by computing MML distances between randomly sampled homogeneous subsets across the reference and ATO cells. This allowed us to identify the gene set that was significantly different between reference and ATOs (diffset$_{UNTREATED}$) and the gene set that was significantly different between reference and ATO$_{TNF}$ (diffset$_{TREATED}$). Next, we compared the average distance of genes in diffset$_{UNTREATED}$ to the average distance of genes in diffset$_{TREATED}$, while testing the significance of the change in the distances using Mann–Whitney $U$-test.

### Software and computational requirements

The G2G framework and all analyses were implemented in Python (≥v.3.8) with commonly used scientific-computing and visualization libraries and the specific libraries mentioned in the Methods. G2G was tested on two operating systems: Ubuntu 20.04.1 LTS and Debian GNU/Linux 10 (buster). The G2G runtime depends on the number of cells in the reference and query datasets, the number of interpolation time points and the number of genes to align. G2G (v.0.1.0) used in this work is a less-efficient version that utilizes concurrency through Python multiprocessing to speed up the gene-level alignment process. It creates a number of processes equal to the number of cores in the system and each process performs a single gene-level alignment at one time. Per-gene runtime for a case aligning 100 genes between 20,327 reference cells and 17,176 query cells (pan fetal reference and ATO datasets), with 14 interpolation points and 16 cores in a Linux (SMP Debian 4.19.304-1 (2024-01-09)) system, was approximately 5.57 s. A notably faster, latest Genes2Genes v.0.2.0 is now available, running sequentially with 0.60 s per-gene for the same case (SMP Debian 5.10.216-1 (2024-05-03)).

### Reporting summary

Further information on research design is available in the Nature Portfolio Reporting Summary linked to this article.

## Data availability

Data used to perform analyses in the manuscript are available at https://zenodo.org/records/11182400 (ref. 86) and https://github.com/Teichlab/G2G_notebooks. All generated alignments are available as Supplementary Data. Raw sequencing data for newly generated sequencing libraries have been deposited in ArrayExpress (accession no. E-MTAB-12720).

## Code availability

G2G is an open-source Python package with a tutorial available at https://github.com/Teichlab/Genes2Genes. Code used to perform analyses in the manuscript using G2G v.0.1.0 is available at https://github.com/Teichlab/G2G_notebooks.

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

## Acknowledgements

This paper is dedicated to the memory of our dear friend and colleague D. Muraro who contributed to this work. We thank the Crooks laboratory (A. Montel-Hagen, S. Lopez and G. Crooks from University of California, Los Angeles) for their kind help in setting up the ATO experiments; N. Huang, L. Dratva, R. Lindeboom, R. Elmentaite and A. Maartens from the Wellcome Sanger Institute, C.H. Ek from the University of Cambridge, Z. Miao from Guangzhou Laboratory, Y. Chen and T. Wang from the London School of Economics and Political Science for their helpful discussions. We used BioRender.com and Adobe Illustrator 2023 for graphical illustrations. We gratefully acknowledge the Sanger Flow Cytometry Facility and Sanger Core Sequencing pipeline for support with sample processing and sequencing library preparation. We acknowledge the Wellcome Sanger Institute as the source of HPSI0114i-kolf_2 and HPSI0514i-fiaj_1 human iPS cell lines, which were generated under the Human Induced Pluripotent Stem Cell Initiative funded by a grant from the Wellcome Trust and Medical Research Council, supported by the Wellcome Trust (WT098051) and the NIHR/Wellcome Trust Clinical Research Facility and acknowledge Life Science Technologies Corporation as the provider of Cytotune. This publication is part of the Human Cell Atlas: www.humancellatlas.org/publications. This project has received funding from the European Union's Horizon 2020 research and innovation programme under the Marie Skłodowska-Curie grant agreement no. 101026506. D.S. is supported by the Marie Curie Individual Fellowship; C.S. is supported by a Wellcome Trust PhD Fellowship for Clinicians. S.A.T. is funded by the Wellcome Trust (WT206194), the ERC Consolidator grant ThDEFINE (646794), Wellcome Trust (203151/Z/16/Z, 203151/A/16/Z) and the UKRI Medical Research Council (MC_PC_17230). This work was also supported by a grant from the Wellcome Sanger Institute's Translation Committee Fund.

## Author contributions

D.S., C.S. and S.A.T. conceived the initial project. D.S. and C.S. set up and directed the study. D.S., C.S., A.C, E.D. and K.P. performed bioinformatic analyses. D.S. designed and developed the software. C.S., A.S.S., W.L. and J.P. performed cell culture experiments. D.M., E.D., A.J.O., K.B.M., B.D. and S.A.T. provided intellectual input. S.A.T. acquired funding. D.S., C.S. and A.C. wrote the manuscript. All authors read and/or edited the manuscript.

## Competing interests

S.A.T. is a scientific advisory board member of ForeSite Labs, OMass Therapeutics, QIAGEN, a co-founder and equity holder of TransitionBio and EnsoCell Therapeutics, a nonexecutive director of 10x Genomics and a part-time employee of GlaxoSmithKline. The remaining authors declare no competing interests.

## Additional information

**Extended data** is available for this paper at https://doi.org/10.1038/s41592-024-02378-4.

**Correspondence and requests for materials** should be addressed to Sarah A. Teichmann.

Primary Handling Editor: Madhura Mukhopadhyay, in collaboration with the *Nature Methods* team. Peer reviewer reports are available.

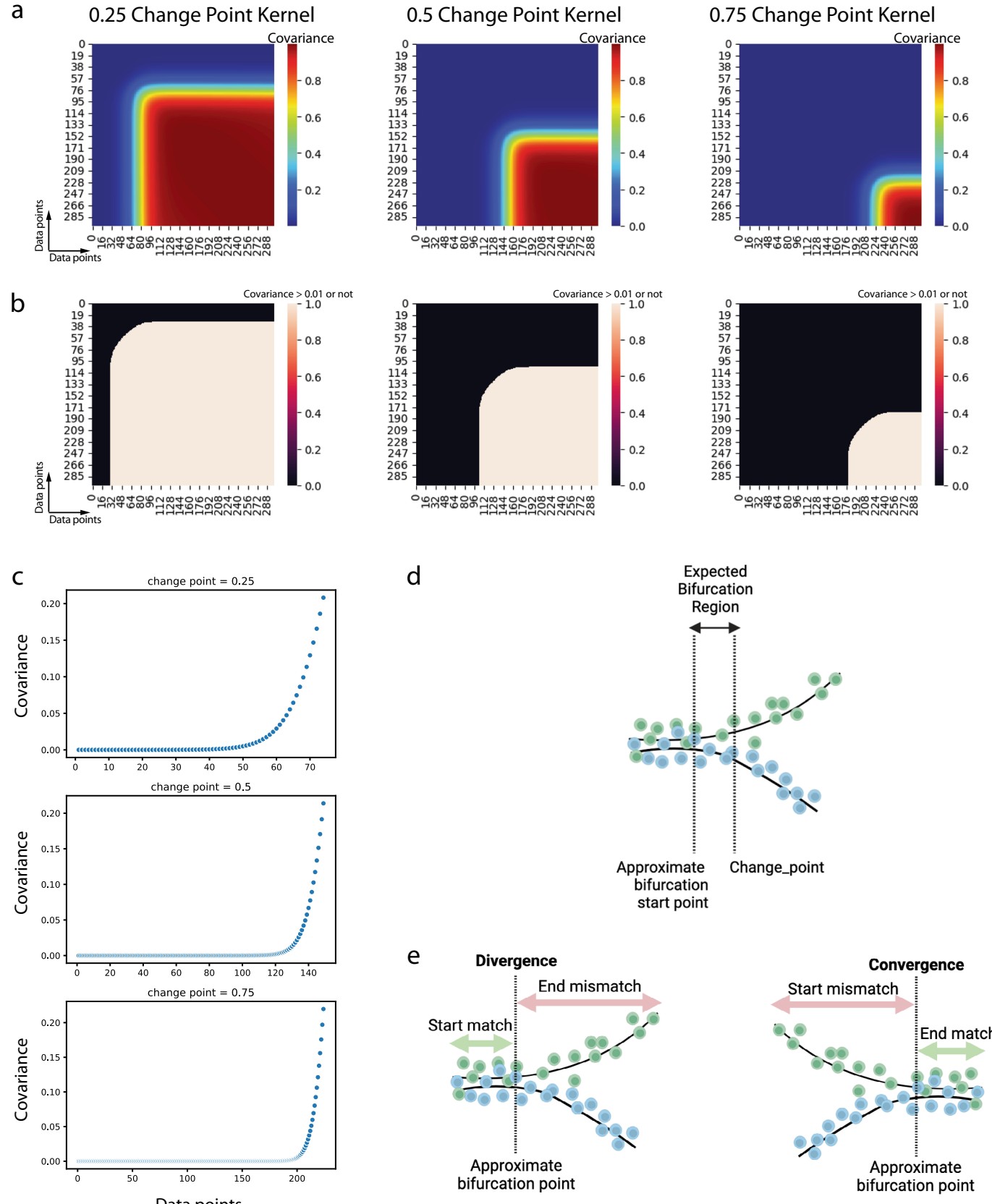

Extended Data Fig. 1 | See next page for caption.

**Extended Data Fig. 1 | Simulating the bifurcation of reference and query trajectories using change point kernels.** (Note: A change point kernel defines shifts and changes in covariance between discrete time points in a time series that describes a particular Gaussian process. In the context of a single-cell pseudotime trajectory, each discrete time point corresponds to a single cell. The change point kernel can be represented by a pairwise covariance matrix between those time points, visualized using heatmaps). **a**, Change point kernel heatmaps for each approximate bifurcation point (change point) $\in [0.25, 0.5, 0.75]$. **b**, The same change point kernels binarized based on the 0.01 covariance threshold (top), **c**, The average covariance plotted for each $i \times i$ sub square matrix from $i = 0$ to $i =$ change point, showing that the branching effect can approximately start before the specified change point. **d**, Expected bifurcation region is taken from the point where we begin to see > 0.01 covariance in the change point kernel, until the particular change point. **e**, Illustration of the main regions of match and mismatch expected in trajectory alignment under *Divergence* class (left) and *Convergence* class (right). A *Divergence* alignment is described by a start-match region followed by an end-mismatch region, whereas a *Convergence* alignment is described by a start-mismatch region followed by an end-match region. Illustrations in **d-e** were created using BioRender (https://biorender.com).

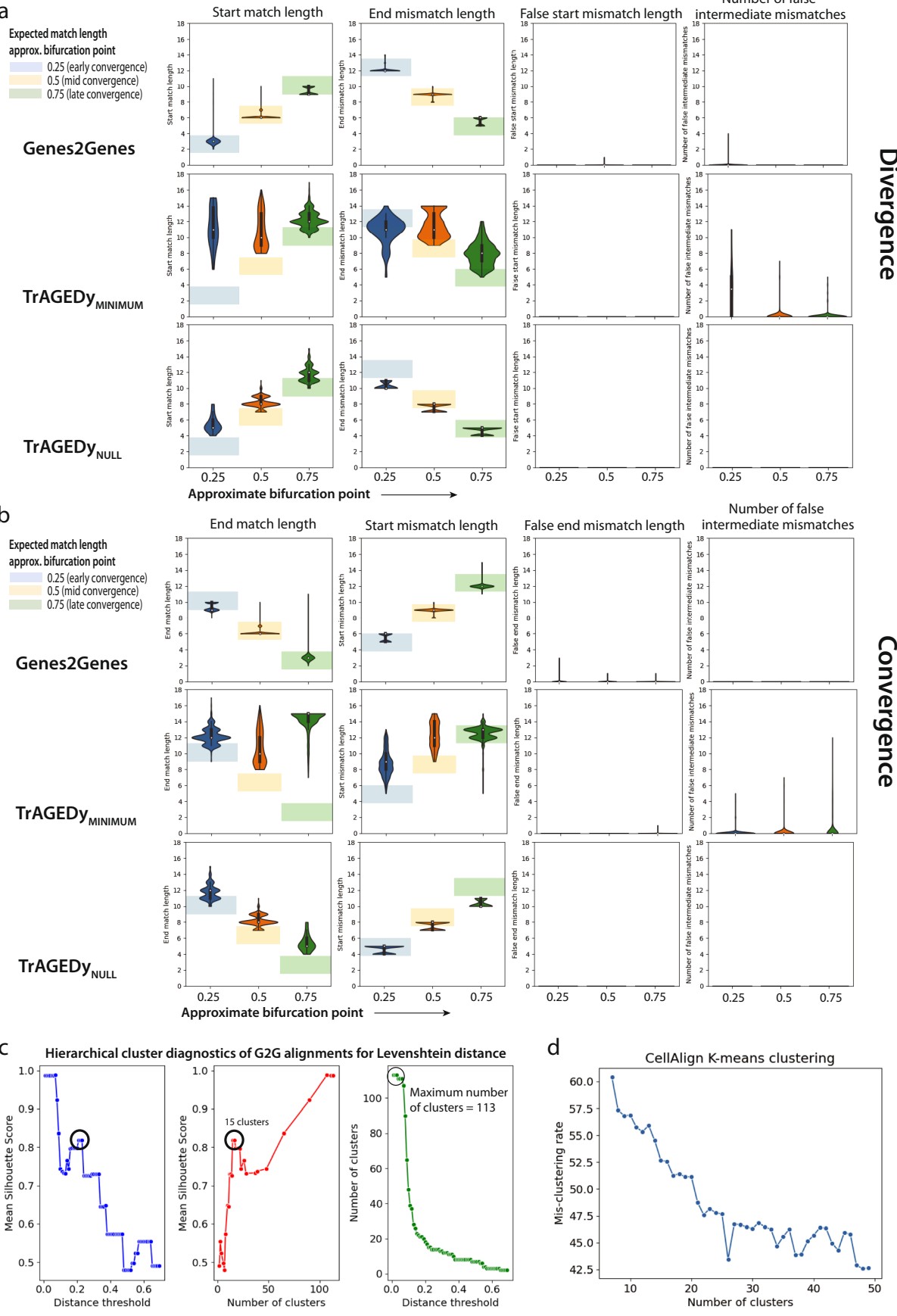

**Extended Data Fig. 2 | See next page for caption.**

**Extended Data Fig. 2 | Simulation Data Experiment 1. a**, Distributions of start-match lengths (following a false mismatch if any), end-mismatch lengths (prior to a false match if any), start-mismatch (false mismatch) lengths (number of false mismatches starting from time point 0), and the number of intermediate false mismatches within the match regions, in the 1500 *Divergence* alignments across the three bifurcation subgroups (that is, under approximate bifurcation point [0.25,0.5,0.75]; each subgroup has 500 alignments), generated by Genes2Genes, TrAGEDy$_{MINIMUM}$, and TrAGEDy$_{NULL}$. 15 equispaced time points over pseudotime [0,1] were used for distribution interpolation and alignment. Colored boxes (in blue, orange, and green) in the two leftmost columns display possible ranges of expected match lengths corresponding to the three different, approximate bifurcation points: [0.25,0.5,0.75], respectively. Each violin plot shows the length distribution across all n = 500 alignments in each group as a kernel density estimation. Inside the violin is a box showing the interquartile range (covering the 25% and 75% quantiles with a point indicating median). **b**, Similar statistics as

in **a**, reported for the 1500 *Convergence* alignments across the three bifurcation subgroups, generated by Genes2Genes, TrAGEDy$_{MINIMUM}$, and TrAGEDy$_{NULL}$. Distributions of end-match lengths (prior to a false mismatch if there is any), start-mismatch lengths (following a false match if there is any), end-mismatch lengths (number of false mismatches until time point 1), and the number of intermediate false mismatches within the match regions. **c**, Cluster diagnostic plots for the hierarchical agglomerative clustering of the 3500 alignments across all pattern classes (including 500 *Matching* alignments, *1500 Divergence* alignments, *1500 Convergence* alignments), in terms of the mean Silhouette score when varying the Levenshtein distance threshold (or the number of clusters). The highest number of clusters represent the number of all unique 5-state alignment strings (that is 113 strings). Bold highlighted circles mark the local optimal mean Silhouette score which gives 15 optimal clusters for the genes at 0.22 distance threshold. **d**, Mis-clustering rates of the CellAlign k-means clustering outputs for all 3500 alignments, versus the number of clusters (k) ranging from k = 7 to k = 50.

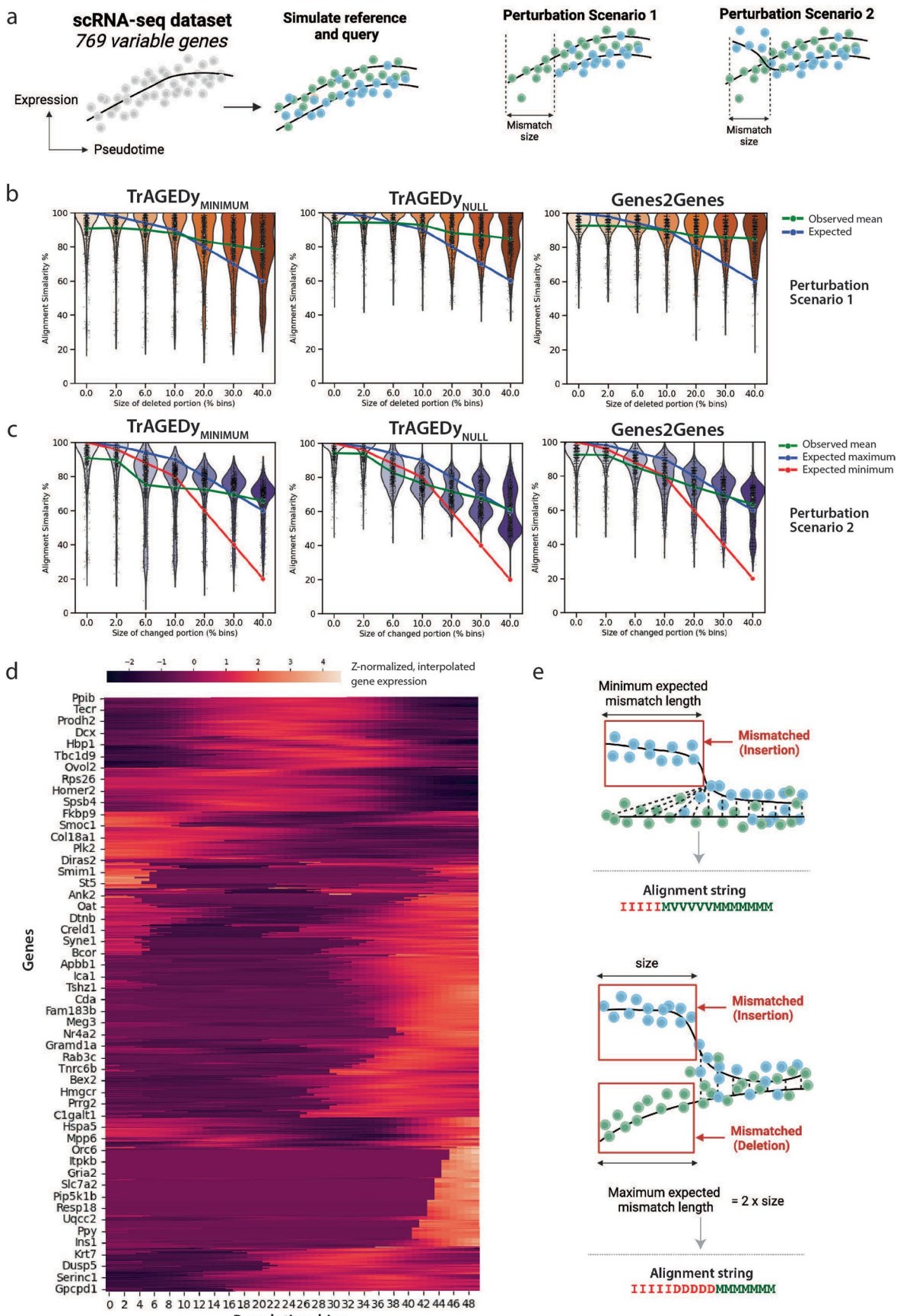

**Extended Data Fig. 3 | See next page for caption.**

**Extended Data Fig. 3 | Simulation Data Experiment 2. a,** Experiment 2 uses 769 genes in the mouse pancreas development (Beta lineage) scRNA-seq dataset[29] to generate perturbed pairs of alignment from the expected *Matching* alignments. Perturbation scenario 1 deletes the start region from the reference trajectory, whereas perturbation scenario 2 changes the start region of the query trajectory. **b,** Alignment similarity distributions for varying sizes of perturbation (perturbed percentage of the 50 pseudotime interpolation points) under perturbation scenario 1, resulted from gene-level alignment using TrAGEDy$_{MINIMUM}$, TrAGEDy$_{NULL}$, and Genes2Genes. Each point in the violin plot represents a gene (total number of genes n = 769). In each plot, the observed average alignment similarity across different perturbation sizes is shown by the green line. Blue line shows the expected alignment similarity across different perturbation sizes. Each violin plot shows the distribution of alignment similarities across all gene alignments in each group as a kernel density estimation. Inside the violin is a box showing the interquartile range (covering the 25% and 75% quantiles with a point

indicating median). **c,** The alignment similarity distributions for varying sizes of perturbation under perturbation scenario 2 similar to **b**. There are two expected lines: maximum (in blue) and minimum (in red). The maximum mismatch length is expected when both reference and query time points form insertions and deletions, making the maximum expected length *size*\*2. The minimum mismatch length is expected when only the changed reference time points are mismatched as insertions, while the corresponding query time points are matched to the non-perturbed reference time points (illustrated in **e**). **d,** Overall smoothened (interpolated) and z-normalized mean gene expression along pseudotime (across 50 equispaced interpolation time points) for all genes in the dataset. **e,** Example illustrations of the two types of trajectory alignment that gives the minimum and maximum expected mismatch lengths under the perturbation scenario 2, where a start portion of a particular size in the query trajectory (in blue) is changed with respect to the reference trajectory (in green). Illustrations in **a,e** were created using BioRender (https://biorender.com).

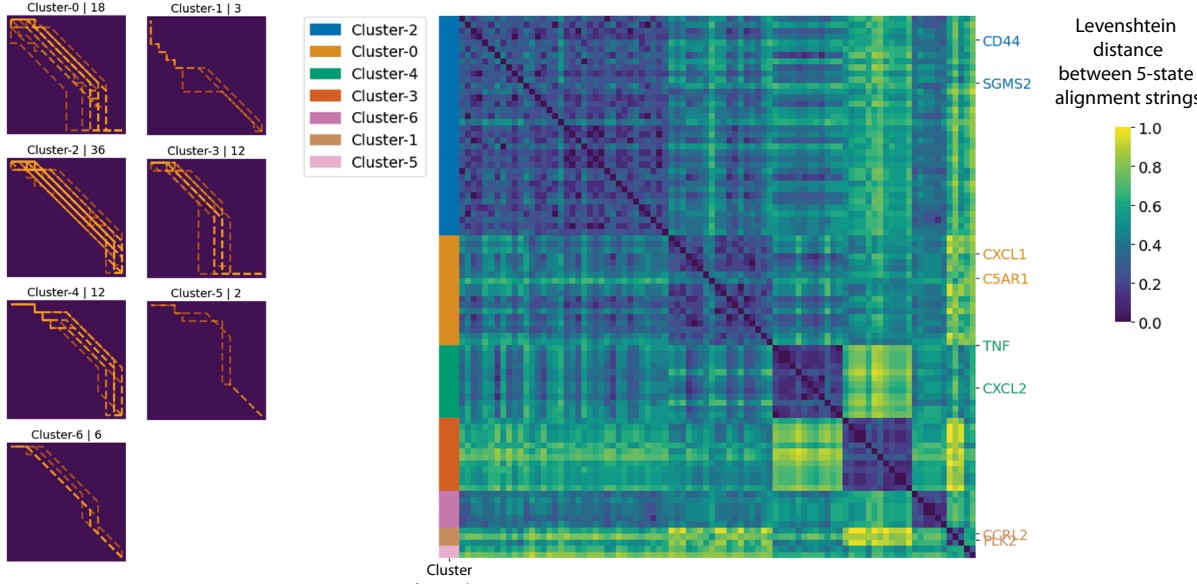

**a** Core antiviral genes (99 genes)

**b** Hierarchical cluster diagnostics for Levenshtein distance

Cluster-0 | 96    Cluster-1 | 3

**c** Peaked inflammatory genes (89 genes)

**d** Hierarchical cluster diagnostics for Levenshtein distance

**e** High resolution clustering structure (distance threshold = 0.37)

**Extended Data Fig. 4 | See next page for caption.**

**Extended Data Fig. 4 | The PAM vs. LPS alignment using Genes2Genes. a**, Density plot of the alignment similarity distribution (that is distribution of the percentage of matches/warps across all the alignment outputs) for the 99 genes in the 'core antiviral module'. **b**, Top: Cluster diagnostic plots for the hierarchical agglomerative clustering of those 99 gene alignments in terms of the mean Silhouette score when varying the Levenshtein distance threshold (or the number of clusters). The highest number of clusters represent the number of all unique 5-state alignment strings (that is 48 strings). Bold highlighted circles mark the local optimal mean Silhouette score which gives two optimal clusters for the genes at 0.4 distance threshold. Bottom: Each plot titled by "Cluster-x | n" is the pairwise matrix of reference time points (columns) and query time points (rows), visualizing alignment paths for the total of n genes in cluster x. **c**, Density plot of the alignment similarity distribution (that is distribution of the percentage of matches/warps across all the alignment outputs) for the 89 genes in the 'peaked inflammatory module'. **d**, Cluster diagnostic plots for the hierarchical agglomerative clustering of those 89 gene alignments, reported similarly to **b**. The identified optimal clustering structure has 7 clusters (at distance threshold=0.37 corresponding to the local optimal mean Silhouette score, highlighted by bold circles). **e**, The clustermap of the pairwise Levenshtein distance matrix of all 89 gene alignments, which illustrates the identified 7 clusters.

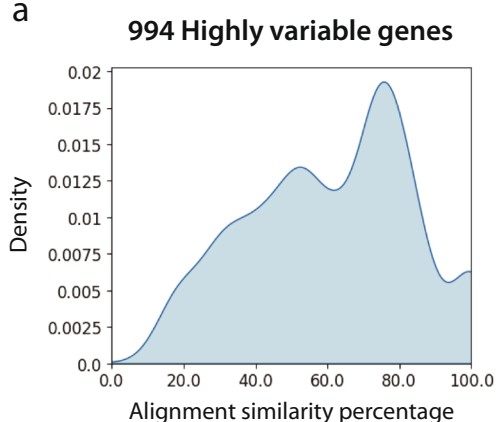

**a** 994 Highly variable genes

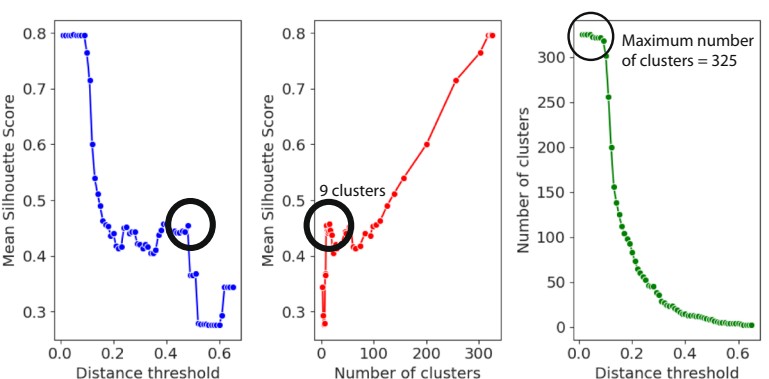

**b** Hierarchical cluster diagnostics for Levenshtein distance

**c** **Clustering structure** (distance threshold = 0.48)

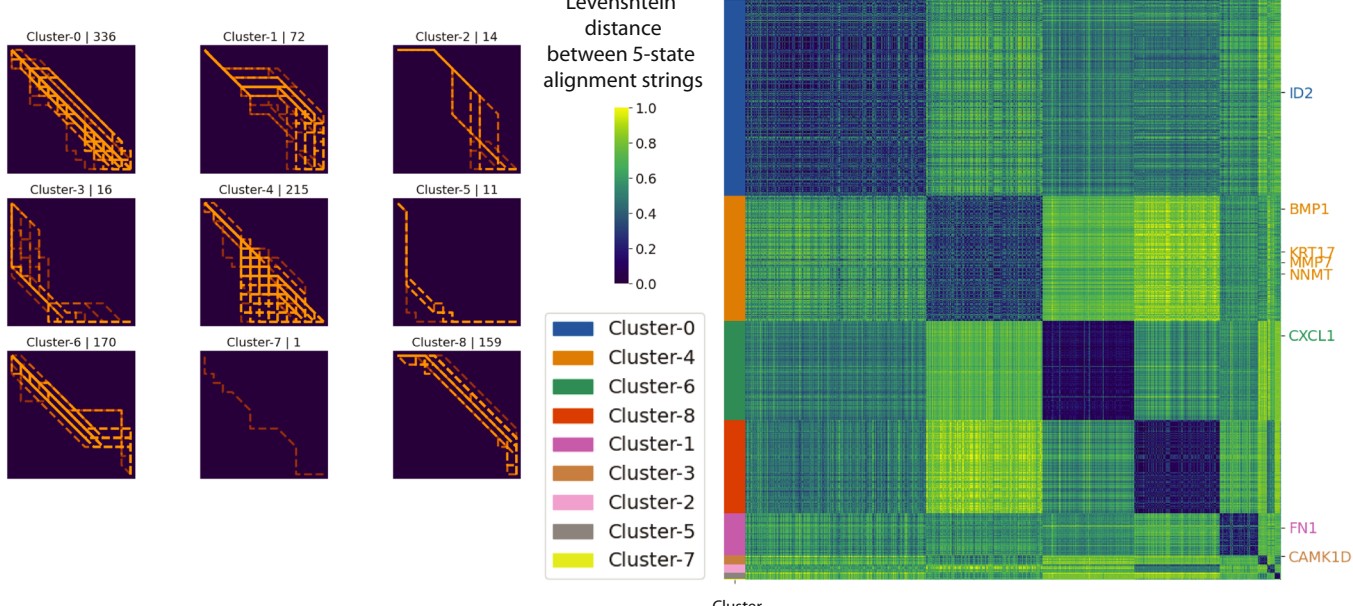

**Extended Data Fig. 5 | The healthy versus Idiopathic Pulmonary Fibrosis (IPF) disease case study alignment clustering outputs. a,** Density plot of the alignment similarity distribution (that is distribution of the percentage of matches/warps across all the alignment outputs) for the 994 highly variable genes in the dataset. **b,** Cluster diagnostic plots for the hierarchical agglomerative clustering of those 994 gene alignments in terms of the mean Silhouette score when varying the Levenshtein distance threshold (or the number of clusters). The highest number of clusters represent the number of all unique 5-state alignment strings (that is 325 strings). Bold highlighted circles mark the local optimal mean Silhouette score which gives nine optimal clusters for the genes at 0.48 distance threshold. **c,** The identified clustering structure. Left: Each plot titled by "Cluster-x | n" is the pairwise matrix of reference and query time points, visualizing alignment paths for all the genes (one alignment per gene and a total of n genes in the cluster) in a cluster x. Right: The clustermap of the pairwise Levenshtein distance matrix of all 994 gene alignments.

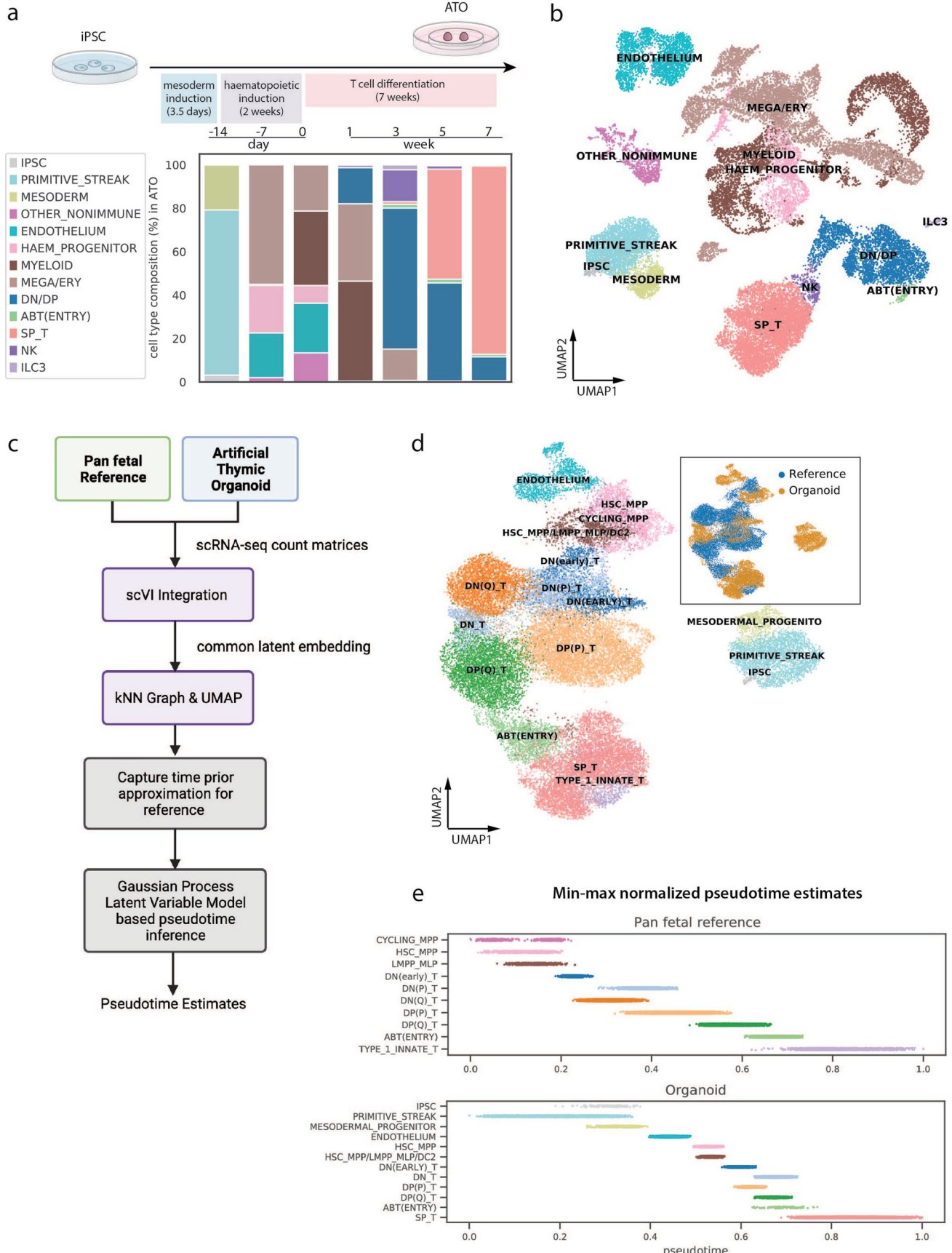

**Extended Data Fig. 6 | See next page for caption.**

**Extended Data Fig. 6 | In vivo, in vitro human T cell development data integration and pseudotime inference. a**, Top: schematic showing the experimental setup of T cell differentiation from iPSCs in ATOs. Bottom: barplot of cell type composition in ATO at different time points during differentiation. **b**, UMAP visualization of different cell types in the ATO dataset (low-level annotation, number of cells $n$ = 31,483), with more refined annotation in Extended Data Fig. 7a. **c**, Workflow of integrating in vitro (that is ATO) and in vivo (that is pan

fetal reference from Suo et al.[39]) human T cell development data and pseudotime inference using GPLVM. **d**, Main: UMAP visualization of integrated in vivo and in vitro human T cell development data, colored by the cell types. Right insert: the same UMAP visualization colored by the data source. **e**, Stripplot of the inferred pseudotime ($x$ axis) against different cell types ($y$ axis), colored by the cell types, of in vivo pan fetal reference data (top) and in vitro organoid data (bottom). Illustrations in **a,c** were created using BioRender (https://biorender.com).

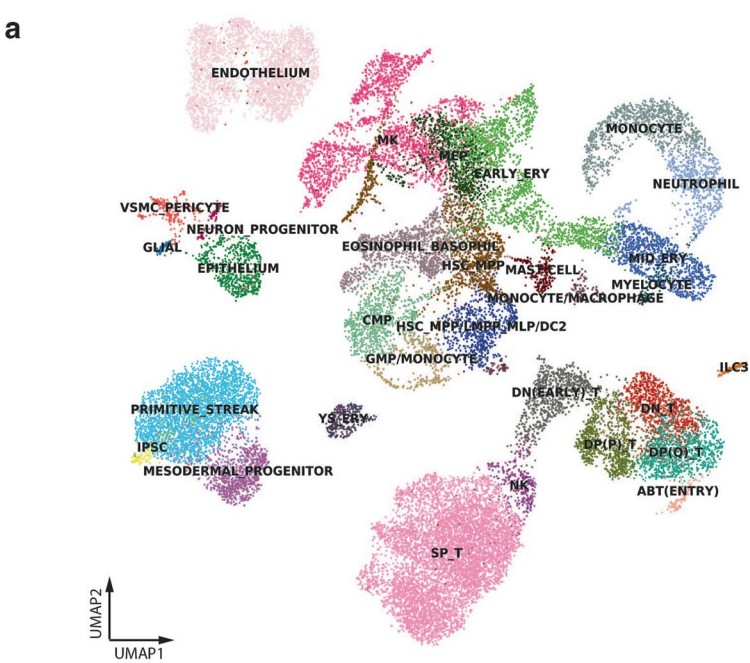

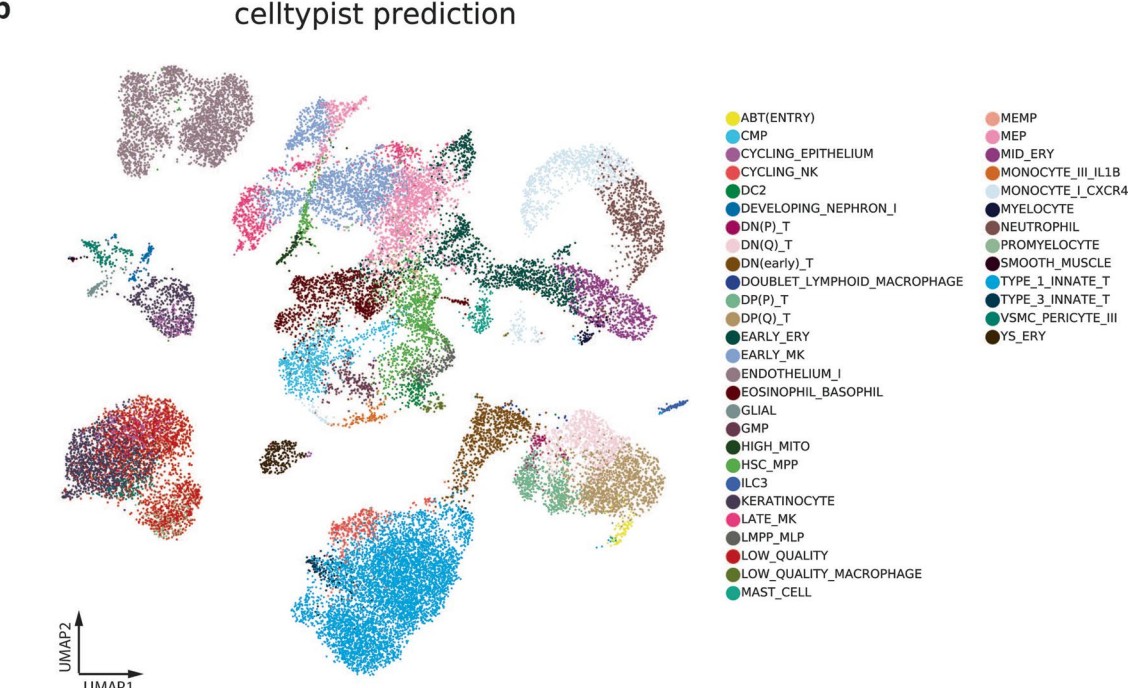

**Extended Data Fig. 7 | Analysis of artificial thymic organoid scRNA-seq data.**
**a**, UMAP visualization of different cell types in the ATO (refined annotation).
iPSC: induced pluripotent stem cell, HSC_MPP: hematopoietic stem cell, and
multipotent progenitor, LMPP_MLP: lymphoid-primed multipotent progenitor
and multi lymphoid progenitor, DC: dendritic cell, CMP: common myeloid
progenitor, GMP: granulocyte and monocyte progenitor, MK: megakaryocyte,
MEP: megakaryocyte erythroid progenitor, YS_ERY: yolk sac-like erythrocyte,
EARLY_ERY: early erythrocyte, MID_ERY: mid-stage erythrocyte, DN(EARLY) T:
early double negative T cell, DN T: double negative T cell, DP(P) T: proliferating
double positive T cell, DP(Q) T: quiescent double positive T cell, SP T: single
positive T cell, NK: natural killer cell, ILC: innate lymphoid cell. **b**, Predicted
annotations from logistic regression model with CellTypist[41] using the
developing human immune atlas[39] as the training dataset, overlaid on the same
UMAP plot as in **a**.

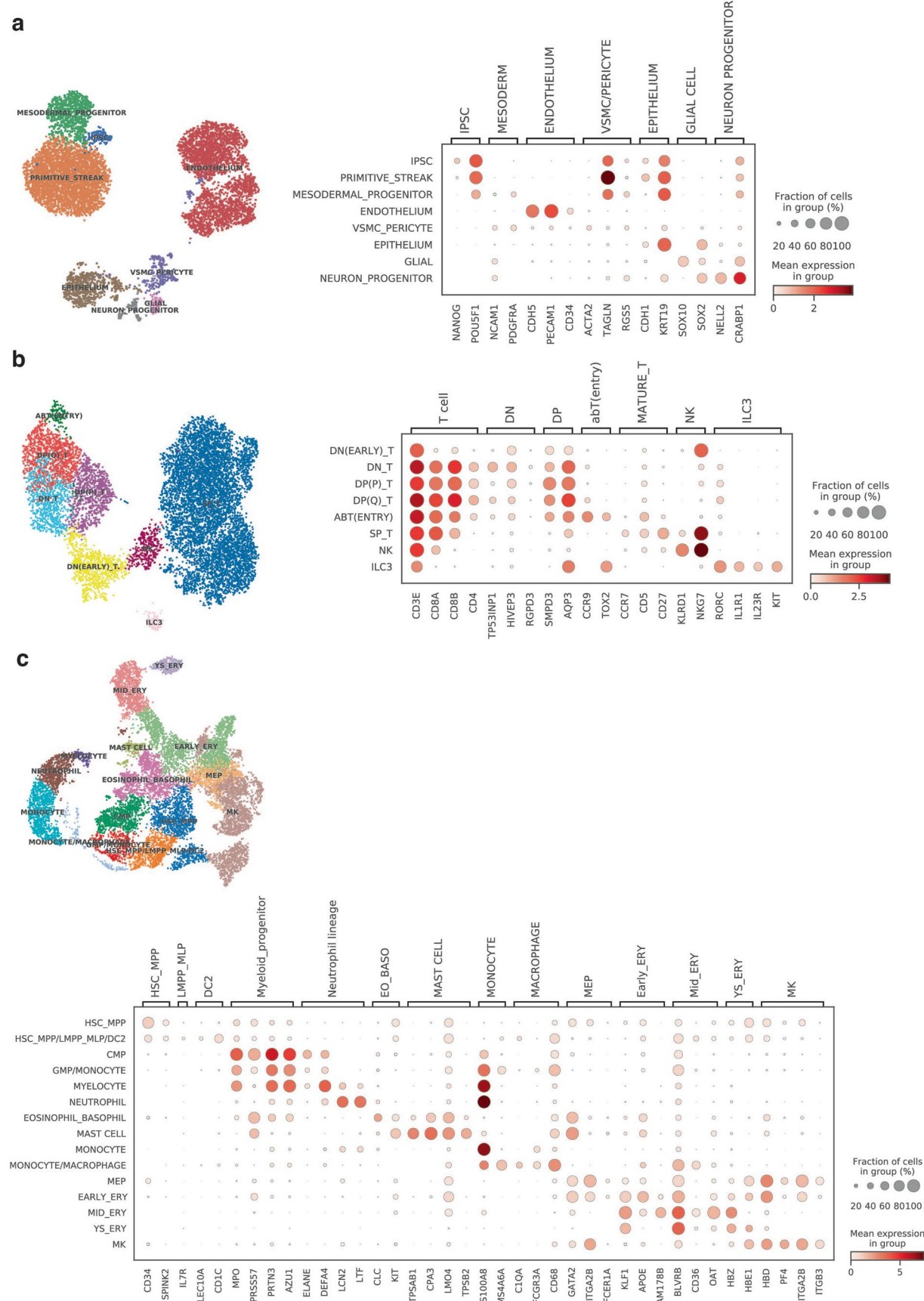

**Extended Data Fig. 8 | Annotation of artificial thymic organoid scRNA-seq data.** For each subset lineage embedding generated through scVI, we show UMAP embeddings of cells colored by annotated cell populations and dot plots of mean expression (log-normalized counts, dot color) and fraction of expressing cells (dot size) of marker genes (columns) used for cell population annotation (rows). **a**, Annotation of non-hematopoietic cells. **b**, Annotation of T/ILC/NK lineage cells. **c**, Annotation of other hematopoietic cells that are not in **b**.

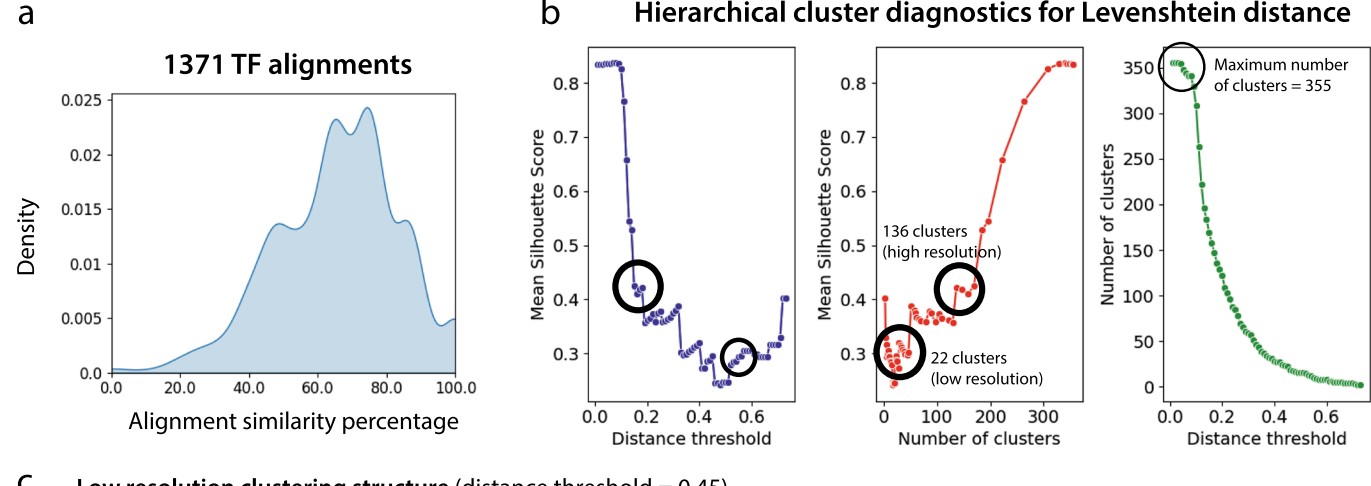

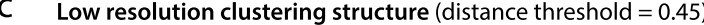

**Extended Data Fig. 9 | Pan fetal reference vs artificial thymic organoid alignment clustering outputs. a**, Density plot of the alignment similarity distribution (that is distribution of the percentage of matches/warps across all the alignment outputs) for all 1371 transcription factors. **b**, Cluster diagnostic plots for the hierarchical agglomerative clustering of those 1371 TF alignments in terms of the mean Silhouette score when varying the Levenshtein distance threshold (or the number of clusters). The highest number of clusters represent the number of all unique 5-state alignment strings (that is 355 strings). Bold highlighted circles mark the local optimal mean Silhouette scores which give 22 optimal clusters for the genes at 0.45 distance threshold (low resolution),

and 136 clusters at 0.18 distance threshold (high resolution). **c**, The identified clustering structure. Left: Each plot titled by "Cluster-x | n" is the pairwise matrix of reference and query time points, visualizing alignment paths for all the genes (one alignment per gene and a total of n genes in the cluster) in a cluster x. Right: The clustermap of the pairwise Levenshtein distance matrix of all TF alignments. Bottom: Identified interesting clusters (that is Cluster 2 representing early mismatched TFs, Cluster 0 representing middle mismatched TFs, Cluster 5 & 10 representing almost 100% mismatched TFs), with their aggregate alignments as 5-state strings.

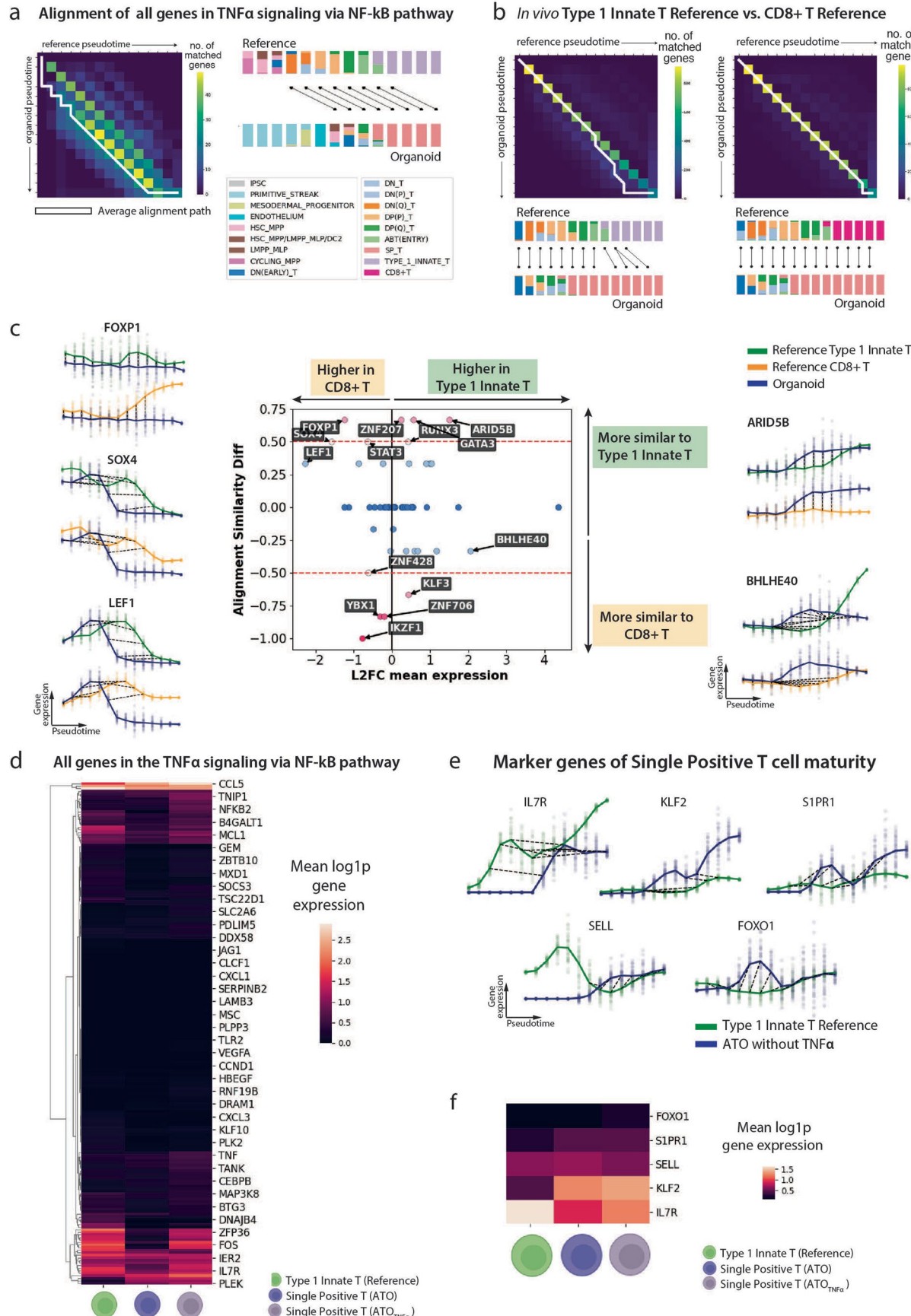

**Extended Data Fig. 10 | See next page for caption.**

**Extended Data Fig. 10 | Further downstream analysis of in vivo, in vitro human T cell development alignment. a**, Genes2Genes aggregate alignment for all 196 genes in the TNFα signaling via NF-κB pathway, between in vitro organoid (ATO) and in vivo reference. Left: pairwise time point matrix between reference and organoid. Color represents total gene count showing a match between corresponding time points. White line represents the average alignment path. Right: Schematic aggregate mapping between pseudotime points. Stacked bar-plots represent reference and organoid cell-type compositions across 14 equispaced pseudotime points; Black lines represent matches. **b**, Aggregate alignment similar to **a**, between in vivo Type 1 Innate T cell reference (T1) and ATO across 1220 TFs (left), and CD8+ Reference and ATO across 1219 TFs. **c**, Alignment differences between in vivo conventional CD8+ T lineage versus organoid, and T1 lineage versus organoid. Middle: alignment similarity difference (*y* axis) against log$_2$ fold change of mean expression between CD8 + T and T1 cells (*x* axis). Color reflects the absolute value of alignment similarity difference. Surrounding plots: the interpolated log1p-normalized (that is per-cell total

sum of the raw transcript counts normalized to 10,000 and log1p-transformed) expression (*y* axis) against pseudotime (*x* axis) showing the alignment between T1 lineage (green) and ATO (blue) (top), and the alignment between CD8 + T lineage (orange) and ATO (blue) (bottom), for four selected genes. Bold lines represent mean expression trends; Faded data points are 50 random samples from the estimated expression distribution at each time point. Black dashed lines represent matches. **d**, Heatmap of mean log1p-normalized gene expression of all 196 genes within TNF signaling pathway in in vivo type 1 innate T cells, ATO SP T cells, and TNF-treated-ATO SP T cells. **e**, Gene-level alignments (similar to plots in **c**) for five, single positive (SP) T cell maturity markers (*IL7R, KLF2, S1PR1, SELL, FOXO1*) between T1 and ATO. **f**, Mean log1p-normalized gene expression of those marker genes compared across reference type 1 innate T cells, ATO SP T cells, and TNF-treated-ATO SP T cells. Illustrations in **d-e** were created using BioRender (https://biorender.com). All interpolations and statistics were generated using our Genes2Genes framework.

# Reporting Summary

## Statistics

For all statistical analyses, confirm that the following items are present in the figure legend, table legend, main text, or Methods section.

| n/a | Confirmed | |
|---|---|---|
| ☐ | ☒ | The exact sample size (*n*) for each experimental group/condition, given as a discrete number and unit of measurement |
| ☐ | ☒ | A statement on whether measurements were taken from distinct samples or whether the same sample was measured repeatedly |
| ☐ | ☒ | The statistical test(s) used AND whether they are one- or two-sided *Only common tests should be described solely by name; describe more complex techniques in the Methods section.* |
| ☐ | ☒ | A description of all covariates tested |
| ☐ | ☒ | A description of any assumptions or corrections, such as tests of normality and adjustment for multiple comparisons |
| ☐ | ☒ | A full description of the statistical parameters including central tendency (e.g. means) or other basic estimates (e.g. regression coefficient) AND variation (e.g. standard deviation) or associated estimates of uncertainty (e.g. confidence intervals) |
| ☐ | ☒ | For null hypothesis testing, the test statistic (e.g. *F*, *t*, *r*) with confidence intervals, effect sizes, degrees of freedom and *P* value noted *Give P values as exact values whenever suitable.* |
| ☐ | ☒ | For Bayesian analysis, information on the choice of priors and Markov chain Monte Carlo settings |
| ☒ | ☐ | For hierarchical and complex designs, identification of the appropriate level for tests and full reporting of outcomes |
| ☒ | ☐ | Estimates of effect sizes (e.g. Cohen's *d*, Pearson's *r*), indicating how they were calculated |

*Our web collection on statistics for biologists contains articles on many of the points above.*

## Software and code

Policy information about availability of computer code

| | |
|---|---|
| Data collection | No software was used during data collection |
| Data analysis | We used the following Python software libraries for our analyses and software development.<br><br>Python (v3.8), cellranger (v3.0.2), souporcell (v2.4.0), scrublet (v0.2.3), scanpy (v1.9.6), scvi (v0.14.5), pyro (v1.8.0), bbknn (v1.5.1), celltypist (v0.1.4), leven (v1.0.4), scipy (v1.10.1), optBinning (v0.18.0), gpytorch (v1.5.1), statsmodels (v0.13.5), gseapy (v1.0.4), cellalign (v0.1.0), sklearn (v1.2.2), seaborn >=v0.12.2 pandas>=v2.0.3, regex >= v2.5.135, matplotlib >= v3.7.1, numpy<v2.<br><br>Code availability: All Python source code and data analysis notebooks are publicly available at: https://github.com/Teichlab/Genes2Genes and https://github.com/Teichlab/G2G_notebooks. Genes2Genes is implemented as an open-source Python package. Our GitHub repository provides installation instructions and technical documentation for interpreting results. |

For manuscripts utilizing custom algorithms or software that are central to the research but not yet described in published literature, software must be made available to editors and reviewers. We strongly encourage code deposition in a community repository (e.g. GitHub). See the Nature Portfolio guidelines for submitting code & software for further information.

## Data

Policy information about availability of data

All manuscripts must include a data availability statement. This statement should provide the following information, where applicable:
- Accession codes, unique identifiers, or web links for publicly available datasets
- A description of any restrictions on data availability
- For clinical datasets or third party data, please ensure that the statement adheres to our policy

> Data used to perform analyses in the manuscript are available at: https://zenodo.org/records/11182400 and https://github.com/Teichlab/G2G_notebooks. All generated alignments are available as Supplementary Data. Raw sequencing data for newly generated sequencing libraries have been deposited in ArrayExpress (accession number E-MTAB-12720).

## Human research participants

Policy information about studies involving human research participants and Sex and Gender in Research.

| | |
|---|---|
| Reporting on sex and gender | N/A |
| Population characteristics | N/A |
| Recruitment | N/A |
| Ethics oversight | N/A |

Note that full information on the approval of the study protocol must also be provided in the manuscript.

# Field-specific reporting

Please select the one below that is the best fit for your research. If you are not sure, read the appropriate sections before making your selection.

☒ Life sciences   ☐ Behavioural & social sciences   ☐ Ecological, evolutionary & environmental sciences

For a reference copy of the document with all sections, see nature.com/documents/nr-reporting-summary-flat.pdf

# Life sciences study design

All studies must disclose on these points even when the disclosure is negative.

| | |
|---|---|
| Sample size | No sample size calculations were performed. Sample sizes were determined based on the availability of datasets. When sub-sampling from each cell population, we defined a sufficient minimum number of 500 samples, otherwise the small populations were retained. |
| Data exclusions | No data were excluded. |
| Replication | Our artificial thymic organoid had two different iPSC lines. Downloaded Pan fetal reference (Suo et al. 2022) had 33 batches (due to multiple donors and 3' vs 5' 10X chemistry). Downloaded Healthy/IPF datasets (Adams et al. 2020) had 28 donors and 31 donors, respectively. Downloaded PAM/LPS and mouse pancreas trajectory datasets did not have the notion of replicates. |
| Randomization | N/A as genome-wide single cell RNA-sequencing was performed unbiased without needing any a priori information |
| Blinding | N/A as genome-wide single cell RNA-sequencing was performed unbiased without needing any a priori information |

# Reporting for specific materials, systems and methods

We require information from authors about some types of materials, experimental systems and methods used in many studies. Here, indicate whether each material, system or method listed is relevant to your study. If you are not sure if a list item applies to your research, read the appropriate section before selecting a response.

## Materials & experimental systems

| n/a | Involved in the study |
|---|---|
| ☐ | ☒ Antibodies |
| ☐ | ☒ Eukaryotic cell lines |
| ☒ | ☐ Palaeontology and archaeology |
| ☒ | ☐ Animals and other organisms |
| ☒ | ☐ Clinical data |
| ☒ | ☐ Dual use research of concern |

## Methods

| n/a | Involved in the study |
|---|---|
| ☒ | ☐ ChIP-seq |
| ☐ | ☒ Flow cytometry |
| ☒ | ☐ MRI-based neuroimaging |

# Antibodies

| Antibodies used | PE anti-human CD326 antibody, Biolegend, 324205<br>APC anti-human CD56 antibody, Biolegend, 318309<br>APC/Cy7 anti-mouse CD29 antibody, Biolegend, 102225<br>BV785 anti-human CD45 antibody, Biolegend, 304047<br>TotalSeq-C0251, Biolegend, 394661<br>TotalSeq-C0252, Biolegend, 394663<br>TotalSeq-C0253, Biolegend, 394665<br>TotalSeq-C0254, Biolegend, 394667<br>TotalSeq-C0255, Biolegend, 394669 |
|---|---|
| Validation | All antibodies were obtained from commercial vendors and were validated by the vendors.<br>PE anti-human CD326 antibody, Biolegend, 324205<br>https://d1spbj2x7qk4bg.cloudfront.net/en-gb/products/pe-anti-human-cd326-epcam-antibody-3757?pdf=true&displayInline=true&leftRightMargin=15&topBottomMargin=15&filename=PE%20anti-human%20CD326%20(EpCAM)%20Antibody.pdf&v=20240411093413<br><br>APC anti-human CD56 antibody, Biolegend, 318309<br>https://d1spbj2x7qk4bg.cloudfront.net/en-ie/products/apc-anti-human-cd56-ncam-antibody-3798?pdf=true&displayInline=true&leftRightMargin=15&topBottomMargin=15&filename=APC%20anti-human%20CD56%20(NCAM)%20Antibody.pdf&v=20240412063148<br><br>APC/Cy7 anti-mouse CD29 antibody, Biolegend, 102225<br>https://d1spbj2x7qk4bg.cloudfront.net/en-gb/products/apc-cyanine7-anti-mouse-rat-cd29-antibody-6184?pdf=true&displayInline=true&leftRightMargin=15&topBottomMargin=15&filename=APC/Cyanine7%20anti-mouse/rat%20CD29%20Antibody.pdf&v=20240410063626<br><br>BV785 anti-human CD45 antibody, Biolegend, 304047<br>https://d1spbj2x7qk4bg.cloudfront.net/nl-nl/products/brilliant-violet-785-anti-human-cd45-antibody-9325?pdf=true&displayInline=true&leftRightMargin=15&topBottomMargin=15&filename=Brilliant%20Violet%20785%E2%84%A2%20anti-human%20CD45%20Antibody.pdf&v=20240411093413<br><br>TotalSeq-C0251, Biolegend, 394661<br>https://d1spbj2x7qk4bg.cloudfront.net/nl-nl/products/totalseq-c0251-anti-human-hashtag-1-antibody-17162?pdf=true&displayInline=true&leftRightMargin=15&topBottomMargin=15&filename=TotalSeq%E2%84%A2-C0251%20anti-human%20Hashtag%201%20Antibody.pdf&v=20240208073156<br><br>TotalSeq-C0252, Biolegend, 394663<br>https://d1spbj2x7qk4bg.cloudfront.net/nl-nl/products/totalseq-c0252-anti-human-hashtag-2-antibody-17163?pdf=true&displayInline=true&leftRightMargin=15&topBottomMargin=15&filename=TotalSeq%E2%84%A2-C0252%20anti-human%20Hashtag%202%20Antibody.pdf&v=20240208073156<br><br>TotalSeq-C0253, Biolegend, 394665<br>https://d1spbj2x7qk4bg.cloudfront.net/nl-nl/products/totalseq-c0253-anti-human-hashtag-3-antibody-17164?pdf=true&displayInline=true&leftRightMargin=15&topBottomMargin=15&filename=TotalSeq%E2%84%A2-C0253%20anti-human%20Hashtag%203%20Antibody.pdf&v=20240208073156<br><br>TotalSeq-C0254, Biolegend, 394667<br>https://d1spbj2x7qk4bg.cloudfront.net/nl-nl/products/totalseq-c0254-anti-human-hashtag-4-antibody-17165?pdf=true&displayInline=true&leftRightMargin=15&topBottomMargin=15&filename=TotalSeq%E2%84%A2-C0254%20anti-human%20Hashtag%204%20Antibody.pdf&v=20240208073156<br><br>TotalSeq-C0255, Biolegend, 394669<br>https://d1spbj2x7qk4bg.cloudfront.net/nl-nl/products/totalseq-c0255-anti-human-hashtag-5-antibody-17166?pdf=true&displayInline=true&leftRightMargin=15&topBottomMargin=15&filename=TotalSeq%E2%84%A2-C0255%20anti-human%20Hashtag%205%20Antibody.pdf&v=20240208073156 |

# Eukaryotic cell lines

Policy information about cell lines and Sex and Gender in Research

| | |
|---|---|
| Cell line source(s) | MS5 line transduced with human DLL4 was obtained from G. Crooks (UCLA) as a gift. Two iPSC lines were used in this study. Cell lines HPSI0114i-kolf_2 (Kolf) and HPSI0514i-fiaj_1 (Fiaj) were obtained from the Human Induced Pluripotent Stem Cell initiative (HipSci: www.hipsci.org) collection. |
| Authentication | None of the cell lines used were authenticated. |
| Mycoplasma contamination | All lines were tested negative for mycoplasma contamination. |
| Commonly misidentified lines (See ICLAC register) | Nil |

# Flow Cytometry

## Plots

Confirm that:

☒ The axis labels state the marker and fluorochrome used (e.g. CD4-FITC).

☒ The axis scales are clearly visible. Include numbers along axes only for bottom left plot of group (a 'group' is an analysis of identical markers).

☒ All plots are contour plots with outliers or pseudocolor plots.

☒ A numerical value for number of cells or percentage (with statistics) is provided.

## Methodology

| | |
|---|---|
| Sample preparation | Cells were harvested by centrifugation, resuspended in FACS buffer (PBS + 0.5% FBS + 2mM EDTA) and stained with antibody mixes for 30 min at 4 degree celsius. All antibodies used were added in as 2 µl per antibody in a total of 100 µl staining solution. Cells were then washed once with FACS buffer before flow cytometry. |
| Instrument | Beckman Coulter CytoFLEX, BD Influx, ThermoFisher Bigfoot Spectral Cell sorters |
| Software | Manufacturer's default softward was used. Analysis was done in Flowjo v10. |
| Cell population abundance | Shown in Supplementary Fig.7. |
| Gating strategy | For flow cytometry gating, cells were gated on FSC/SSC; then SSC-W vs SSC-H or FSC-W vs FSC-A for singlets; and individual stains were gated based on negative controls. |

☒ Tick this box to confirm that a figure exemplifying the gating strategy is provided in the Supplementary Information.

