## [Peer Review File · Nature Methods]

Peer Review Information

Manuscript Title: Gene-level alignment of single cell trajectories

Corresponding author name(s): Sarah Teichmann

Editorial Notes: None

Reviewer Comments & Decisions:

Decision Letter, initial peer review:

Dear Sarah,

Your Article, "Gene-level alignment of single cell trajectories", has now been seen by 3 reviewers. As you will see from their comments below, although the reviewers find your work of considerable potential interest, they have raised a number of concerns. We are interested in the possibility of publishing your paper in Nature Methods, but would like to consider your response to these concerns before we reach a final decision on publication.

We therefore invite you to revise your manuscript to address these concerns especially by adding adequate benchmarking against existing similar methods.

[Redacted]

We hope to receive your revised paper within 8 weeks. If you cannot send it within this time, please let us know. In this event, we will still be happy to reconsider your paper at a later date so long as nothing similar has been accepted for publication at Nature Methods or published elsewhere.

OPEN SCIENCE REQUIREMENTS

REPORTING SUMMARY AND EDITORIAL POLICY CHECKLISTS

DATA AVAILABILITY

All novel DNA and RNA sequencing data, protein sequences, genetic polymorphisms, linked genotype and phenotype data, gene expression data, macromolecular structures, and proteomics data must be deposited in a publicly accessible database, and accession codes and associated hyperlinks must be provided in the "Data Availability" section.

CODE AVAILABILITY

Please include a "Code Availability" subsection in the Online Methods which details how your custom code is made available. Only in rare cases (where code is not central to the main conclusions of the paper) is the statement "available upon request" allowed (and reasons should be specified).

For more information on our code sharing policy and requirements, please see: <https://www.nature.com/nature-research/editorial-policies/reporting-standards#availability-of-computer-code>

MATERIALS AVAILABILITY

More details about our materials availability policy can be found at <https://www.nature.com/nature->

portfolio/editorial-policies/reporting-standards#availability-of-materials

SUPPLEMENTARY PROTOCOL

To help facilitate reproducibility and uptake of your method, we ask you to prepare a step-by-step Supplementary Protocol for the method described in this paper. We encourage authors to share their step-by-step experimental protocols on a protocol sharing platform of their choice and report the protocol DOI in the reference list. Nature Portfolio 's Protocol Exchange is a free-to-use and open resource for protocols; protocols deposited in Protocol Exchange are citable and can be linked from the published article. More details can found at www.nature.com/protocolexchange/about.

ORCID

Nature Methods is committed to improving transparency in authorship. As part of our efforts in this direction, we are now requesting that all authors identified as 'corresponding author' on published papers create and link their Open Researcher and Contributor Identifier (ORCID) with their account on the Manuscript Tracking System (MTS), prior to acceptance. This applies to primary research papers only. ORCID helps the scientific community achieve unambiguous attribution of all scholarly contributions. You can create and link your ORCID from the home page of the MTS by clicking on 'Modify my Springer Nature account'. For more information please visit please visit www.springernature.com/orcid.

Happy Holidays!

Sincerely,
Madhura

Madhura Mukhopadhyay, PhD
Senior Editor
Nature Methods

Reviewers' Comments:

Reviewer #1:

Remarks to the Author:

This work presents an approach for aligning single-cell transcriptomic trajectories. The framework, Genes2Genes (G2G), improves upon existing approaches primarily by relaxing assumptions regarding how trajectories align and by more directly accommodating mismatches. The model development is well motivated and sufficiently detailed so that it is clear what the authors are proposing. The paper is also well written.

Trajectory alignment remains an important challenge in scRNA-seq studies, and while there are a few existing approaches, there is substantial room for improvement. Given the comprehensive method proposed and the results shown, I expect G2G will become a popular approach with major impact. I hope that the authors will address the following relatively minor comments.

\log_{1p} should be defined. If it's what I think it is ($\log_{1p}(x) = \ln(1+x)$), it's not what people typically refer to as normalization, where sequencing depth is adjusted. Please clarify.

The authors should comment on computational requirements, running times.

The authors refer to "missing data" to mean zero expression or a cell state missing in a given condition.

This isn't missing data in the traditional sense; I might call these unobserved.

The authors note that other approaches suffer from requiring ad-hoc thresholds, but the authors should acknowledge that G2G also requires some fine tuning. The "optimally-chosen 0.22 Levenshtein distance" is one example. A method is proposed in "Choosing the distance threshold for hierarchical clustering" for estimating a threshold; it amounts to calculating mean Silhouette coefficients for clusters estimated with thresholds varying within a range.

They note that the best clustering structure is typically the one with the highest mean Silhouette coefficient, but then end up choosing 0.22 which is the second highest. What is recommended in practice?

G2G relies on estimated trajectories and the authors recommend users "select a reliable method". I'd suggest users evaluate results under a few different methods in an effort to determine how robust G2G results are to different pseudotime estimates.

The legend of Figure 1a does not describe what is in that figure. It should be updated.

It's helpful to see the actual gene expression values (e.g. as in Figure 4c and 5e); please show for 3h.

Reviewer #2:

Remarks to the Author:

Sumanaweera et al present Genes2Genes, an algorithm that compares two single cell trajectories at the per-gene level. The approach is novel in its use and extension of approaches, typically used for sequence alignment, in a new context of expression profiles arranged by their pseudotime/developmental trajectory. While the concept is interesting and the authors describe some positive results from using Genes2Genes in comparison of T cell differentiation in vitro and in vivo, there are challenges to be addressed.

- G2G doesn't aim to generate a single alignment between single cell trajectories, but rather characterises the differences between them per gene via the DTW approach. While an approach like CellAlign (mentioned in the manuscript) assumes there is some mapping (potentially non-linear & potentially with gaps) of one trajectory into the same pseudotime reference coordinates of another trajectory, G2G doesn't make the same kind of assumption. In this regard G2G can be thought of as

an approach for differential trajectory analysis. There are other approaches e.g. 10.1101/2021.07.10.451910 and 10.1101/2021.03.09.433671 for which G2G should be placed in context and compared.

- In the five-state model, there doesn't appear to be a state for situations where there is simply not enough information to be able to select a state (e.g. low or highly variable expression). How are these handled and is there a 'default' class? I imagine some genes may 'match' in the trajectory because they are equally unimportant in defining each trajectory. One could simulate and artificially downsample cells/counts to directly examine this behaviour.

- It would be worth spelling out what is the drawbacks of any naive kind of implementation, e.g. calculating correlation or MI of binned expression along the two trajectories. Might it be in sensitivity of identifying differences, or in difficulty of interpretation?

- The method relies extremely heavily on the trajectory of reference and query, and very little attention appears to be made on the choices that go into building a single cell pseudotime/developmental trajectory in the manuscript. Fundamentally this is estimated using the gene expression profiles, which can be given more or less importance via feature selection or weighting, therefore the effect of the G2G results can be driven by the choices taken a priori in the trajectory inference method. To what extent does the trajectory inference affect G2G results? Here, diffusion pseudotime is used, could an approach like monocle or slingshot also be appropriate?

- It would be helpful to characterise the concept of indels in a single cell trajectory context. In the sequence alignment context this is very well characterised as mutations/variations that insert or delete into the DNA/RNA sequence, but for a single cell trajectory this needs to be clarified much further - could this represent new transient cell states? or bypassing some state?

- G2G handles two linear trajectories and at present cannot handle branched trajectories. One could consider a sixth state in the five-state alignment approach that is presence of a branching.

- It's unclear whether the other approaches mentioned (DTW or CellAlign) fail to identify or describe the differences in the in vivo and in vitro T cell differentiation dynamics.

- G2G is performed on each gene independently, and then per-gene match/mismatch sequences are clustered to identify groups of genes with a similar alignment pattern. It may be useful to combine genes according to a priori knowledge to increase robustness in the alignment results. This could be done by performing the denoising after some dimensionality reduction step.

- Unclear how symmetric the G2G approach is, if you swap reference and query, do you get consistent results? For the in vitro vs in vivo comparison, it's not clear which really should be reference and which should be query. How would a user decide how to allocate trajectories to reference/query?

- There is some speculation on the drivers behind the unexpected results related to Supplementary Fig. 3a, the authors could continue the simulation study to identify what are the drivers behind the unexpected behaviour, e.g. remove cells at different points along the trajectory.

Reviewer #3:

Remarks to the Author:

The manuscript "Gene-level alignment of single cell trajectories" by Sumanaweera et al offers a novel algorithm for aligning pseudotime profiles between different conditions or datasets. It seems like an exciting approach and it tackles an important question (would have been of help in numerous of our own datasets over the years) in what I believe is a novel way that makes a lot of sense conceptually. Also test their methods on lots of data, which is always good to see. There are a few comments (mostly minor, a few a bit more major), however, that I think should be addressed before publication (listed below more or less in the order I came across them in the paper, so minor and less minor ones are mixed in):

1. Maybe add a few words about the choice of normalization used for the algorithm and if it matters or was somewhat arbitrary (log TPM vs log TP10K vs log counts vs others)—at the very least not clear from the main text which normalization is used (based on the methods think it is TP10K, yes?). Related, talk about log1p normalized expression, should make clear what is being log scaled (TP10K?). At the very least figure legends should be explicit about how the data was normalized, not just say 'normalized expression'.
2. Since the alignment is at the gene level, would be good to see how gene level meta data (most notable expression level) might affect the results.
3. Very minor, but should mention number of cells in experiment 2 when first talking about dataset in results section.
4. More major: Experiment 2 gives useful information, but in practice when aligning cells they will come from different 10X channels, individuals, etc, not just be one dataset split in half. Would be great to see the results when you split by batch instead of at the cell level. If not realistic at least need to discuss.
5. Related to the above, can confounders be taken into account? Either by regressing them out before hand or in the model itself? The methods seem to hint at this possibility but would be great to see examples.
6. Very minor, but for healthy vs IPF mention number of individuals in each group.
7. More major: The healthy vs IPF example is great, my one concern is that the approach does not take into account individual to individual variability (unless I am missing it?). Given that such variability is known to lead to inflated DE and cell type composition results if not taken care of, would be good to know how it effects the method when comparing between groups (I am particularly worried about in cases with more subtle changes). Would be great to address this somehow. Some ideas (don't need to do these particular approaches, should just do something): one possibility would be to take a dataset (maybe the IPF dataset, maybe just the controls from it?), split it in half (assign half the individuals as cases and half as controls), and see what kinds of significant results one gets trying to align them. Another approach could be some kind of down sampling (sample one or a few individuals from each condition and do the analysis just on them and repeat the analysis (with a few such splits), see if the results hold up). Note having false positives here isn't make or break (can often do follow up experiments as you do with the TNF bit later), just important to know about when interpreting results/deciding how much faith to give them (particularly in the absence of follow up experiments).
8. In the organoid vs in vitro data comparison a lot of the mismatched genes you display (a lot of the TNF related ones) are immediate early genes (as you kind of get at with the comment about stress activation in the supplement). Given that such genes are known to show up due to more technical reasons (handling, etc), and do so differentially between different cell types (<https://www.nature.com/articles/s41593-022-01022-8>) might want to add a few words discussing why you think this is a real signal and not an artifact of some type (might be obvious but if so I

missed it). You get close to addressing this in the supplement when talking about the relation to stress but would like a little more detail on how it might (or might not) relate to handling in particular.

9. Add more details on what the change point kernel plots are to the legend for extended fig 1 (Since not a common plot)

10. For legend for extended data fig 2a/b explain what the color boxes mean

11. What kind of runtime is involved? Saw some big-O type analysis, but unless I missed it no benchmarking of runtime? Would be good to have some info on that.

12. For the analysis of the data (the standard clustering, pseudotime, etc) make sure the versions of the packages used and the non-default arguments for those tools (if any) are given, or the associated scripts are available on your github.

Tested the code for the package a bit. Got it to install and performed the alignment. The one issue I had was with visualizations that did not seem to work with the system I was using (got the error "qt.qpa.plugin: Could not load the Qt platform plugin "xcb" in "" even though it was found."). Once I updated Seaborn to a newer version, however, it started to work (might be useful for others having similar issues). Might be worth putting a warning on the github or add something to require more up to date Seaborn installation (not sure if this is a wide spread issue or specific to the system I was on).

Author Rebuttal to Initial comments

Response to referees

Please find enclosed our revised manuscript. We would first like to thank you and all reviewers for your thoughtful comments and suggestions, which helped us to improve the quality of the manuscript. We have considered each of the comments carefully and have responded to each one in the attached point-by-point rebuttal.

Please note: In the response column, we have used *<black, italic text>* to refer to any originally existing text in our manuscript, and *<blue, italic text>* to refer to any newly added/modified text as part of revision. We used **<bold, yellow highlighted text>** to refer to any page numbers/figures/sections in the manuscript that got revised and/or are relevant to our responses, and **<bold, green highlighted text>** to refer to a reviewer comment numbered in this document. In our revised manuscript, all revised text is highlighted in blue text.

Reviewer #1	
Comment by the reviewer	Response by the authors
1. Reviewer 1 This work presents an approach for aligning single-cell transcriptomic trajectories. The framework, Genes2Genes (G2G), improves upon existing approaches primarily by relaxing assumptions regarding how trajectories align and by more directly accommodating mismatches. The model development is well motivated and sufficiently detailed so that it is clear what the authors are proposing. The paper is also well written. Trajectory alignment remains an important challenge in scRNA-seq studies, and while there are a few existing approaches, there is substantial room for improvement. Given the comprehensive method proposed and the results shown, I expect G2G will become a popular approach with major impact. I hope that the authors will address the following relatively minor comments.	Thank you very much for your accurate summary and positive feedback of our work.
1.1 log_{1p} should be defined. If it's what I think it is (log_{1p}(x) = ln(1+x)), it's not what people typically refer to as normalization, where sequencing depth is adjusted. Please clarify.	Thank you for pointing this out. We have now added the definition of log_{1p} normalization on Page 5: G2G's inputs are log_{1p} normalized (raw transcript counts are normalized over total counts of all genes

	per cell so that every cell has the same total count after normalization, then logarithmically transformed with $\log(\text{normalized count} + 1)$ scRNA-seq matrices of the reference and query systems, and their pseudotime estimates.
1.2 The authors should comment on computational requirements, running times.	Thank you for the suggestion. We have now described computational requirements and runtimes on Page 63: Software packages used in the work Software and computational requirements Genes2Genes framework and all analysis related code (including plot generation) were implemented using the standard Python libraries (Numpy, Pandas, Seaborn, scikit-learn, SciPy, statmodels), GPyTorch, GSEAPy, Scanpy. Illustrations were made using Adobe Illustrator 2023 and BioRender G2G has been tested with Python version ≥ 3.8, on two operating systems: Ubuntu 20.04.1 LTS, and Debian GNU/Linux 10 (buster). The runtime of the algorithm depends on the number of cells in the reference and query datasets, the number of interpolation time points, and the number of genes to align. The current version of G2G utilizes concurrency through Python multiprocessing to speed up the gene-level alignment process. It creates a number of processes equal to the number of cores in the system, and each process performs a single gene-level alignment at one time. Per-gene runtime for a case aligning 100 genes between 20,327 reference cells and 17,176 query cells (i.e. pan fetal reference and ATO datasets), with 14 interpolation points and 16 cores in a Linux (SMP Debian 4.19.304-1 (2024-01-09)) system, was approximately 5.57 seconds.
1.3 The authors refer to "missing data" to mean zero expression or a cell state missing in a given condition. This isn't missing data in the traditional sense; I might call these unobserved.	Many thanks for the comment. We have changed 'missing data' to 'unobserved state' on Page 3 and 5.
1.4 The authors note that other approaches suffer from requiring ad-hoc thresholds, but the authors should	Thank you for raising this point. We discuss the optimal parameter settings for all related parameters in

acknowledge that G2G also requires some fine tuning. The "optimally-chosen 0.22 Levenshtein distance" is one example. A method is proposed in "Choosing the distance threshold for hierarchical clustering" for estimating a threshold; it amounts to calculating mean Silhouette coefficients for clusters estimated with thresholds varying within a range.

They note that the best clustering structure is typically the one with the highest mean Silhouette coefficient, but then end up choosing 0.22 which is the second highest. What is recommended in practice?

Methods section 'Determining the best parameter setting' on Page 51-52. The main set of parameters that requires fine-tuning in the G2G alignment algorithm are the 3 free parameters of the 5-state machine, and this has been optimized on our simulated dataset, as stated on Page 7 (Results):

We used the accuracy rate (the proportion of correct gene alignments) to fine-tune the optimal parameter setting (i.e. state transition probabilities) of the 5-state machine used by our DP algorithm, which was set as the default for G2G (See Methods, Supplementary Table 2).

We would recommend the users to use this optimal setting. To further emphasize this point, we have now added the following on Page 5 when first introducing G2G algorithm:

The five-state machine allows to compute a symmetric cost of assigning an alignment state for R_i and Q_i out of the five possible states. This machine has been empirically fine-tuned over a simulated dataset.

and on Page 45 (Methods):

While we recommend users to use this setting (set as default), we also note that these parameters can be automatically inferred using an added layer of optimization and time complexity on top of the main DP optimization, which is an interesting future direction to follow.

Interpreting the hierarchical clustering results is part of the downstream analysis of G2G alignments, and we would like to note that choosing the optimal clustering structure from running any hierarchical clustering algorithm is an open problem in unsupervised learning.

To initially emphasize that we need to choose an optimal distance threshold, we have now added the text below on Page 6:

The genes displaying similar alignment patterns are then clustered together using agglomerative

hierarchical clustering (where an optimal grouping is determined by inspecting the mean silhouette coefficients with different distance thresholds of the linkage criterion in the clustering algorithm). Alignments and their cluster memberships allow further downstream analysis such as gene set overrepresentation analysis

Commonly, the best clustering structure from a hierarchical clustering of any dataset, is chosen as the one corresponding to the highest mean Silhouette coefficient. However, there is a tradeoff between the Silhouette coefficients and the number of clusters. As we are clustering strings, and there is a high number of unique alignment strings, we observe that the maximum possible number of clusters is equal to the number of those unique strings, and it also always corresponds to the highest mean Silhouette coefficient. **We were unable to find a report on this specific behavior of hierarchical string clustering in literature (and therefore believe that reporting this observation in this work will be useful for others who are studying string clustering in general).** Since this clustering is highly noisy with a large number of clusters (where all unique strings are individual clusters), it is less biologically interpretable. However, we observe that there are clustering structures with suboptimal mean Silhouette coefficients (e.g. **Extended Data Fig. 4b, 4d, Extended Data Fig. 5b, Extended Data Fig. 9b**) that reasonably describe the different and broad groups of alignment patterns. This is why we have chosen the clustering with 0.22 distance threshold that gives the second highest mean Silhouette coefficient for our simulated dataset. This threshold could differ from dataset to dataset, and therefore, we recommend users to inspect the clustering structures and decide on the distance threshold by balancing both silhouette scores and biologically interpretable number of clusters for the final clustering structure.

We have detailed and discussed all the above points on **Page 51**, including the below recommendation:

	we conclude that it is reasonable to choose the distance threshold by inspecting the clustering structures corresponding to the locally optimal mean silhouette scores. We note the importance of a manual inspection to decide on the trade-off between biologically interpretable number of clusters vs. cluster resolution. Our G2G package enables the user to inspect cluster diagnostics through distance vs. mean silhouette score plots, as well as the average alignment pattern of each cluster, to make a final decision.
1.5 G2G relies on estimated trajectories and the authors recommend users "select a reliable method". I'd suggest users evaluate results under a few different methods in an effort to determine how robust G2G results are to different pseudotime estimates.	Thank you very much for pointing this out. We have now added your suggestion below on Page 15. Thus we recommend the user to select a reliable method suitable for their datasets to infer pseudotime trajectories¹ and/or to run G2G with a few different pseudotime inference methods and evaluate how robust the G2G results are to different pseudotime estimates. We also now include a Supplementary section on Page 101-104 on trajectory alignment across different pseudotime estimations, discussing challenges and further user recommendations.
1.6 The legend of Figure 1a does not describe what is in that figure. It should be updated.	Thank you for the comment. We have now updated the Figure 1a legend as below to be more descriptive of the actual figure: Schematic illustration of the concept of single-cell trajectory alignment. It takes the input of single-cell transcriptomic data of a reference and a query system that dynamically changes (left panel). For example, trajectory alignment is important for comparing different single-cell reference and query systems that dynamically change. This could be between in vivo cell development and in vitro cell differentiation, or between control and drug-treated cells in response to the same perturbation, or between the responses to vaccination or pathogen challenge in healthy versus diseased individuals. A complete alignment between

	the reference and the query them can capture matches and mismatches (middle panel), and the alignment results can be processed for further downstream analysis (right panel).
1.7 It's helpful to see the actual gene expression values (e.g. as in Figure 4c and 5e); please show for 3h.	Thank you for the suggestion. We have now updated Figure 3h accordingly.

Reviewer #2	
Comment by the reviewer	Response by the authors
2. Reviewer 2 Sumanaweera et al present Genes2Genes, an algorithm that compares two single cell trajectories at the per-gene level. The approach is novel in its use and extension of approaches, typically used for sequence alignment, in a new context of expression profiles arranged by their pseudotime/developmental trajectory. While the concept is interesting and the authors describe some positive results from using Genes2Genes in comparison of T cell differentiation in vitro and in vivo, there are challenges to be addressed.	Thank you very much for identifying our framework as a novel approach, and for the constructive feedback. We agree that this is indeed a problem with fundamental challenges which we hope to overcome with our new framework.
2.1 G2G doesn't aim to generate a single alignment between single cell trajectories, but rather characterises the differences between them per gene via the DTW approach. While an approach like CellAlign (mentioned in the manuscript) assumes there is some mapping (potentially non-linear & potentially with gaps) of one trajectory into the same pseudotime reference coordinates of another trajectory, G2G doesn't make the same kind of assumption. In this regard G2G can be thought of as an approach for differential trajectory analysis. There are other approaches e.g. 10.1101/2021.07.10.451910 and 10.1101/2021.03.09.433671 for which G2G should be placed in context and compared.	Thank you for raising this point. G2G does not simply characterize the differences between trajectories per gene, but performs the task of computational trajectory alignment per gene by unifying Dynamic Time Warping (DTW) and gap modeling consistently under a new dynamic programming alignment algorithm. G2G does make the assumption of a non-linear mapping (describing one-to-one matches and warps). CellAlign also assumes a non-linear mapping by implementing the DTW algorithm, however, the DTW algorithm is not designed to model gaps, and therefore, CellAlign does not make the assumption of gaps in its mapping. Unlike CellAlign, G2G non-linear mapping accommodates gaps as well. Due to this, while a CellAlign output can be described only by a 3-state alignment string giving all

time points matched, G2G generates a 5-state alignment string, capturing both matches and mismatches between time points. Therefore, G2G expands the capacity of DTW, as described on Page 6. We also illustrate the differences between G2G, CellAlign and the other state of the art trajectory alignment method, TrAGEDy, in Figure 3a and 3b.

When the single-cell trajectories are discretized with pseudotime points, the term “alignment” of two such trajectories refers to finding a mapping between these pseudotime points, regardless of the number of genes being used as dimensions. We can perform this alignment either as a high-dimensional alignment (taking all gene dimensions together), or as multiple alignments with one for each gene dimension. This is echoed in CellAlign paper’s Supplementary Note Page 16, which states that “*Single gene alignment - In cases where comparison of dynamics behavior of key genes is desired, single-gene alignment is recommended. Single genes alignment within a sample can be further used to identify gene-modules via clustering*”. G2G also expands on this idea, and reports both gene-level alignments, as well as a single cell-level alignment aggregated across all those gene-level alignments. In other words, **G2G does aim to generate a single alignment between trajectories**, but as an average mapping based on all individual gene-level alignments rather than a high-dimensional alignment, while also informing the intrinsic differences in the alignment patterns of different genes.

Such alignment is indeed a means to differential expression (DE) analysis over time. However **the current temporal DE analysis methods (including the literature mentioned by the reviewer) do not perform alignment**, and thus they are unable to capture warps and pinpoint the exact locations of mismatch occurrences in genes. For instance, the Lamian test published in 10.1101/2021.07.10.451910 is a hypothesis test to determine if the sample-specific gene expression functions (modeled using B-spline basis functions) are different or not by testing if their regression coefficients are zero, equal, or different. This test only outputs

p-values for the significance of that difference, and does not output a non-linear mapping of pseudotime points. In contrast, **G2G is an algorithm for alignment, which outputs a non-linear mapping between pseudotime points for each gene, an average mapping between pseudotime points across all genes, and a clustering of different alignment patterns.** Therefore the outputs from **such non-alignment methods are not comparable and relevant for benchmarking our G2G trajectory alignment framework.**

Upon reviewer's suggestion, we tested the Lamian DE tool over all gene trajectories in our simulated dataset (taking reference and query as the covariate groups). Note: Lamian implementation does not work for covariate groups with single samples. Therefore we replicated the same reference and query sample as second samples for each covariate group. (This does not create any bias since we are just duplicating the current samples). Running the test for each alignment pattern group out of the 7 patterns covering 100% matches, convergence and divergence, Lamian test outputs all genes as DE for both convergence and divergence groups as expected. However, the Lamian test gives 288 false positive genes (~57.6%) (at 0.05 significance) for DE in the matching dataset of 500 genes. For 0.001 significance, this is still a 43.8% false positive rate, which is a quite significant false positive rate, compared to that of G2G which gives expected alignments (i.e. 100% matched alignments) for those genes with just 1.8% false positive rate. Beyond this level, **we are unable to compare Lamian DE test and G2G alignment, since they are addressing completely different computational problems and generating completely different outputs** (i.e. Lamian does not produce any mapping of time points as previously explained).

The current state of the art methods for trajectory alignment are CellAlign and TrAGEDy. Therefore in our manuscript, **we have comprehensively benchmarked G2G against both CellAlign and TrAGEDy** using simulated datasets through 3 different experiments,

	validating that G2G accurately captures expected alignment patterns that include complete-match, divergence, convergence and complete-mismatch. All comparisons are described in the Results section. (Please see Figure 3c-h, Extended Data Figure 2, Extended Data Figure 3 for all results). These cover the evaluation of both gene-level alignment, cell-level alignment and the clustering structures. We have also further shown how G2G identifies expected mismatches in two other published datasets from literature (i.e. PAM stimulated vs. LPS stimulated dendritic cell trajectories; healthy vs. IPF disease trajectories). Overall results clearly show that G2G outperforms the current state of the art in terms of the accuracy, as well as the level of information in the produced alignment outputs.
2.2 In the five-state model, there doesn't appear to be a state for situations where there is simply not enough information to be able to select a state (e.g. low or highly variable expression). How are these handled and is there a 'default' class? I imagine some genes may 'match' in the trajectory because they are equally unimportant in defining each trajectory. One could simulate and artificially downsample cells/counts to directly examine this behaviour.	As introduced on Page 5 and detailed in Page 45, the 5-state model is a finite state model which defines all 5 possible states of match and mismatch (i.e. 1-1 match, 1-many match, many-1 match, insertion, deletion) between two pseudotime points and their state transitions. Each state transition has a probability associated with it. This model controls the expected length of a match and mismatch (gap) in an alignment between two trajectories. i.e., equivalent to the sequence alignment analogy, this state model enforces gap (mismatch) penalties. The dynamic programming (DP) recurrence relations described in Page 46 incorporates cost for all possible state transition scenarios based on this model, and therefore there is no 'default class'. Together with the state transition costs, the matching and mismatching between time points is decided based on distributional distance of reference and query gene expression, computed with the DP scoring scheme. This distributional distance, as detailed in the methods section, quantifies the difference of expression (whether high or low) based on both mean and variance, regardless of whether low or highly variable expression. Users can control the parameters of the finite state model if they want to. Just like in DNA/protein sequence alignment where ad hoc gap penalties result in different alignments, changing the state model

parameters will also result in different alignments. (For more details, please see Durbin, Eddy et al (1998) biological sequence alignment textbook). We have optimized these parameters using our simulated dataset (since there is no ground truth trajectory dataset). We also note that this could be optimized by the user if a user happens to have a benchmark trajectory dataset where they are certain of matches and mismatches.

G2G generates alignments under the assumption that the datasets well-represents the complete trajectories and they have smooth trends of transcriptomic state change. G2G interpolation step performs the smoothing of gene expression data through a weighted approach considering all cells along the entire trajectory, as introduced on Page 6 and detailed on Page 39. Therefore if certain time points representations are missing (i.e. missing cells representing certain time points), the interpolation still estimates expression distribution for those time points based on the ongoing trend at that point (nearby cells contribute more to this estimation than far away cells as this is a weighted approach). If this is a genuine missing data scenario of cells under the same gene expression trend, interpolation ensures that we still account for the correct trend. However, if this is a scenario where the missing cells would have started a completely different trend or shown abrupt changes in their expression behavior, then our smooth trend assumption breaks down and the resultant alignment would not be accurate.

We have now added the following sentences in the Discussion on Page 15 to highlight the above limitation and the importance of having datasets that sufficiently represent the entire trajectories in comparison.

We also recommend users to inspect whether the cell density in each pseudotime trajectory represents the entire dynamic process well. As G2G assumes smooth trajectories for comparison, the estimated gene expression distribution of an interpolation pseudotime point is heavily influenced by the observed cells near the adjacent pseudotime points. When there are missing

	data (unobserved cells) with sudden changes in gene expression, this assumption breaks and may limit G2G from generating an accurate alignment.
2.3 It would be worth spelling out what is the drawbacks of any naive kind of implementation, e.g. calculating correlation or MI of binned expression along the two trajectories. Might it be in sensitivity of identifying differences, or in difficulty of interpretation?	Thank you for the suggestion; this is a good point to highlight. Naive comparison assumes a 1-1 direct mapping of all pseudotime points. Therefore we cannot infer warps (one-many/many-one matched timepoints) nor mismatches – insertions and deletions. This is particularly relevant when, e.g., cell development does not happen at the same speed, in vivo vs. in vitro. Calculating correlation is also based on mean expression only. Due to these limitations, many groups have preferred proper trajectory alignment (using DTW algorithms since 2003). We have now included the following sentence in the main text (Introduction) in Page 4 to highlight this point. In the contexts of warps/unobserved states, a non-alignment approach such as analyzing the correlation or mutual information along a discrete binning of pseudotime will be inaccurate, as it assumes only a one-to-one mapping of pseudotime.
2.4 The method relies extremely heavily on the trajectory of reference and query, and very little attention appears to be made on the choices that go into building a single cell pseudotime/developmental trajectory in the manuscript. Fundamentally this is estimated using the gene expression profiles, which can be given more or less importance via feature selection or weighting, therefore the effect of the G2G results can be driven by the choices taken a priori in the trajectory inference method. To what extent does the trajectory inference affect G2G results? Here, diffusion pseudotime is used, could an approach like monocle or slingshot also be appropriate?	Thank you for highlighting this point. G2G can be used to align pseudotime trajectories estimated using any available pseudotime inference method such as Diffusion pseudotime, Palantir, GPLVM, Monocle, Slingshot etc. In this manuscript, our scope is the computational alignment of already estimated single-cell pseudotime trajectories. Therefore we do not discuss all the different possible choices that go into inferring single-cell trajectories, as it is not in the scope of our work. However we strongly agree with the reviewer's point that the trajectory alignments are driven by the choices made a priori for trajectory inference. This is indeed true for any other computational model where the output depends on the input. As there are many combinatorial parameter settings, feature selection and data integration

strategies we can use to infer trajectories across many different tools, we see it as impractical to evaluate them all (especially given that we do not know the ground truth). Regardless of the different choices, if multiple pseudotime inference methods output similar pseudotime orderings, then we expect the results from G2G alignment to agree as well. All the choices we have made in our trajectory inference approaches follow the standard practice (i.e. tutorial guidelines and default settings of those tools – diffusion time and GPVLM), and therefore we believe that they represent the majority of the use cases.

Overall, we agree with the reviewer that it is indeed important to discuss this in the manuscript. Therefore we have now added a **Supplementary** section (Trajectory alignment across different pseudotime estimations) on **Page 101-104** to highlight the below points including recommendations to users for selecting a reasonable trajectory:

- *the trajectory alignment of datasets is heavily driven by the choices made a priori for their trajectory inference, and therefore it is important that the user initially verifies the reliability of their input trajectories, and validate the information gained from final gene-level alignments (experimentally, functionally and/or based on current literature) before coming to biological conclusions.*
- *Users could always go for a reasonable pseudotime distribution as long as it fairly represents the expected cell-type compositions along the entire trajectory. Users could refer to benchmark studies in literature such as the comprehensive review paper by Saelens et al (2019) and explore several trajectory inference approaches before deciding on the final pseudotime trajectories to align. Users can also use methods that allow the incorporation of time priors (e.g. the GPLVM approach we used for ATO pseudotime estimation based on the real sampling time points of the cells) to obtain reliable estimates.*

Additionally, we have added a sentence on **Page 15**, recommending users to explore different pseudotime estimators, as suggested by Reviewer 1 (Please see **Comment 1.5**).

We have also now tested pseudotime estimations from Monocle3, Slingshot, and Palantir in addition to DPT and GPLVM, on the pan fetal reference/ATO datasets and healthy/IPF datasets, and found that the final results remain consistent with what we originally reported. Accordingly, we have added the below text in the **Supplementary** section.

*To test the robustness of results from G2G trajectory alignment across the differences in the pseudotime estimates, we explored the tools: Monocle3, Slingshot, Palantir, and Diffusion pseudotime (DPT) (in addition to the GPLVM estimation reported in the main text) on our pan fetal reference and ATO datasets (for trajectories starting from the early DN stage onwards). We observed that only Monocle3 gave a reliable and reasonable pseudotime estimation for these datasets, with the expected cell type compositions fairly represented along pseudotime. (See **Figure S1** for the pseudotime density plots and cell type composition plots along time across all the pseudotime estimators). Accordingly, we re-performed G2G alignment using Monocle3 estimates and found that the TNFa pathway remains the one under which the most of the TFs are significantly different with an end mismatch, consistent with our results from using GPLVM based pseudotime estimates. Nevertheless, we see GPLVM estimates to be more reliable in this context, as we were able to incorporate time priors based on the real sampling time points of the ATO.*

*We also re-estimated pseudotime for healthy/IPF datasets using Monocle3, Slingshot and Palantir (in addition to DPT estimation reported in the main text) which gave expected cell type compositions fairly represented along pseudotime (See **Figure S2**). We then re-performed G2G alignment across all tools, resulting in gene-level alignments for which the most genes are*

	significantly different under the Epithelial Mesenchymal Transition pathway (as the top hit in the overrepresentation analysis), consistent with our results from using DPT estimates. (Note: We did not compare GPLVM for this, since there are no time priors available for the healthy/IPF datasets).
2.5 It would be helpful to characterise the concept of indels in a single cell trajectory context. In the sequence alignment context this is very well characterised as mutations/variations that insert or delete into the DNA/RNA sequence, but for a single cell trajectory this needs to be clarified much further - could this represent new transient cell states? or bypassing some state?	We have described the concept of indels (mismatches) on Page 3: A mismatch could either imply missing data unobserved state or differential expression (DE). For instance, a sudden rise or drop in expression of one system relative to the other might indicate that it is transitioning through a different cell state. A mismatch can also occur when a considerable fraction of cells in one condition has a significantly different distribution of expression for some genes compared to the other condition. Any deviation in expression (convergence/divergence) could mean a different cell state within the same cell type or a different cell type, or a bypassing case as well.
2.6 G2G handles two linear trajectories and at present cannot handle branched trajectories. One could consider a sixth state in the five-state alignment approach that is presence of a branching.	Thank you for the suggestion. Since the 5-state model controls the match mismatch occurrence, a branch state would mean to calculate match/mismatch scoring against both branch 1 and branch 2, which would result in many more possible combinations and would slow down the runtime massively. We have also commented on this in our Discussion section on Page 15: We are aware of other efforts in aligning branched trajectories with DTW based tree alignment⁹. Output from such alignments, i.e., identified pairs of correspondences, could be input into G2G for a comprehensive pairwise lineage alignment to capture mismatches. We hope to look into this further in our future work.
2.7 It's unclear whether the other approaches mentioned (DTW or CellAlign) fail to identify or describe the differences in the in vivo and in vitro T	As described on Page 6 and illustrated in Figure 3a-b, the limitation of any DTW approach (e.g. CellAlign) is that, it can only output a mapping where all time points

cell differentiation dynamics.	are matched regardless of the mismatched states. Thus CellAlign only generates an output describable by a 3-state string of all matches as below: MVVVVVMMWWVWWWVWWVWW, failing to identify the differences in the in vivo and in vitro T cell differentiation. For completeness, we also tested the high-dimensional alignment from TrAGEDy (which post-hoc processes the DTW alignment output to extract mismatches) under its two available distance measures. TrAGEDy with Spearman correlation (the measure recommended by the TrAGEDy paper) generates an alignment describable by the 5-state string: IIIIIMMMWWWWWMMMM, which captures only the early mismatches (corresponding to the pluripotency stage). On the other hand, TrAGEDy with Euclidean distance correctly captures both early and late mismatch stages, generating an alignment describable by the 5-state string: IIIIIMMMWWWWWVWWIDIDID. This output agrees with what G2G alignment captures as well. We have now highlighted this as below on Page 12: Independently, TrAGEDy¹⁶ high-dimensional alignment was also able to capture mismatches at the beginning and end, verifying the strong mismatch signal in early and late stages of in vitro, in vivo T cell differentiation. with a paragraph on Page 60 (Methods).
2.8 G2G is performed on each gene independently, and then per-gene match/mismatch sequences are clustered to identify groups of genes with a similar alignment pattern. It may be useful to combine genes according to a priori knowledge to increase robustness in the alignment results. This could be done by performing the denoising after some dimensionality reduction step.	Thank you. This is a very good point. G2G framework facilitates such custom alignment. A user can test the gene alignments and their single aggregate alignment for the a priori defined gene modules. This is commented on in the Discussion section on Page 14: The gene sets to compare can be either all genes, or restricted to gene sets of interest, e.g., TFs, regulons, highly variable genes or genes associated with a certain biological/signaling pathway. As suggested, the user could also provide results after dimensionality reduction steps, e.g. principal

components (PCs) from principal component analysis or latent variables from e.g., scVI² as input to G2G. We tested this with the scVI and PC embeddings of our pan fetal reference/ATO datasets, as well as the PC embeddings of the healthy/IPF datasets, and found that the aggregate result is consistent with what we originally reported for the respective studies. To further emphasize this point, we now include a **Supplementary** section (Trajectory alignment using low-dimensional, latent embeddings) on **Page 105** which reflects reviewer's suggestion and test results as below:

G2G currently aligns each specified gene independently, thus requiring a clustering of gene-level alignment strings to identify gene groups with similar alignment patterns. As noted in the Discussion, we can infer aggregate alignments for a priori known gene modules (e.g. biological pathway gene sets or regulons). In addition, we can also align dimensions of a low dimensional, latent embedding of the reference and query datasets rather than their actual gene dimensions. For instance, a user could input data to G2G after steps such as batch correction and dimensionality reduction, i.e., latent variables from an scVI embedding or principal components (PCs) from principal component analysis.

To test this, we aligned the 10 scVI latent dimensions of the pan fetal reference and ATO data using G2G in the same way as performed for the 1371 TFs in our reported study. The average alignment (IIIDIDIDMMMMMMMMDD) gives early mismatches and late mismatches consistent with the average alignment reported in the main text. Similar result (IIIDMMMMMMMMDD) was observed when aligning the first 50 PC components of the reference and query datasets. Separately, testing the PC component alignment of healthy/IPF datasets, G2G again outputs an average alignment (MMMMMIIDDDMMDD) of early matches and late mismatches consistent with the average alignment reported in the main text.

	Overall, the aggregated results from latent dimension alignment could act as cell-level alignment, and such a low-dimensional alignment process is more time-efficient than the current gene-level approach. However, the alignment result from each PC or latent variable might be difficult to interpret biologically, thus requiring further research to identify genes that contribute to the alignment pattern of each latent dimension. We hope to look into disentangling the genes associated with each latent dimension alignment in our future research work.
2.9 Unclear how symmetric the G2G approach is, if you swap reference and query, do you get consistent results? For the in vitro vs in vivo comparison, it's not clear which really should be reference and which should be query. How would a user decide how to allocate trajectories to reference/query?	The distributional distance measure we use is a symmetric measure, and the 5-state machine is a symmetric machine, as described in the Methods section. (Page 43 Line 28, Page 45 Line 18). Therefore the gene-level alignment cost computations are symmetric, which in turn makes the optimal alignment output symmetric (i.e. final alignment output will stay the same if the reference and query are interchanged). The only difference we will see is the swap of each insertion (I) and deletion (D) states to one another, and the swap of each expansion warp and compression warp states to one another, as these states are symmetrically defined relative to the reference and query. (i.e., a time point representation present in the reference but missing in the query is a D state, while a time point representation present in the query but missing in the reference is an I state). For example: the alignment string MMMMDDDDWWWIDIIIIII will change to its symmetric string MMMMIIIVVVVDDDDDDDD when the reference and query are interchanged. We have now re-emphasised this point in the main text on Page 6 as below: (Note: the alignment string will be symmetric regardless of which dataset is taken as the reference, due to the total cost of alignment being symmetric; the only difference will be the swap of the symmetric states: $I \leftrightarrow D$ and $W \leftrightarrow V$).

2.10 There is some speculation on the drivers behind the unexpected results related to Supplementary Fig. 3a, the authors could continue the simulation study to identify what are the drivers behind the unexpected behaviour, e.g. remove cells at different points along the trajectory.

Thank you for the suggestion. We see the below observation, consistently across both G2G and TrAGEDy results, implying that it is due to the inherent characteristics of the perturbed trajectories.

This is also evident from the inspection we did as follows: We split all gene alignments under the perturbation scenario of 20–40% deletion of bins, into 2 groups: 1. genes with alignment similarity % $\geq 80\%$ (which do not follow the expectation), and 2. genes with alignment similarity % $< 80\%$ (which follow the expectation). Then for each of 20%, 30% and 40% deletion cases, we looked at their distribution of the total alignment cost (in *nits*). As expected, we see low alignment costs for group 1, and high alignment costs for group 2. This difference implies that group 1 genes have low costs due to the presence of many matches, also consistent with the non-varying gene expression for many genes across pseudotime bins 10 to 20 that we see from the heatmap visualization in **Extended Data Fig. 3d**.

3d.

20% deletion case

30% deletion case

40% deletion case

For example, the gene *Uggt1* shows higher alignment similarity $\geq 80\%$ due to its non-varying gene expression region.

***Uggt1* without any perturbation**

***Uggt1* after 40% deletion from the start**

and therefore we have now noted this on **Page 8**:

For perturbation scenario 1, both G2G and TrAGEDy alignment similarity decreases with increasing deletion sizes as expected across smaller perturbation sizes, although the detected mismatch length is shorter than expected for deletions larger than 20%. This is ~~could be~~ due to the relatively non-varying gene expression between pseudotime bin 10 to 20 (Extended Data Fig. 3d), which causes warps instead of mismatches. Both G2G and TrAGEDy alignments are consistent in capturing this behavior which is inherent to the perturbed trajectories.

While we could also continue perturbing the trajectories by removing cells at different time points, we cannot

	guarantee a single, specific alignment pattern (i.e. unable to define an expected alignment for such perturbation) for each gene, thus it is difficult to evaluate their accuracy. For this reason we had originally run the experiment only by perturbing the start terminal of the trajectories.
--	---

Reviewer #3	
Comment by the reviewer	Response by the authors
3. Reviewer 3 The manuscript “Gene-level alignment of single cell trajectories” by Sumanaweera et al offers a novel algorithm for aligning pseudotime profiles between different conditions or datasets. It seems like an exciting approach and it tackles an important question (would have been of help in numerous of our own datasets over the years) in what I believe is a novel way that makes a lot of sense conceptually. Also test their methods on lots of data, which is always good to see. There are a few comments (mostly minor, a few a bit more major), however, that I think should be addressed before publication (listed below more or less in the order I came across them in the paper, so minor and less minor ones are mixed in).	Thank you very much for your accurate summary and positive evaluation of our work. We are also extremely thankful for your very useful suggestions on improving our manuscript.
3.1 Maybe add a few words about the choice of normalization used for the algorithm and if it matters or was somewhat arbitrary (log TPM vs log TP10K vs log counts vs others)—at the very least not clear from the main text which normalization is used (based on the methods think it is TP10K, yes?). Related, talk about log1p normalized expression, should make clear what is being log scaled (TP10K?). At the very least figure legends should be explicit about how the data was normalized, not just say ‘normalized expression	Thank you for pointing this out. We have now added the definition of log1p normalization on Page 5: G2G’s inputs are log1p normalized (raw transcript counts are normalized over total counts of all genes per cell so that every cell has the same total count after normalization, then logarithmically transformed with $\log(\text{normalized count} + 1)$) scRNA-seq matrices of some reference and query, and their pseudotime estimates. Normalizing to 10K or 1 million would not change the results of the alignment, since it is consistently done for both reference and query datasets.

	We have now clarified in all relevant figure legends how the data was normalized, by adding the below text following the text log1p normalized: (i.e. per-cell total sum of the raw transcript counts normalized to 10,000 and log1p transformed)
3.2 Since the alignment is at the gene level, would be good to see how gene level meta data (most notable expression level) might affect the results.	The matching cost of two time points is driven by the MML distance computed between the estimated gene expression distributions of the corresponding time points, regardless of whether the expression level is high or low. To further demonstrate this, we plotted the maximum mean of gene expression (the highest mean gene expression out of reference and query) along the trajectories versus the alignment similarity percentage (colored by group of alignment pattern) across all alignments of our simulated experiment 1 dataset, grouped into 3 plots based on their main alignment pattern (100% match, convergence and divergence, respectively). 
	Overall, these plots show that the alignment similarity percentages are following the expected patterns regardless of the notable levels of average gene expression, i.e., the alignment accuracy is not influenced by whether the average gene expression level is high or low.
3.3 Very minor, but should mention number of cells in experiment 2 when first talking about dataset in results section.	Thank you for your suggestion. We have now added the cell number in the Results section on Page 8: To test how G2G detects matching patterns in real scRNA-seq data, we used a dataset of E15.5 murine pancreatic development³, subsetted to cells in the beta-cell lineage (1845 cells) and considered 769 genes varying in expression during beta-cell differentiation.
3.4 More major: Experiment 2 gives useful information, but in practice when aligning cells they will come from different 10X channels, individuals, etc, not just be one dataset split in half. Would be great to see the results when you split by batch instead of at the cell level. If not realistic at least need to discuss.	This is a very good point. Many thanks for the suggestion in this comment, as well as in the follow-up Comments in 3.5 and 3.7 regarding batch/confounding variable effects. Our responses for all these 3 comments are related as they address how G2G alignment is affected by batch variability, and further recommendations to users. We now include a Supplementary section on Page 105-108 to discuss Alignment in the presence of batch effects across points arising in our following responses to Comment 3.4, 3.5 and 3.7 together. G2G accounts for the batch effects (e.g. from different 10X channels, individuals etc.) within a single trajectory by modeling gene expression at each time point as a Gaussian distribution. This means, the variance of the

	estimated distribution accounts for the possible spread from different batches. However, G2G does not model confounders nor correct for batch effects between the two reference and query trajectories. We agree with the reviewer that it is important to discuss how a batch split could affect G2G alignment. To test this, we performed a stability test on both healthy/IPF datasets and pan fetal reference/ATO datasets, in terms of the individuals (donors as batch). In the stability test, we ran multiple iterations of G2G alignment, taking only a random 50% subset of donors for alignment at each iteration. The resultant alignment from each iteration was consistent with our reported alignments that considered all donors, and therefore we conclude that G2G alignments are robust despite the changes in the donor batch compositions, and the overall alignment results were not influenced by any outlier individual. We now describe this test in the relevant Supplementary section on Page 106 (as detailed in our response to Comment 3.7). We also recommend that: A user could also perform a similar stability test across alignment using different random batch subsets, to ensure robustness of the final conclusions. We also note that: even though in theory we expect 100% matched trajectories between individual batches that represent the same system (different 10X channels, individuals etc), we cannot guarantee such alignment when aligning two such batch trajectories. This is because they may have differences in their trajectory representations (cell-type compositions) and/or large batch effects caused by biological/technical confounding variables. Therefore it is difficult to benchmark batch alignments without knowing the expected (true) matching and mismatching behaviors across pseudotime. However G2G can be a tool to identify such differences between batches and support evaluating batch effects.
3.5 Related to the above, can confounders be taken into account? Either by regressing them out before hand or in the model itself? The methods seem to hint at this possibility but would be great to see examples.	As previously described, G2G accounts for the batch effects within each single trajectory, but does not model confounders nor correct for batch effects between the two reference and query trajectories in comparison. Regressing out or correction of batches/confounders might overregress/overcorrect and obscure true

	biological differences. Therefore, identifying all mismatches and investigating them in downstream analysis will be a much safer approach. However if a user is aware of batch/confounder effects and prefers a particular method which corrects gene expression for those batch/confounder effects, they could input gene expression data processed using that particular method prior to G2G alignment. Alternatively, we can also align the latent representations outputted from integration methods such as scVI that can model confounders and batch effects. This means, instead of actual genes, we are aligning the latent features obtained through batch correction and dimensionality reduction. To test this, we used the 10 dimensional scVI latent embedding representation we obtained previously for the pan fetal reference and ATO dataset in our T cell development study. This scVI embedding was obtained by correcting for in vivo donors and the in vitro cell lines (which ultimately corrects for the batch effects between the reference and query systems as well). We aligned the 10 latent features using G2G in the same way as previously done for the TFs in our study. This latent dimensional alignment inferred the average alignment between in vivo and in vitro trajectories as: IIIDIDIDMMMMMMMMDD, describing early mismatches and late mismatches, which was consistent with our previous average alignment output from aligning the 1371 TFs. We now include the above points and present this as an example in the Supplementary on Page 105 to show that trajectory alignment could be done not only for batch uncorrected data but also for batch corrected data. However, we also note that it is difficult to biologically interpret an alignment done on latent dimensions, and highlight that this is an open area for future research. (This is also highlighted in our response to Comment 2.8 related to latent, low dimensional alignment).
3.6 Very minor, but for healthy vs IPF mention number of individuals in each group.	We have now specified the number of individuals in each group in the Methods section, Page 56, as below. This includes 3157 healthy cells (2655 AT2, 502 AT1) coming from 28 individuals, and 890 IPF cells (442 AT2, 448 ABCs) coming from 31 individuals.
3.7 More major: The healthy vs IPF example is great, my one concern is that the approach does not take into	Many thanks for raising this point and for the suggestions. As also highlighted in our response to

account individual to individual variability (unless I am missing it?). Given that such variability is known to lead to inflated DE and cell type composition results if not taken care of, would be good to know how it effects the method when comparing between groups (I am particularly worried about in cases with more subtle changes). Would be great to address this somehow. Some ideas (don't need to do these particular approaches, should just do something): one possibility would be to take a dataset (maybe the IPF dataset, maybe just the controls from it?), split it in half (assign half the individuals as cases and half as controls), and see what kinds of significant results one gets trying to align them. Another approach could be some kind of down sampling (sample one or a few individuals from each condition and do the analysis just on them and repeat the analysis (with a few such splits), see if the results hold up). Note having false positives here isn't make or break (can often do follow up experiments as you do with the TNF bit later), just important to know about when interpreting results/deciding how much faith to give them (particularly in the absence of follow up experiments).

Comment 3.4 and 3.5 above, G2G accounts for the batch effects (including individual-to-individual variability) within each trajectory. We have now emphasized this in the newly added **Supplementary** section on **Page 105**:

A key feature of G2G is that it models gene expression at a particular time point t in a trajectory as a Gaussian distribution, taking into account both mean and variance of gene expression as weighted estimates based on all the cells in the trajectory. (i.e. the cells in time points closer to t have greater contributions towards the estimation, whereas cells in far away time points have lesser — and if they are too far, almost negligible — contributions towards the estimation). Thus, for a given particular trajectory dataset (which represents either a reference or query system), the estimated gene expression distribution corresponding to time point t represents the variance caused by unknown batch effects and other confounders (donor-to-donor variability, technology, sample etc.). When there is a small batch effect, we expect low variance in the estimated Gaussian which mostly represents the natural variance of RNA expression amongst the similar type of cells. If there are large batch/confounder effects, then we expect a much higher variance in the estimated Gaussian.

However, G2G does not model batch effects between the two trajectories in comparison, and we have now added the below text in the same section to discuss this point.

Modeling gene expression as distributions enables G2G to account for the batch effects within the same system (either reference or query). However, G2G does not model or correct for batch effects between the reference and query systems. Batch correction may overcorrect and obscure true biological differences, thus the approach of identifying all mismatches and investigating them in downstream analysis is a much safer choice. However if a user prefers a particular batch correction method, they could input data batch corrected (or confounders regressed) between reference and query, and then perform G2G alignment. Particularly, as described in the previous section, we can perform a G2G

alignment between the batch-corrected latent dimensional embeddings (generated using a tool such as scVI) of the reference and query datasets.

We strongly agree with the point made by the reviewer on the possibility of inflated DE when comparing two trajectories that have a larger batch effect that contributes to the majority of the differences in gene expression across time. One approach to overcome this (as also described in our response to Comment 3.5), is by using batch-corrected gene expression data or batch corrected latent representations for alignment.

As suggested by the reviewer, **to test how the G2G method is affected by individual-to-individual variability in trajectories**, we performed a stability test on IPF/healthy datasets alignment, as well as on our *in vivo*, *in vitro* T cell datasets alignment (also addressing Comment 3.4). Across all iterations of random donor subsetting, the resultant G2G alignments were consistent with what we originally reported using all donors. We now describe this test in the relevant Supplementary section on Page 106-108 as below:

Stability test on healthy/IPF datasets: We performed a leave-50%-donors-out stability test on the healthy and IPF datasets. There were 28 donors and 31 donors in the healthy and IPF datasets, respectively. For the random sampling experiment, we considered only the 12 healthy donors and 7 IPF donors who had sufficient data (i.e. ≥ 50 cells in each donor batch). We then randomly sampled 50% donors for 5 times, independently for healthy and IPF, creating 10 random donor subsets (including both, the sampled 5 subsets as well as their complementary subsets). Next we performed G2G alignment for all 10 donor subset pairs between healthy and IPF under the same setting of parameters and 994 genes we had for our previous alignment reported in the manuscript main text. We observed that the overall alignment pattern of early match and late mismatch which we found from our previously reported alignment is consistent across all these new 10 alignments as well

We also repeated the healthy vs IPF G2G alignment for 10 iterations by downsampling each dataset (i.e. by 50%

	subsampling of cells) and again observed that the overall alignment pattern of early match and late mismatch is consistent Please see Page 106-107 for alignment results. Stability test on in vitro/in vivo T cell datasets: Additionally, we applied the same donor subset sampling strategy to the pan fetal reference dataset where we had 22 donors with sufficient data (>=50 cells) out of 33 donors. With 10 random donor subsets (each having 50% donors = 11 random donors), each reference random donor subset was aligned against the complete ATO dataset (Note: ATO dataset does not contain donors, but just 2 cell lines, therefore it was not downsampled). We again observed that the overall alignment results across all the 10 alignment iterations (as shown below) are consistent with our previously reported alignment (i.e. early mismatches and late mismatches) of the complete dataset alignment. Please see Page 107 for alignment results. Based on the above described stability tests, we conclude that the G2G method was still able to produce consistent alignment results despite the changes in donor compositions, and the overall alignment results were not influenced by any outlier individual.
3.8 In the organoid vs in vitro data comparison a lot of the mismatched genes you display (a lot of the TNF related ones) are immediate early genes (as you kind of get at with the comment about stress activation in the supplement). Given that such genes are known to show up due to more technical reasons (handling, etc), and do so differentially between different cell types (https://www.nature.com/articles/s41593-022-01022-8 [nature.com]) might want to add a few words discussing why you think this is a real signal and not an artifact of some type (might be obvious but if so I missed it). You get close to addressing this in the supplement when talking about the relation to stress	Thank you for raising this important point. In the literature, TNFα activation of NFκB pathway has been implicated in the final functional maturation of murine T cells within the thymus^{4,5}. To further validate that this is not related to ex vivo stress and handling artifacts during single cell dissociation from tissue, we followed the recommendation from Marsh et al. 2022⁶ and investigated orthogonal in situ methods such as spatial transcriptomics. We took the Visium spatial transcriptomic data of 3 fetal thymic slides from our developing human immune atlas⁷ and compared the TNFα pathway gene expressions between cortex (where T cell progenitors are) and medulla (where mature T cells are) regions. The annotation of tissue

but would like a little more detail on how it might (or might not) relate to handling in particular.	regions on Visium spots were inferred by clustering of H&E image features⁷. We selected the TFs in TNFα pathway that showed increased expression at the end of the in vivo T cell development but no increase in in vitro T cell development in Fig. 6d, together with mature T cell marker CD27 and T cell progenitor markers RAG1 and RAG2. Results are shown in Supplementary Table 10. TFs in TNFα pathway have higher expression in the medulla than cortex, which is consistent across all 3 slides, corroborating with what we have observed in the scRNA-seq data with the increase in expression at the end of in vivo T cell development. As expected, CD27 expression is higher in medulla than cortex, whereas RAG1 and RAG2 have higher expression in cortex than medulla. We have now included this result in the Results Section on Page 12 as well as in the Supplementary Note on Page 97-98 (Spatial variation in TNFα pathway genes in thymus).
3.9 Add more details on what the change point kernel plots are to the legend for extended fig 1 (Since not a common plot)	We have now defined the change point kernel and the heatmap plot in the Extended Data Figure 1 legend as follow: (Note: A change point kernel defines shifts and changes in covariance between discrete time points in a time series that describes a particular Gaussian process. In the context of a single-cell pseudotime trajectory, each discrete time point corresponds to a single cell. The change point kernel can be represented by a pairwise covariance matrix between those time points, visualized using heatmaps).
3.10 For legend for extended data fig 2a/b explain what the color boxes mean	We have now defined the color boxes in the Extended Data Figure 2a-b as follow: The colored boxes (in blue, orange, and green) in the two leftmost columns display the possible range of expected match lengths corresponding to the three different, approximate bifurcation points: [0.25, 0.5, 0.75], respectively.
3.11 What kind of runtime is involved? Saw some big-O type analysis, but unless I missed it no benchmarking of runtime? Would be good to have some info on that.	Thank you for the suggestion. Compared to CellAlign and TrAGEDy, the current version of G2G demands more computational time as the number of genes increases and/or the number of interpolation time points increases and/or the number of cells in the scRNAseq

	datasets increase. This is mostly because G2G performs a DP alignment for each gene, interpolating across all cells and computing distributional distance for each pair of interpolation time points, which goes beyond the quadratic time complexity expected from the standard dynamic programming alignment algorithm. However G2G currently utilizes parallel processing through Python multiprocessing to speed this up. It creates processes equal to the number of cores in the system, and each process performs a single gene-level alignment at one time. We have now included information about this on Page 63 as specified in our response to Comment 1.2.
3.12 For the analysis of the data (the standard clustering, pseudotime, etc) make sure the versions of the packages used and the non-default arguments for those tools (if any) are given, or the associated scripts are available on your github.	We have now added version numbers of the following relevant packages used for the analysis. using the leven python package v1.0.4 on Page 49 Python sklearn.cluster function in package sklearn v1.2.2 on Page 49 scipy.spatial.distance.cdist (in SciPy package v1.10.1) on Page 49 We use the GSEAPy (package v1.0.4) Enrichr^{71,73,74} wrapper on Page 51 optBinning⁷⁷ python package (v0.18.0) on Page 52 (CellAlign v0.1.0) on Page 52 We used GPyTorch package (v1.5.1) for implementing the code for these simulations. on Page 53 Diffusion Pseudotime³⁸ implemented in SCANPY⁶⁰ (v1.9.6) on Page 56 statsmodels package (v0.13.5) on Page 62 All associated scripts are available in the form of Jupyter notebooks at: https://github.com/Teichlab/G2G_notebooks
3.13 Tested the code for the package a bit. Got it to install and performed the alignment. The one issue I had was with visualizations that did not seem to work with the system I was using (got the error "qt.qpa.plugin: Could not load the Qt platform plugin "xcb" in "" even though it was found."). Once I updated Seaborn to a newer version, however, it	Thank you very much for testing the code, and for letting us know about this error due to Seaborn version differences. We have now updated the pyproject.toml file to specify the minimum Seaborn version requirement, in our Git repository: https://github.com/Teichlab/Genes2Genes. We also recommend the user to create a new Conda environment and install G2G to avoid any version conflicts and

started to work (might be useful for others having similar issues). Might be worth putting a warning on the github or add something to require more up to date Seaborn installation (not sure if this is a wide spread issue or specific to the system I was on).

dependency issues.

References:

1. Saelens, W., Cannoodt, R., Todorov, H. & Saeys, Y. A comparison of single-cell trajectory inference methods. *Nat. Biotechnol.* **37**, 547–554 (2019).
2. Lopez, R., Regier, J., Cole, M. B., Jordan, M. I. & Yosef, N. Deep generative modeling for single-cell transcriptomics. *Nat. Methods* **15**, 1053–1058 (2018).
3. Bastidas-Ponce, A. *et al.* Comprehensive single cell mRNA profiling reveals a detailed roadmap for pancreatic endocrinogenesis. *Development* **146**, (2019).
4. Webb, L. V., Ley, S. C. & Seddon, B. TNF activation of NF- κ B is essential for development of single-positive thymocytes. *J. Exp. Med.* **213**, 1399–1407 (2016).
5. Xing, Y., Wang, X., Jameson, S. C. & Hogquist, K. A. Late stages of T cell maturation in the thymus involve NF- κ B and tonic type I interferon signaling. *Nat. Immunol.* **17**, 565–573 (2016).
6. Marsh, S. E. *et al.* Dissection of artifactual and confounding glial signatures by single-cell sequencing of mouse and human brain. *Nat. Neurosci.* **25**, 306–316 (2022).
7. Suo, C. *et al.* Mapping the developing human immune system across organs. *Science* **376**, eabo0510 (2022).

Decision Letter, second revision:

Dear Sarah,

Thank you for submitting your revised manuscript "Gene-level alignment of single cell trajectories" (N METH-A54183B). It has now been seen by the original referees and their comments are below. The reviewers find that the paper has improved in revision, and therefore we'll be happy in principle to publish it in Nature Methods, pending minor revisions to satisfy the referees' final requests and to comply with our editorial and formatting guidelines.

TRANSPARENT PEER REVIEW

Please note: we allow redactions to authors' rebuttal and reviewer comments in the interest of confidentiality. If you are concerned about the release of confidential data, please let us know specifically what information you would like to have removed. Please note that we cannot incorporate redactions for any other reasons. Reviewer names will be published in the peer review files if the reviewer signed the comments to authors, or if reviewers explicitly agree to release their name. For more information, please refer to our FAQ page.

ORCID

Sincerely,
Madhura

Madhura Mukhopadhyay, PhD
Senior Editor
Nature Methods

Reviewer #1 (Remarks to the Author):

The authors have addressed all of my concerns. Congratulations on this nice work!

Reviewer #2 (Remarks to the Author):

Thank you to the authors for their careful responses and actions addressing my comments. I appreciate the work in clarifying the G2G approach with differential trajectory analysis, depending on the biological context, these can have significant overlap and making such distinction is helpful for the research community. I also appreciate the further work in checking robustness of G2G with respect to different choices of trajectory estimation methods.

In response to point 2.1, the authors make clear that G2G does make the assumption of non-linear mapping. This I did not doubt, and apologies for lack of clarity in the original review comment. The original comment actually is about the assumption of there being a *single mapping* between trajectories, as opposed to multiple mappings that are generated for each gene. In their response, the authors make it clear that, while multiple per-gene mappings are generated, these are averaged and result in what G2G aims as a single alignment.

It's still unclear to what extent one needs to perform feature selection for G2G to perform reliably. Since G2G performs DTW alignment per-gene and then an average mapping, it suggests G2G may suffer in quality if too many uninformative genes are retained as input to G2G. This can be explored and, depending on the observations, highlighted as a strength of G2G, or further information for analysts to beware when applying the method.

In the revised text, "We observed that only Monocle 3 gave a reliable and reasonable pseudotime estimation... Nevertheless we see GPLVM estimates to be more reliable in this context, as we were able to incorporate time priors based on the real sampling time points of the ATO". I'm not sure I follow the logic here, is it that the authors simply select GPLVM as the method is flexible enough to take time priors as input, despite making observations that Monocle3 is more reasonable? Perhaps there is some additional information that could be provided here to make a bit more clear the motivation for selecting GPLVM.

Reviewer #2 (Remarks on code availability):

I successfully installed the software and ran the tutorial provided on github.

Reviewer #3 (Remarks to the Author):

The authors have done a great job addressing my concerns. Great work!

Reviewer #3 (Remarks on code availability):

When I tested during the last round of reviews it worked well.

Author Rebuttal, third revision:

Response to referees

We thank you and all the reviewers for your positive evaluation, thoughtful comments and suggestions, which helped us to improve the quality of the manuscript during revision. We have responded to each new comment in the point-by-point table below, followed by a table which lists slight updates to several descriptions in the revised manuscript Methods section.

Please note: In the response column, we have used *<black, italic text>* to refer to any originally existing text in our manuscript, and *<blue, italic text>* to refer to any newly added/modified text as part of revision. We used **<bold, yellow highlighted text>** to refer to any page numbers/figures/sections in the manuscript that got revised and/or are relevant to our responses.

Reviewer #1	
Comment by the reviewer	Response by the authors
1. Reviewer 1 The authors have addressed all of my concerns. Congratulations on this nice work!	Thank you very much.

Reviewer #2	
Comment by the reviewer	Response by the authors
2. Reviewer 2 Thank you to the authors for their careful responses and actions addressing my comments. I appreciate the work in clarifying the G2G approach with differential trajectory analysis, depending on the biological context, these can have significant overlap and making such distinction is helpful for the research community. I also appreciate the further work in checking robustness of G2G with respect to different choices of trajectory estimation methods. In response to point 2.1, the authors make clear that G2G does make the assumption of non-linear mapping. This I did not doubt, and apologies for lack of clarity in the original review comment. The original comment actually is about the assumption of there	Thank you very much for your positive evaluation of our revised manuscript and further suggestions on adding clarity to the text regarding gene feature selection and our pseudotime inference method selection. We have now updated our text to discuss those two points as detailed in our responses below.

being a *single mapping* between trajectories, as opposed to multiple mappings that are generated for each gene. In their response, the authors make it clear that, while multiple per-gene mappings are generated, these are averaged and result in what G2G aims as a single alignment.	
2.1 It's still unclear to what extent one needs to perform feature selection for G2G to perform reliably. Since G2G performs DTW alignment per-gene and then an average mapping, it suggests G2G may suffer in quality if too many uninformative genes are retained as input to G2G. This can be explored and, depending on the observations, highlighted as a strength of G2G, or further information for analysts to beware when applying the method.	We agree that G2G average mapping indeed completely depends on the input genes. The strength of G2G is that it goes into clustering genes based on their individual alignments rather than focusing only on the average mapping. In such cases, the user can inspect different gene clusters to extract the genes that are biologically relevant/ interesting and meaningful, while ignoring the uninformative genes, e.g., housekeeping genes that do not vary in the trajectory and show matches between the reference and the query. However, the user can also select the specific gene sets of interest relevant to the context of their study (or the biological questions they try to answer), to obtain more informative results. For instance, given a cell lineage, they can first identify which top genes are driving the reference lineage (i.e. the highly significant, upregulated and downregulated genes) using a single-cell trajectory analysis framework such as CellRank, and then align those genes between the reference and query to check whether the query follows the same dynamics or not. In general, the user can align all transcription factors or all the highly variable genes. We have now added the below text in the Discussion The distribution of alignments can inform gene clusters with broadly similar alignment patterns, and their average alignments. Since such aggregated results depend on the genes we choose to align, we recommend to select genes as informative as possible (e.g. lineage-relevant driver genes or regulons). For instance, we can align the significantly upregulated and downregulated genes in the reference to investigate whether the query follows the same dynamics. Aligning TFs can inform differential regulation. When aligning all or highly variable genes, we can inspect gene clusters (e.g. paired with over-representation analysis) to extract biologically-meaningful groups, e.g.,

	revealing biological/signaling pathways that drive mismatches at different times.
2.2 In the revised text, "We observed that only Monocle 3 gave a reliable and reasonable pseudotime estimation... Nevertheless we see GPLVM estimates to be more reliable in this context, as we were able to incorporate time priors based on the real sampling time points of the ATO". I'm not sure I follow the logic here, is it that the authors simply select GPLVM as the method is flexible enough to take time priors as input, despite making observations that Monocle3 is more reasonable? Perhaps there is some additional information that could be provided here to make a bit more clear the motivation for selecting GPLVM.	The main reason for selecting GPLVM as reliable in this context is due to its ability to incorporate the actual sampling time points of the organoid cells as time priors for pseudotime estimation, unlike Monocle which does not facilitate to do so. Further, when we inspect the pseudotime distributions in Figure S1, we see that the GPLVM estimates have a more evenly spread cell density across the time points than those of the Monocle3 estimates. Overall, our work highlights how GPLVM is useful for estimating pseudotime of cells from organoids which have been sampled at known (multiple) time points. To avoid confusion, we have now modified the relevant sentences in the Supplementary Note as below to make the motivation for selecting GPLVM more clear. We observed that Apart from GPLVM, only Monocle3 gave a reliable and reasonable pseudotime estimation for these datasets, with the expected cell type compositions fairly represented along pseudotime. Nevertheless, we take see GPLVM estimates to be more reliable in this context, because unlike Monocle3as, GPLVM enabled us to we were able to incorporate time priors based on the real sampling time points of the ATO. Further, inspecting the cell density across pseudotime, GPLVM gives more evenly distributed estimates than the estimates from Monocle3 (Figure S1).

Reviewer #3	
Comment by the reviewer	Response by the authors
3. Reviewer 3 The authors have done a great job addressing my concerns. Great work!	Thank you very much.

Further revised points in the manuscript

We note the following minor updates to several descriptions in our revised manuscript **Methods** section. The first two were previously missed to be updated from an old setting tested during our experiments, and now refer to the relevant setting used to generate results reported in our manuscript.

Type of update	Updated text
Description of the technique used to simulate insertions in perturbation scenario 2 of our simulated experiment 2. (This was previously missed to be updated from an old setting tested during our experiments).	To simulate mismatches of n bins, we found the pseudotime bin with highest mean expression for the gene of interest in the query cells, and calculated mean (max_mean) and standard deviation (max_std) of expression of query cells for this bin; then, for each of the first n pseudotime bins, we substitute expression values of the query cells with a sample from a normal distribution with mean = max_mean + max_std and standard deviation = max_std. To simulate insertions of n bins, we shifted the query cell expression of the first n bins by the standard deviation calculated across all bins for the gene of interest in the query cells. Methods ('Simulating mismatches on real scRNA-seq data' subsection)
Description of the Parameter-setting used for initial alignment state costs. (This was previously missed to be updated from an old setting tested during our experiments).	Note: For the cases of  (i.e. before the first state transition), we assign either a uniform transition cost: $I(M) = I(I) = I(D) = -\log_e(1/3)$ or a setting with lower cost for M can be assigned. Methods ('DP recurrence relations' subsection) For initial states, we use $[Pr(M)=0.99 \text{ and } Pr(D) = Pr(I) = 5 \times 10^{-5}]$. Methods ('Determining the best parameter setting' subsection)
New sentence added to refer to the exact original data file containing the ABC-specific marker gene list we used from Adams et al (2020) paper.	ABC-specific marker genes were obtained from the original paper's²¹ supplementary file aba1983_data_s2.txt. Methods ('Dataset for Healthy versus IPF case study' subsection under 'Datasets')
New sentence added to refer to the latest version v0.2.0 of our framework which is much faster than v0.1.0 used for work in the paper.	A significantly faster, latest Genes2Genes v0.2.0 is now available, running sequentially with 0.60 seconds per-gene for the same case (SMP Debian 5.10.216-1 (2024-05-03)) Methods ('Software and computational requirements')

Final Decision Letter:

Dear Sarah,

I am delighted to inform you that your Article, "Gene-level alignment of single cell trajectories", has now been accepted for publication in Nature Methods. The received and accepted dates will be Nov 9, 2023 and Jul 12, 2024. This note is intended to let you know what to expect from us over the next month or so, and to let you know where to address any further questions.

Over the next few weeks, your paper will be copyedited to ensure that it conforms to Nature Methods style. Once your paper is typeset, you will receive an email with a link to choose the appropriate publishing options for your paper and our Author Services team will be in touch regarding any additional information that may be required. It is extremely important that you let us know now whether you will be difficult to contact over the next month. If this is the case, we ask that you send us the contact information (email, phone and fax) of someone who will be able to check the proofs and deal with any last-minute problems.

Please note that *Nature Methods* is a Transformative Journal (TJ). Authors may publish their research with us through the traditional subscription access route or make their paper immediately open access through payment of an article-processing charge (APC). Authors will not be required to make a final decision about access to their article until it has been accepted. Find out more about Transformative Journals

You may wish to make your media relations office aware of your accepted publication, in case they consider it appropriate to organize some internal or external publicity. Once your paper has been scheduled you will receive an email confirming the publication details. This is normally 3-4 working days in advance of publication. If you need additional notice of the date and time of publication,

please let the production team know when you receive the proof of your article to ensure there is sufficient time to coordinate. Further information on our embargo policies can be found here: <https://www.nature.com/authors/policies/embargo.html>

If you are active on Twitter/X, please e-mail me your and your coauthors' handles so that we may tag you when the paper is published.

Best regards,
Madhura

Madhura Mukhopadhyay, PhD
Senior Editor
Nature Methods